# Fixed-Point Probing for GNN Depth Diagnostics: A Geometry-Consistent Protocol with a Patent-Citation Case Study

## Abstract

Deep graph neural networks (GNNs) can lose predictive performance as depth increases, but final-layer performance alone does not show how their internal representations fail. We propose fixed-point probing, a post-training evaluation workflow that reuses the same nodes and edges at every layer and jointly measures interlayer displacement, edgewise variation, class structure, and, for hyperbolic models, proximity to the Poincaré-ball boundary.

On a large temporally split patent-citation graph, the operational full-sweep summaries are nonmonotonic. A complete-test aligned audit at $L=32$, restricted to width-preserving hidden transitions, yields non-overlapping drift-ratio ranges across the tested training seeds and node subsets: Euclidean late/middle ratios lie in 0.011–0.330, whereas the HYPA/HYPT ratios lie in 0.972–1.001 after early boundary saturation. Fixed-edge energy also contracts from the middle to the late hidden layers in all three Euclidean seeds. Together with endpoint collapse and late contraction of the class-dispersion components, the Euclidean audit supports an oversmoothing-like interpretation; the hyperbolic audit supports a bounded boundary-associated interpretation. The audit does not make the full depth sweep intrinsic or establish an invariant onset layer.

A complete three-seed OGBN-Arxiv instantiation provides a complementary public-benchmark result. Boundary saturation coexists with useful performance at $L=16$, while the reproducible $L=32$ endpoint collapse is accompanied by different validation-selected diagnostic states across seeds. Thus neither endpoint performance nor any single probe is used as a failure label. The contribution is a reproducible joint diagnostic workflow, not a universal numerical threshold or failure mechanism.

## 1 Introduction

Deep graph neural networks (GNNs) often lose predictive performance as depth increases, but a final-layer score answers only whether a trained model works at its endpoint. It does not reveal whether the internal representations stabilized, contracted toward an oversmoothed state, or entered a geometry-specific high-variation regime. After training, a depth-$L$ message-passing network induces a sequence of representations $\{h^{(0)}, \ldots, h^{(L)}\}$; we analyze this sequence as an observed trajectory rather than treating depth only as a model-size hyperparameter.

We introduce *fixed-point probing*, a fixed-support post-training workflow for comparing such trajectories. The same probe nodes and evaluation edges are reused across candidate depths, geometries, seeds, and ablations. On those fixed supports, the workflow jointly reports interlayer drift, metric-aware Dirichlet energy, label separability, and, for hyperbolic representations, boundary statistics. Here *fixed point* means an observed low-drift plateau under an aligned measurement (Section D.1), not an exact fixed point of a

---

A large language model was used for limited writing assistance (English copyediting and phrasing suggestions) during manuscript preparation. The scientific ideas, technical claims, experimental design, analyses, and final verification were performed by the authors.

dynamical system. In the aligned $L=32$ patent audit, the workflow supports contraction/oversmoothing-like and boundary-amplified outcomes rather than the stable-plateau outcome, so *fixed-point probing* names the diagnostic question rather than a presupposed result. The methodological invariant is the evaluation support, not an assumption that every layerwise coordinate map is intrinsically aligned; Protocol 1 states the outputs and stability checks.

Geometry makes this distinction important. In the Poincaré ball, the local metric expands near the boundary (Nickel & Kiela, 2017; Ganea et al., 2018), so modest ambient-coordinate changes can correspond to large geodesic displacements. A useful diagnosis must therefore read geodesic drift and graph-local variation together with boundary proximity and an ambient-coordinate audit rather than comparing raw Euclidean and hyperbolic magnitudes.

The patent citation graph is our main stress test because it is large, temporally split, many-class, and plausibly compatible with hierarchical representations. At $L=32$, all three model families reach similarly poor endpoints but have different joint signatures. The aligned audit covers all three GraphSAGE modes. Across the tested full-split seeds and node subsets, the Euclidean drift-ratio range (0.011–0.330) and the hyperbolic range (0.972–1.001) do not overlap; fixed-edge energy also shows middle-to-late contraction in all three Euclidean seeds. The full-sweep common-space values remain operational, but the $L=32$ contrast no longer relies on layer-indexed drift alone. A tuned control and a radius-targeted perturbation bound the hyperbolic interpretation.

The public-benchmark evidence supplies the complementary result needed to interpret that case study. On OGBN-Arxiv, boundary saturation is compatible with non-collapsed performance at $L=16$, while the failed $L=32$ checkpoints share an endpoint outcome but occupy different validation-selected diagnostic states. Cora and Texas provide replicated Euclidean components, and Euclidean GCN checks bound backbone dependence. These results motivate a joint diagnostic readout: no single probe, endpoint score, or onset layer is treated as a universal failure rule.

**Contributions.**

- **Operational protocol.** We specify a reproducible fixed-support workflow with explicit inputs, geometry-aware measurements, stability checks, diagnostic outputs, and a limited model-selection use case.

- **Bounded mechanism study.** On the patent graph, the aligned $L=32$ audit supports a seed- and subset-stable separation between a Euclidean late-contraction signature and elevated hidden-to-hidden drift plateaus for the boundary-saturated hyperbolic checkpoints. Endpoint and dispersion evidence support the oversmoothing-like Euclidean interpretation. The audit remains limited to $L=32$ width-preserving transitions and does not make the full sweep or its onset intrinsic.

- **Cross-dataset diagnostic evidence.** A complete three-seed OGBN-Arxiv replay demonstrates both saturation without failure at $L=16$ and state–outcome dissociation under the reproducible $L=32$ training-and-selection failure; Cora, Texas, and GCN checks provide additional component and backbone coverage.

The method is diagnostic rather than remedial. It can identify a stable candidate range and flag when a deeper model enters an unstable or indeterminate regime, but it does not by itself prescribe a new width, convolution, or regularizer.

## 2 Related Work

**Depth pathologies in deep GNNs.** A large literature studies why message-passing GNNs degrade with depth, most prominently under the notion of oversmoothing, where repeated propagation drives node representations toward low-frequency or weakly distinguishable states (Li et al., 2018; Oono & Suzuki, 2020; Rusch et al., 2023; Wu et al., 2023). Prior analyses characterize this effect through spectral viewpoints, contraction or mixing behavior of graph operators, and asymptotic limits of message passing. A related but

distinct line of work studies oversquashing and long-range information bottlenecks in graph propagation (Alon & Yahav, 2021; Topping et al., 2021). In parallel, a broad range of architectural and training modifications—including residual or identity mappings, normalization schemes, multiscale aggregation, diffusion-style propagation, stochastic edge dropping, and very-deep/reversible constructions—have been proposed to stabilize deeper GNNs (Chen et al., 2020b; Xu et al., 2018; Gasteiger et al., 2019; Rong et al., 2020; Zhao & Akoglu, 2020; Li et al., 2021).

These studies provide important theoretical and empirical insights, but they are commonly assessed through endpoint performance, asymptotic behavior, or a single diagnostic viewed in isolation. Our focus is complementary: we study how internal representations evolve across intermediate layers after training.

**Geometric representations and hyperbolic GNNs.** To better capture hierarchical and non-Euclidean structure, graph representation learning has increasingly explored curved latent spaces, especially hyperbolic geometry (Nickel & Kiela, 2017; Ganea et al., 2018; Chami et al., 2019; Bachmann et al., 2020; Yang et al., 2022). Hyperbolic embeddings can represent tree-like or hierarchical relations efficiently, and hyperbolic GNNs extend message passing to such spaces through manifold-aware operations, tangent-space approximations, or curvature-aware transformations. Recent work has also emphasized that conclusions in hyperbolic graph learning can be sensitive to baseline parity and evaluation protocol choices (Katsman & Gilbert, 2025), and that geometry-task alignment may matter separately from graph hyperbolicity alone (Naddeo et al., 2026).

Much of this literature evaluates geometric models primarily through downstream performance, distortion, or final-layer embedding quality. Comparisons between Euclidean and hyperbolic models are therefore often reported at the level of endpoint metrics, leaving open the question of whether their internal representation dynamics differ qualitatively with depth.

**Post-hoc representation diagnostics.** A separate line of work analyzes learned graph representations using post-hoc diagnostics such as interlayer drift, smoothness or Dirichlet-style quantities, class separability, and auxiliary probing models (Chen et al., 2020a; Zhou et al., 2021; Rusch et al., 2023; Guan & Shi, 2025). These signals are often informative, but they are usually applied one at a time and under paper-specific sampling choices, node subsets, or distance conventions. Recent work has also questioned whether Dirichlet-like measures alone reliably track oversmoothing-related performance loss across realistic depth ranges, motivating alternative rank-based views in some settings (Zhang et al., 2026). As a result, the resulting measurements are often difficult to compare directly across depths, models, or embedding spaces.

Our work does not introduce new primitives for each of these views. Instead, it coordinates familiar diagnostics under a fixed probe subset, shared evaluation edges, and geometry-consistent metrics so that they become jointly reproducible and jointly interpretable.

**Position of this work.** Our work is closest to post-hoc depth analysis. It does not introduce a new GNN layer or a new scalar oversmoothing index; it coordinates familiar diagnostics on fixed supports, records how their conclusions depend on geometry and observation maps, and returns an auditable trajectory-level diagnosis. The patent and public-benchmark experiments test what this workflow reveals beyond endpoint performance.

## 3 Background and problem setting

### 3.1 A representation-centric view of depth

Oversmoothing is commonly described as a depth-induced phenomenon in which node representations become increasingly indistinguishable. Many studies diagnose this through downstream accuracy loss, implicitly using performance decay as a proxy for representational failure. However, accuracy degradation can be delayed or even absent depending on the task and supervision, whereas representational collapse is a property of the learned depth dynamics themselves (Li et al., 2018; Oono & Suzuki, 2020; Wu et al., 2023). We therefore adopt a representation-centric view: after training, a depth-$L$ message-passing network induces a discrete

sequence of layerwise representations, and depth corresponds to the composition of the learned update maps. In general, these maps are layer-specific, so the resulting dynamics are non-autonomous.

Oversmoothing is distinct from *oversquashing*, in which information from distant nodes is compressed through graph bottlenecks (Alon & Yahav, 2021; Topping et al., 2021). Our focus is on depthwise representation dynamics and on geometry-aware diagnostics for oversmoothing-like degradation and boundary-associated metric amplification.

### 3.2 Why a joint protocol is needed

Graph smoothness measures, such as (metric-aware) Dirichlet energy, quantify neighborhood-level variation and have been widely used to diagnose or mitigate oversmoothing (Chen et al., 2020a; Zhou et al., 2021; Rusch et al., 2023), with recent refinements beyond first-order Dirichlet energy (Guan & Shi, 2025). Drift-based and separability-based views are likewise informative, but no single signal is uniquely diagnostic. The same low-energy or low-drift pattern can reflect either useful convergence or degenerate homogenization, and in non-Euclidean settings it can coexist with geometry-driven effects that are invisible to a single scalar.

These observations motivate a joint protocol that explicitly tracks the co-evolution of interlayer drift, graph-local roughness, class structure, and, in hyperbolic space, boundary pressure across depth. Fixing probe nodes and evaluation edges is essential here: without fixed subsets and geometry-consistent metrics, depthwise trends are confounded by resampling variation or by incomparable distance conventions across models and embedding spaces.

### 3.3 Hyperbolic geometry and boundary amplification

Hyperbolic GNNs embed representations in a Poincaré ball, where distances can grow rapidly near the boundary. This geometry is well-suited to hierarchical structure (Nickel & Kiela, 2017; Chami et al., 2019; Bachmann et al., 2020; Yang et al., 2022), but it changes how depth diagnostics should be interpreted. In particular, low coordinate drift or low local variation need not imply harmful collapse: hyperbolic representations can remain stable while preserving angular, class-relevant structure. At larger depths, however, representations may saturate toward the boundary ($\|x\| \to 1$), where small directional updates can induce large geodesic displacements, leading to boundary-associated metric amplification (Ganea et al., 2018). Concretely, for a sufficiently small coordinate update $\Delta x$ in the Poincaré ball,

$$d_{\text{Hyp}}(x, x + \Delta x) \approx \lambda(x)\|\Delta x\|_2, \qquad \lambda(x) = \frac{2}{1 - \|x\|_2^2},$$

so identical coordinate-scale updates become more visible in measured geodesic drift near the boundary, and squared-distance quantities such as metric-aware Dirichlet energy are locally amplified by approximately $\lambda(x)^2$. This is why explicit boundary monitoring is incorporated into our probing framework.

## 4 Method: Fixed-point (plateau) probing on fixed subsets

Throughout this section, *fixed-point-like* denotes a near-stationary plateau in an *aligned* observed trajectory, as formalized in Appendix D.1. Drift measures layer-to-layer displacement, Dirichlet energy measures graph-local variation, separability summarizes class structure, and boundary statistics track hyperbolic metric sensitivity. The workflow reads these quantities jointly and reports uncertainty when the qualitative signature is not stable across available seeds, supports, or observation maps.

Let $G = (V, \mathcal{E})$ be a graph with node features $\{x_i\}_{i \in V}$. A trained message-passing GNN of depth $L$ produces layerwise *internal activations* $\{\tilde{\mathbf{h}}_i^{(\ell)}\}_{i \in V, \ell=0}^{L}$, where $\tilde{\mathbf{h}}_i^{(0)}$ is the input projection of $x_i$. We write the depth-indexed layer update as

$$\tilde{H}^{(\ell+1)} = F_{\theta^{(\ell)}}\left(\tilde{H}^{(\ell)}; G\right), \quad \ell = 0, \ldots, L-1, \tag{1}$$

where $\tilde{H}^{(\ell)} = \{\tilde{\mathbf{h}}_i^{(\ell)}\}_{i \in V}$ and $F_{\theta^{(\ell)}}$ denotes the trained message-passing layer at depth $\ell$. For notational simplicity, we occasionally write $F_\theta$ when layer-specific parameters are not essential.

**Representation geometry.** We consider two representation manifolds: (i) Euclidean space $\mathbb{R}^d$ with $d_{\mathrm{Euc}}(u, v) = \|u - v\|_2$; and (ii) a $d$-dimensional Poincaré ball with curvature parameter $c > 0$,

$$\mathbb{B}_c^d = \{u \in \mathbb{R}^d : c\,\|u\|_2^2 < 1\},$$

equipped with the geodesic distance $d_{\mathbb{B}_c}(u, v)$ (Appendix D.4). Throughout this paper, we write $d_{\mathrm{Hyp}}(u, v) := d_{\mathbb{B}_c}(u, v)$. In the main patent GraphSAGE configuration, HYPA uses $c = 1.0$ and HYPT uses $c = 3.0$ (see Table A.3). For readability, some expressions are written in unit-ball form ($c = 1$); all probe computations use the model-specific curvature, and the full $c$-dependent formulas are provided in Appendix D.4. All probes below are defined for a generic metric $d(\cdot, \cdot)$.

**Observed representations for probing.** Intermediate activations may be parameterized in different coordinate systems, depending on the model. We therefore distinguish between the *internal activations* $\tilde{\mathbf{h}}_i^{(\ell)}$ and the *observed representations* $\mathbf{z}_i^{(\ell)}$ used by the probes. For Euclidean models, we set $\mathbf{z}_i^{(\ell)} := \tilde{\mathbf{h}}_i^{(\ell)} \in \mathbb{R}^d$. For hyperbolic models, the internal activations are mapped to the Poincaré ball via the exponential map at the origin:

$$\mathbf{z}_i^{(\ell)} := \exp_0\!\left(\tilde{\mathbf{h}}_i^{(\ell)}\right) \in \mathbb{B}_c^d, \tag{2}$$

where $\exp_0(\cdot)$ is defined in Appendix D.4. All subsequent probes are evaluated on the observed representations $\{\mathbf{z}_i^{(\ell)}\}$ under the metric induced by the target geometry.

**Common probe dimension and measurement role.** The input, hidden, and output widths are 384, 128, and 64 in the main patent configuration. Full-sweep summaries map every layer to a 64-dimensional operational probe space: width-64 layers use the identity; wider layers use fixed, untrained orthogonal projectors determined by the evaluation seed and layer index and reused across runs.

Because consecutive hidden layers can use different projectors, common-space drift mixes representation change with observation-map change. Energy uses one projector within each layer and avoids this additive cross-layer term, but energy and class-dispersion curves remain subspace-dependent. We therefore treat full-sweep and width-changing values as *operational observation-space summaries*, not intrinsic trajectories or fixed-point tests.

Appendix E.7 audits the $L{=}32$ EUCLID/HYPA/HYPT hidden-to-hidden trajectories without layer-indexed dimension changes: the primary replay uses each model's native 128-dimensional hidden parameterization while retaining its geometry-specific observation map and metric, and the sensitivity replay uses one shared $128{\to}64$ projector across every hidden layer. Euclidean late/middle drift and energy ratios are well below one, whereas the two hyperbolic drift ratios remain near one after early boundary saturation. The audit does not cover shallower depths, separability, or width-changing interfaces and therefore does not replace the operational full-sweep convention.

**Fixed probe sets.** All diagnostics use fixed supports: a probe-node set $\mathcal{S} \subset V$ for drift, separability, and boundary monitoring, and an evaluation-edge set $\mathcal{E}_{\mathrm{eval}} \subset \mathcal{E}$ for Dirichlet energy. For the patent summaries, $\mathcal{S}_{\mathrm{test}}$ is fixed within the temporal test mask and $\mathcal{S}_{\mathrm{val}}$ is fixed within the validation mask; each is reused across depths, geometries, training seeds, and replays. The symmetrized energy support $\mathcal{E}_{\mathrm{eval}}^{\mathrm{sym}}$ is sampled once from the full induced graph and reused throughout.

The node and edge supports therefore answer different questions: node-level summaries are split restricted, whereas the reported energy is a fixed induced-graph statistic rather than a test-only statistic. This distinction is recorded in captions and manifests and is revisited as a limitation in Section 7. Exact node IDs, edge IDs, seeds, and hashes are specified in Appendix A.1; public datasets state their own conventions in Appendix C.

**Practical budget choice.** The main experiments use $|\mathcal{S}| = 50{,}000$ nodes and $|\mathcal{E}_{\mathrm{eval}}| = 200{,}000$ edges. Smaller pilot budgets preserve the tested regime ordering (Appendix E.3). A practitioner can increase each budget until qualitative labels and candidate rankings stabilize across several pilot seeds, then freeze and report the supports.

---

**Protocol 1** Operational fixed-support trajectory diagnosis.

---

**Inputs.** Trained checkpoints for candidate depths/geometries/backbones; a fixed node support $\mathcal{S}$; a fixed edge support $\mathcal{E}_{\text{eval}}$; geometry and observation-map metadata; training seeds and, when available, support/projector resamples.

1. **Freeze the evaluation support.** Reuse the same $\mathcal{S}$ and $\mathcal{E}_{\text{eval}}$ for every candidate in the comparison.

2. **Construct observed representations.** Apply the geometry map and record whether each transition is native/shared-map aligned, layer-indexed common-space, or width changing.

3. **Measure the trajectory.** Compute drift, metric-aware energy, class-dispersion/separability components, and hyperbolic radius/deficit statistics; retain ambient-coordinate steps for hyperbolic checks.

4. **Check stability.** Compare qualitative ordering and onset intervals across training seeds and available support/observation-map audits. Report a confident label only when its defining direction is unchanged across all available seeds and no tested support/map check reverses the ordering; otherwise return *indeterminate.*

5. **Return an output.** Report the evidence vector, a stable candidate-depth interval, an onset interval when supported, and one of the following descriptive labels: stable plateau; contraction/oversmoothing-like; boundary-amplified; or indeterminate.

**Joint interpretation.** A stable plateau requires low or decreasing *aligned* hidden-to-hidden drift together with stable energy and class structure. A contraction/oversmoothing-like label requires contraction in local/dispersion statistics together with loss or fragility of class structure; unaligned drift cannot establish the label by itself. A boundary-amplified label requires persistent boundary occupancy plus elevated geodesic drift/energy relative to earlier layers and the ambient-coordinate check. A single signal is insufficient. No absolute threshold is transferred across geometries or datasets; relative changes, change intervals, and seed/map consistency are the operative evidence.

**Practitioner use.** The output can compare candidate models and identify the deepest candidate before an unstable regime when additional depth provides no endpoint benefit. For example, in the OGBN-Arxiv candidate sweep, $L{=}16$ remains non-collapsed whereas $L{=}32$ has collapsed endpoints and different validation-selected diagnostic states; among those candidates, the workflow retains $L \leq 16$ as the candidate range and flags $L{=}32$. It does not, without separate intervention evidence, prescribe a wider model, a different convolution, or a particular regularizer.

---

## 4.1 Drift: Inter-layer displacement

For node $i$ and transition $\ell \to \ell + 1$, we define the per-node drift as

$$\delta_i^{(\ell)} = d\left(\mathbf{z}_i^{(\ell+1)}, \mathbf{z}_i^{(\ell)}\right), \qquad \ell = 0, \dots, L - 1. \tag{3}$$

We summarize the drift over the probe set $\mathcal{S}$ using

$$D_{\text{mean}}(\ell) = \frac{1}{|\mathcal{S}|} \sum_{i \in \mathcal{S}} \delta_i^{(\ell)}, \tag{4}$$

$$D_{\text{p50}}(\ell) = \text{median}_{i \in \mathcal{S}} \, \delta_i^{(\ell)}. \tag{5}$$

**Reporting convention.** Unless otherwise stated, we report drift using the median summary $D_{\text{p50}}(\ell)$, denoted in figures/tables as "drift (p50)." When a single depth-level drift value is required (e.g., depth-sweep plots), we use the final transition

$$D_{\text{final}}(L) := D_{\text{p50}}(\ell = L - 1). \tag{6}$$

For upper-tail diagnostics reported in the appendix, we analogously use

$$D_{\text{p90}}(\ell) := \text{p90}_{i \in \mathcal{S}} \, \delta_i^{(\ell)}. \tag{7}$$

Across multiple random seeds, reported drift values are summarized as mean±std or explicitly identified per-seed summaries in the corresponding caption.

For hyperbolic models, $d$ in Eq. (3) is the Poincaré geodesic distance. Under a native or shared observation map, persistently small hidden-to-hidden drift can support a near-stationary plateau label (Appendix D.1). Under the layer-indexed common-space transform or across a width-changing interface, the same quantity is an operational transition statistic: it may contain observation-map change and is not, by itself, evidence of a fixed point. In every case drift is interpreted jointly with energy, class structure, boundary statistics, and the available alignment audit.

**Metric scale and cross-geometry comparisons.** Because $d_{\mathrm{Euc}}$ and $d_{\mathrm{Hyp}}$ live on different scales, and $d_{\mathrm{Hyp}}$ can be strongly amplified near the boundary, we do *not* compare absolute drift magnitudes across geometries. Our cross-geometry claims are therefore qualitative rather than quantitative: we compare which regimes arise and how probe couplings (e.g., drift–energy–boundary interactions) evolve within each geometry across layers and trained depths. Accordingly, hyperbolic drift is interpreted jointly with boundary proxies (Section 4.4) and coordinate-step diagnostics (Appendix D.5).

**Why boundary proximity can amplify measured dynamics.** In the Poincaré ball, $g_x = \lambda(x)^2 I$ with $\lambda(x) = 2/(1 - \|x\|_2^2)$. For a sufficiently small ambient-coordinate update $\Delta x$, $d_{\mathrm{Hyp}}(x, x + \Delta x) \approx \lambda(x)\|\Delta x\|_2$, while squared-distance energy is locally approximately $\lambda(x)^2$-sensitive. Hyperbolic drift and energy must therefore be read with boundary and ambient-coordinate statistics.

Together with Dirichlet energy and label separability, interlayer drift serves as a complementary diagnostic signal: drift captures representation dynamics across layers, energy characterizes graph-local variation, and separability reflects task-relevant class structure.

## 4.2 Dirichlet energy: Graph-local roughness

To quantify local variation of node representations over the graph, we measure a metric-aware Dirichlet energy on a fixed evaluation edge set. For directed citation graphs, this means the symmetrized set $\mathcal{E}_{\mathrm{eval}}^{\mathrm{sym}}$ defined in Eq. (9); for undirected graphs, $\mathcal{E}_{\mathrm{eval}}^{\mathrm{sym}} = \mathcal{E}_{\mathrm{eval}}$.

$$E(\ell) = \frac{1}{|\mathcal{E}_{\mathrm{eval}}^{\mathrm{sym}}|} \sum_{(u,v) \in \mathcal{E}_{\mathrm{eval}}^{\mathrm{sym}}} d\left(\mathbf{z}_u^{(\ell)}, \mathbf{z}_v^{(\ell)}\right)^2. \tag{8}$$

Here $d(\cdot, \cdot)$ denotes the distance induced by the representation geometry (e.g., Euclidean or hyperbolic). Intuitively, Dirichlet energy measures how rapidly neighboring node representations vary across edges and therefore provides a geometry-aware notion of graph-local roughness.

For directed citation graphs, we define

$$\mathcal{E}_{\mathrm{eval}}^{\mathrm{sym}} := \{(u, v) \mid (u, v) \in \mathcal{E}_{\mathrm{eval}} \text{ or } (v, u) \in \mathcal{E}_{\mathrm{eval}}\}. \tag{9}$$

This symmetrization ensures that the symmetric squared-distance form used by the energy probe is well-defined and comparable across conditions. It is used only for the energy probe; message passing itself remains directed as described in Section 5.

The evaluation edge subset is sampled once (excluding self-loops) and then reused across depths, geometries, and activation variants in order to eliminate resampling variance in the diagnostic measurements. Under this definition, a decreasing $E(\ell)$ indicates increasing similarity among neighboring node representations, whereas an increase in $E(\ell)$ indicates growing neighborhood-level variation (or representational roughness). For upper-tail summaries reported in the appendix, we also use

$$E_{\mathrm{p90}}(\ell) := \mathrm{p90}_{(u,v) \in \mathcal{E}_{\mathrm{eval}}^{\mathrm{sym}}} d\left(\mathbf{z}_u^{(\ell)}, \mathbf{z}_v^{(\ell)}\right)^2. \tag{10}$$

By fixing the evaluation edges, changes in $E(\ell)$ reflect representation dynamics rather than variance introduced by edge resampling.

### 4.3 Separability: Class structure at the output layer

Drift and energy do not directly measure whether representations remain discriminative. We therefore track label separability as a ratio of between-class to within-class dispersion, where larger values indicate stronger class structure.

Let $\mathcal{C}$ denote the label set, and let $\mathcal{V}_c = \{i \in V : y_i = c\}$ denote the node set for class $c$. Because separability is evaluated on the fixed probe set $\mathcal{S}$, we define the represented class set

$$\mathcal{C}_\mathcal{S} := \{c \in \mathcal{C} : |\mathcal{V}_c \cap \mathcal{S}| > 0\}.$$

Using only the probe nodes $\mathcal{V}_c \cap \mathcal{S}$ for $c \in \mathcal{C}_\mathcal{S}$, we define the class prototypes

$$\mu_c^{(\ell)} = \begin{cases} \dfrac{1}{|\mathcal{V}_c \cap \mathcal{S}|} \displaystyle\sum_{i \in \mathcal{V}_c \cap \mathcal{S}} \mathbf{z}_i^{(\ell)}, & \text{(Euclidean)}, \\[2ex] \exp_0\left( \dfrac{1}{|\mathcal{V}_c \cap \mathcal{S}|} \displaystyle\sum_{i \in \mathcal{V}_c \cap \mathcal{S}} \log_0\left(\mathbf{z}_i^{(\ell)}\right) \right), & \text{(hyperbolic)}, \end{cases} \tag{11}$$

where $\log_0$ and $\exp_0$ denote the Riemannian logarithmic and exponential maps at the origin of the Poincaré ball, respectively (Appendix D.4).

**Implementation note (hyperbolic prototypes).** The hyperbolic prototype in Eq. (11) (and the prototype mean in Eq. (14)) is computed as the tangent-space average at the origin, followed by $\exp_0$. This is a numerically stable and scalable approximation to the Fréchet mean for large probe sets; see Appendix D.2.

We define the within-class dispersion as

$$\mathrm{Disp}_w(\ell) = \sum_{c \in \mathcal{C}_\mathcal{S}} \frac{1}{|\mathcal{V}_c \cap \mathcal{S}|} \sum_{i \in \mathcal{V}_c \cap \mathcal{S}} d\left(\mathbf{z}_i^{(\ell)}, \mu_c^{(\ell)}\right)^2, \tag{12}$$

and the between-class dispersion as

$$\mathrm{Disp}_b(\ell) = \frac{1}{|\mathcal{C}_\mathcal{S}|} \sum_{c \in \mathcal{C}_\mathcal{S}} d\left(\mu_c^{(\ell)}, \bar{\mu}^{(\ell)}\right)^2, \tag{13}$$

where $\bar{\mu}^{(\ell)}$ denotes the mean of the class prototypes:

$$\bar{\mu}^{(\ell)} = \begin{cases} \dfrac{1}{|\mathcal{C}_\mathcal{S}|} \displaystyle\sum_{c \in \mathcal{C}_\mathcal{S}} \mu_c^{(\ell)}, & \text{(Euclidean)}, \\[2ex] \exp_0\left( \dfrac{1}{|\mathcal{C}_\mathcal{S}|} \displaystyle\sum_{c \in \mathcal{C}_\mathcal{S}} \log_0\left(\mu_c^{(\ell)}\right) \right), & \text{(hyperbolic)}. \end{cases} \tag{14}$$

Separability is defined as the global ratio

$$\mathrm{Sep}(\ell) = \frac{\mathrm{Disp}_b(\ell)}{\mathrm{Disp}_w(\ell) + \varepsilon}, \tag{15}$$

where $\varepsilon > 0$ is a fixed stabilizing constant (Table A.3). Separability is a single scalar computed on the probe set; it is not aggregated over nodes by a median or mean. Unless otherwise stated, reported separability values refer to the final layer $\ell = L$ on the test split.

**Interpretation at extreme depth.** $\mathrm{Sep}(\ell)$ is a ratio statistic. In extreme-depth regimes, both $\mathrm{Disp}_b(\ell)$ and $\mathrm{Disp}_w(\ell)$ may shrink, and the ratio in Eq. (15) can increase if the within-class term collapses faster (or because of the stabilizer $\varepsilon$). We therefore treat separability as a *secondary* class-structure summary and interpret it jointly with complementary evidence: drift, energy, boundary statistics when applicable, and the explicitly reported numerator and denominator $\mathrm{Disp}_b(\ell)$ and $\mathrm{Disp}_w(\ell)$ (Figure B.2 and section 4.3).

**Remark.** $\text{Disp}_w(\ell)$ is defined as the sum of per-class dispersions, so each represented class contributes equally regardless of its size. Averaging $\text{Disp}_w(\ell)$ over $|\mathcal{C}_\mathcal{S}|$ would only rescale $\text{Sep}(\ell)$ by a constant factor (up to the stabilizer $\varepsilon$) and does not affect the qualitative depthwise trends. We do not treat separability as a stand-alone diagnostic, but rather as a projection of class structure whose interpretation requires complementary probes and direct inspection of its between- and within-class dispersion components.

### 4.4 Boundary monitoring in hyperbolic models

To monitor boundary effects in hyperbolic models, we define the *curvature-normalized radius*

$$r_i^{(\ell)} := \sqrt{c}\,\|\mathbf{z}_i^{(\ell)}\|_2 \in [0,1),$$

which reduces to the Euclidean ball radius in the unit-curvature case $c = 1$. We then track the mean normalized radius

$$\bar{r}(\ell) = \frac{1}{|\mathcal{S}|} \sum_{i \in \mathcal{S}} r_i^{(\ell)}, \tag{16}$$

which measures average boundary proximity (larger $\bar{r}(\ell)$ means closer to the boundary), and the boundary occupancy ratio

$$B(\ell) = \frac{1}{|\mathcal{S}|} \sum_{i \in \mathcal{S}} \mathbb{I}\left[r_i^{(\ell)} > \tau_r\right], \tag{17}$$

where $\tau_r$ is a fixed threshold close to the boundary (we used $\tau_r = 0.95$; Table A.3).

These quantities are monotone proxies for boundary proximity and should not be confused with geodesic distance from the origin. Accordingly, drift and Dirichlet energy near the boundary should be interpreted jointly with $\bar{r}(\ell)$ and $B(\ell)$ (Appendix D.5).

In some plots (e.g., Figure 3), we also report the upper-tail boundary statistic

$$r_{\text{p90}}(\ell) = \text{p90}_{i \in \mathcal{S}}\, r_i^{(\ell)},$$

which serves as a sensitive indicator of boundary saturation. Appendix-only median boundary summaries use

$$r_{\text{p50}}(\ell) := \text{p50}_{i \in \mathcal{S}}\, r_i^{(\ell)}.$$

When helpful, we additionally report the corresponding boundary deficit

$$\Delta_i^{(\ell)} := 1 - r_i^{(\ell)},$$

whose values near zero indicate near-boundary saturation. The implementation projects hyperbolic observations to $r_i^{(\ell)} \le 1 - \varepsilon_{\text{proj}}$. A reported quantile equal to this cap is therefore a right-censored saturation indicator: it establishes that the corresponding portion of the probe distribution reached the numerical boundary, but it does not measure severity beyond the cap. In the capped regime, the absolute geodesic-drift plateau is strongly shaped by curvature and $\varepsilon_{\text{proj}}$; we use drift to read the approach to saturation, plateau onset, and departures from it, but do not compare saturated plateau magnitudes across curvatures or datasets. In the appendix intervention study on OGBN-Arxiv, $r_{\text{p90}}^{\text{late}}$ denotes the mean of $r_{\text{p90}}(\ell)$ over the designated late-layer window for the corresponding experiment.

### 4.5 Reproducibility and reporting choices

Fixed-point probing requires explicit choices that can affect the diagnostics. We fix and report the following: (i) how $\mathcal{S}$ and $\mathcal{E}_{\text{eval}}$ are sampled (including random seeds); (ii) directed-edge handling for message passing and for $\mathcal{E}_{\text{eval}}$; (iii) metrics and constants for $d_{\text{Hyp}}$ (curvature); and (iv) numerical stabilizers such as $\varepsilon$ and thresholds such as $\tau_r$.

For visualization, we plotted the probe signals on logarithmic scales (and used log1p transforms, where indicated in figure captions) to accommodate a large dynamic range. All analyses used raw values.

# 5 Experimental setup

All probe nodes and evaluation edges are fixed across runs to eliminate resampling variance in diagnostic measurements. All diagnostics follow a fixed measurement protocol with released evaluation artifacts, enabling exact reproduction of the reported depthwise dynamics. Unless otherwise stated, endpoint metrics are reported as mean±std over three training seeds; fixed probe subsets and fixed evaluation edges remove resampling variance from the diagnostics, and the released per-seed CSVs show consistent qualitative depth trends. Main-text captions explicitly state when a plot shows single-checkpoint layerwise traces rather than across-seed aggregates and point to the seed-aggregated counterparts in Appendix E.2 where applicable.

## 5.1 Dataset: Large-scale patent citation graph

We constructed a directed patent citation graph from patent grants issued by the United States Patent and Trademark Office (USPTO). Nodes correspond to patents and directed edges represent citations (citing $\rightarrow$ cited).

**Edge direction and temporal leakage.** Raw citations are stored as directed edges (citing $\rightarrow$ cited). For message passing, we reverse the propagation direction (cited $\rightarrow$ citing) so that information flows from older patents to newer ones. This choice follows the knowledge-flow interpretation of citation graphs and, under our year-based temporal split, prevents training/validation nodes from receiving messages from future test nodes at any hop. Unless explicitly stated, all reported GNN results use this directed propagation graph; the original citation direction is used only for dataset semantics, and the Dirichlet-energy probe uses a separate symmetrized evaluation edge set (Section 4.2).

Each node was assigned a single CPC subclass label derived from the Cooperative Patent Classification (CPC) system, which we used as a representative technological category for the patent. Each patent was also associated with its textual abstract.

The textual abstracts were embedded using a text encoder and used as node features. For reproducibility, we document and release the embedding recipe (text model/version, embedding dimensionality, and preprocessing) together with the GNN code and CSV artifacts.

To reflect realistic deployment settings in which future patents must be inferred from past structures, we used a temporal split by grant year to ensure that training, validation, and test sets are disjoint in time.

The induced subgraph used in our experiments contains 479,533 patents and 1,864,213 citation edges. Node features are 384-dimensional abstract embeddings, and the label space consists of 316 CPC subclasses (see Appendix A.4 for full details).

Citation edges pointing from newer patents to older patents are preserved in the dataset but cannot introduce temporal leakage because message passing is performed in the cited $\rightarrow$ citing direction.

**Why this dataset is the main stress test.** We use the patent citation graph as the primary mechanism-revealing setting because it combines the properties most relevant to our question: large scale, directed citation structure under a temporal split, and a many-class label space with substantial hierarchy and imbalance. These conditions make depth-induced representational regime changes particularly likely to surface and make hyperbolic geometry a plausible representation choice. We therefore treat the patent graph as the main stress test for whether single-signal diagnostics miss boundary-coupled regimes, while the public benchmarks in Appendix C serve as transfer/reference checks rather than as a search for dataset-universal quantitative thresholds.

**Embedding details.** Node features were obtained from patent abstracts using the publicly available `intfloat/e5-small-v2` transformer encoder, which outputs $d_{\text{in}}$=384-dimensional vectors. The released embedding script tokenizes each abstract directly with the encoder tokenizer, truncates to 256 tokens, mean-pools the non-masked token embeddings, and L2-normalizes the resulting vector. All embeddings were computed once prior to GNN training and reused across depths, geometries, and activation variants (no embedding fine-tuning). The review supplement includes the embedding script and public encoder identifier;

the public archive will additionally record the exact upstream model revision and complete environment metadata.

**Reproducibility and release plan.** The anonymized review supplement contains the figure-linked artifacts, public-benchmark summaries, canonical analysis scripts, manifests, and compact diagnostics needed to audit the reported results, including the embedding script and public encoder identifier. Appendix A.5 is the authoritative specification of the five-package archive and the figure-to-artifact mapping; the larger patent-subgraph bundle and exact upstream encoder-revision/environment metadata will be disclosed with the public camera-ready archive.

## 5.2 Models: Euclidean vs. hyperbolic GraphSAGE

We used GraphSAGE (Hamilton et al., 2017) as the common message-passing backbone. The Euclidean variant embeds nodes in $\mathbb{R}^d$. The hyperbolic variant embeds in the Poincaré ball $\mathbb{B}_c^d$ and uses standard hyperbolic operations (tangent-space aggregation and Riemannian maps), similar to those in prior hyperbolic neural network implementations (Ganea et al., 2018; Chami et al., 2019). Both variants share the same neighborhood sampling and aggregation structure; the key difference is the representation geometry and metrics used by the probes.

**Hyperbolic baselines and tuning parity.** To avoid conflating the hyperbolic geometry effects with underoptimization, we report two hyperbolic variants throughout the paper. This emphasis on tuning parity is deliberate: recent audit work shows that conclusions in hyperbolic graph learning can be highly sensitive to Euclidean baseline strength and evaluation protocol design (Katsman & Gilbert, 2025). HYPA is a representative off-the-shelf hyperbolic instantiation under the shared training protocol. HYPT is a minimally tuned hyperbolic control (tangent-space classifier head, input scaling $\alpha_{\text{in}}$, curvature $c$, and a learning-rate split between the encoder and the head; Table A.3) introduced to ensure competitive shallow-depth performance under a comparable tuning budget to the Euclidean baseline. In particular, HYPT matches the Euclidean baseline at shallow depth (e.g., $L{=}2$; Table 3 and Appendix B.1) while preserving the same qualitative late-depth signature. Thus, the hyperbolic regime reported here is not explained by trivial shallow-depth underoptimization or by a uniformly weak hyperbolic baseline. HYPT isolates curvature-aware aggregation without explicit radial control (i.e., without the radius regularization used in Section 6.3).

To address concerns about curvature choice, we additionally report a curvature robustness sweep for HYPT over $c \in \{1, 2, 3, 5\}$ in Appendix B.7, showing that varying $c$ changes the absolute probe scales but does not remove the qualitative very-deep failure regime.

We sweep the depth $L \in \{2, 4, 8, 16, 32\}$ unless otherwise stated (with additional plots and tables in the Appendix). Unless otherwise stated, ReLU activation was used.

## 5.3 Training protocol and pointwise nonlinearity ablation

The models were trained using minibatch neighbor sampling. To reduce undertraining confounds in deep models, we allocated a larger epoch budget for $L \geq 16$ (800 vs. 200 epochs for $L \leq 8$; Table A.3), while keeping the optimizer and other hyperparameters fixed across depths. To isolate the role of *pointwise* nonlinearities in depth-induced dynamics, we compared standard ReLU networks to their ablated counterparts obtained by replacing ReLU with an identity map at the same depth (ReLU→identity). We emphasize that this ablation removes only pointwise activation, whereas other non-Euclidean operations (e.g., Riemannian maps in hyperbolic models) remain nonlinear. Task performance is reported using *test macro-F1* as a reference; however, our analysis focused on layerwise probe signals. Unless otherwise stated, each run is evaluated at the checkpoint selected by the validation metric used in the training pipeline, and the reported test macro-F1 is taken at that validation-selected checkpoint. Because the task involved many classes (316 CPC subclasses) with a strong imbalance under a temporal split, macro-F1 values can be small in absolute terms; therefore, we treated them as supplementary endpoint references rather than as optimization targets.

### 5.4 Diagnostics computation

All probes were computed post-training for each trained model. We fixed $\mathcal{S}$ (for drift, separability, and boundary monitoring) and $\mathcal{E}_{\mathrm{eval}}^{\mathrm{sym}}$ (for Dirichlet energy) and reused them across depths, geometries, and activation variants. Training uses stochastic minibatch neighbor sampling, but the post-training probes are computed on fixed $\mathcal{S}$ and fixed $\mathcal{E}_{\mathrm{eval}}^{\mathrm{sym}}$ with a fixed evaluation-time sampling seed (`seed_eval`=0), so the released diagnostics are deterministic given a trained checkpoint.

**Computational overhead.** Fixed-point probing is computed after training and requires only forward-pass representations of the fixed subsets. With a probe node set $\mathcal{S}$ and evaluation edges $\mathcal{E}_{\mathrm{eval}}^{\mathrm{sym}}$, costs scale linearly with depth: drift, separability, and boundary summaries are $O(L|\mathcal{S}|)$ distance evaluations (plus class-wise aggregation), whereas the Dirichlet energy is $O(L|\mathcal{E}_{\mathrm{eval}}^{\mathrm{sym}}|)$ distance evaluations. No backpropagation is required, and the probes can be computed in a streaming fashion over layers without storing full-graph embeddings, making the overhead comparable to a few evaluation-time forward passes on the same subsets.

## 6 Results

The patent results compare joint signatures under similarly poor deep endpoints. The aligned $L$=32 audit covers all three GraphSAGE modes and separates a late Euclidean contraction signature from boundary-saturated hyperbolic drift plateaus: across the reported training seeds and node subsets, the Euclidean and hyperbolic drift-ratio ranges are disjoint. The full-sweep and width-changing values remain operational summaries. Figures 1–3 provide the core patent evidence, Table 3 the depth sweep, Figure 5 a backbone check, and Figure 6 and table 4 the OGBN-Arxiv instantiation. Ambient-coordinate and secondary checks are in Appendices D.5 and B.

### 6.1 Layerwise dynamics at fixed depth

To characterize the layerwise trajectory at a fixed trained depth, we examine drift, metric-aware Dirichlet energy, and boundary proximity for the seed-0 $L$=16 checkpoints. Figure 1 does *not* show a common monotone early-stabilization phase. In the operational common space, Euclidean drift and energy rise through intermediate transitions, decline later, and rebound at the terminal end. HYPA instead undergoes a rapid early increase in both quantities while its normalized radius approaches the boundary, after which drift and energy remain on an elevated plateau.

Figure 1 is descriptive because it uses the layer-indexed common space at $L$=16. The separate aligned $L$=32 audit yields a stable tested separation: every Euclidean full-split and node-subset drift ratio is at most 0.330, whereas every hyperbolic ratio is at least 0.972. It does not retroactively align the $L$=16 panel or establish an invariant onset. The panel remains useful for showing the operational nonmonotonic shape and the early formation of hyperbolic saturation.

The figure is read within geometry; Appendix D.5 and Figure A.5 give the ambient-step and radius checks.

### 6.2 Illustrative geometry view and aligned $L = 32$ audit

Figure 2 is illustrative, not an onset estimate. The complete-test audit in Table 2 shows two distinct aligned $L$=32 signatures. Across every tested full-split seed and node subset, Euclidean drift ratios lie in 0.011–0.330, while the boundary-saturated HYPA/HYPT ratios lie in 0.972–1.001; the tested ranges do not overlap. All three Euclidean fixed-edge energy ratios are below $4 \times 10^{-6}$. Hyperbolic fixed-edge energy is not load-bearing for the middle-to-late plateau contrast: for HYPT, its direction varies across seeds (Section E.7), so that contrast rests on the aligned drift ratios and boundary saturation. The claim is the aligned regime contrast, not the selected-subset onset.

Table 1: Qualitative patent-graph signatures. The $L{=}16$ rows are operational common-space summaries; the aligned audit applies to all three $L{=}32$ GraphSAGE modes. Absolute Euclidean and Poincaré magnitudes are not compared.

| Observed setting | Drift $D$ | Energy $E$ | Separability Sep | $\bar{r}$ (hyperbolic only) |
|---|---|---|---|---|
| Euclidean, $L{=}16$ | operational; nonmonotonic | nonmonotonic | nonmonotonic | – |
| HYPA, $L{=}16$ | rapid early rise → elevated plateau | rapid early rise → elevated plateau | secondary; nonmonotonic | rapid rise → 1 |
| Euclidean, $L{=}32$ late-depth regime | aligned late contraction | aligned late contraction | ratio fragile / nonmonotonic | – |
| Hyperbolic, $L{=}32$ late-depth regime | aligned elevated drift plateau | elevated; moderate/noisy change | $\approx$ / $\downarrow$ | near 1 |

**Within-geometry trajectory shapes only; Euclidean and Poincaré magnitudes are not commensurate.**

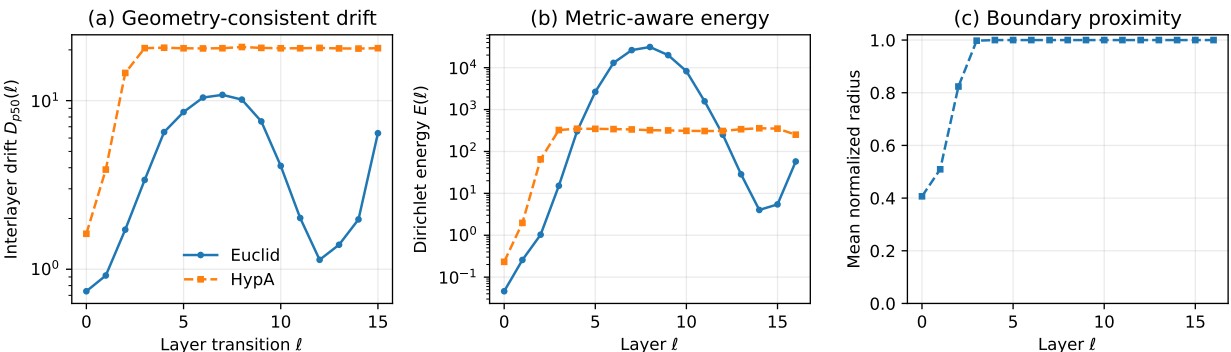

Figure 1: **Layerwise trajectories at $L = 16$ on the patent graph.** Seed-0 operational summaries on fixed supports. Euclidean drift and energy are nonmonotonic; HYPA rises early and plateaus as radius saturates. Appendix E.7 separately audits the $L = 32$ EUCLID/HYPA/HYPT hidden-to-hidden trajectories; it does not make this $L = 16$ panel intrinsic.

## 6.3 Boundary-targeted perturbation probe at fixed depth

Association does not establish mechanism, so we use a radius-targeted training perturbation as a bounded diagnostic probe. It tests whether changing the timing and strength of boundary pressure changes the observed trajectory; it is not proposed as a performance-preserving regularizer.

Concretely, we augment the supervised objective with a hinge-squared penalty on the (normalized) Poincaré ball radius at the final layer:

$$\mathcal{L} = \mathcal{L}_{\text{task}} + \lambda \, \mathcal{L}_{\text{rad}}, \qquad \mathcal{L}_{\text{rad}} = \frac{1}{|\mathcal{B}|} \sum_{i \in \mathcal{B}} \left[ \max\left( 0, \; \sqrt{c} \, \|\mathbf{z}_i^{(L)}\|_2 - \rho_0 \right) \right]^2, \tag{18}$$

where $\mathcal{B}$ denotes the minibatch seed nodes used for supervision, $c$ is the hyperbolic curvature, and $\rho_0 \in (0, 1)$ is the target radius cap, and $\lambda \geq 0$ controls the intervention strength. We used $\rho_0{=}0.90$ for the controlled runs shown in Figure 3. The *late-only* setting applied the same penalty only during the final stage of training (the last 25% of the epochs in our implementation), testing whether the delayed boundary control is sufficient.

All other hyperbolic results use $\lambda = 0$. At $L = 32$, weak late-only control remains saturated and changes little; stronger always-on control reduces the high-energy part of the pre-terminal trajectory, while drift responds nonmonotonically (Appendix E.5). The upper-tail radius is at the projection cap in the primary window for every condition, so panel (a) is right-censored and does *not* establish a bulk-radius reduction. The evidence is therefore limited to a trajectory response under a radius-targeted perturbation. Its energy/drift asymmetry is compatible with first-order geodesic versus squared-distance sensitivity, but it is not a clean control law.

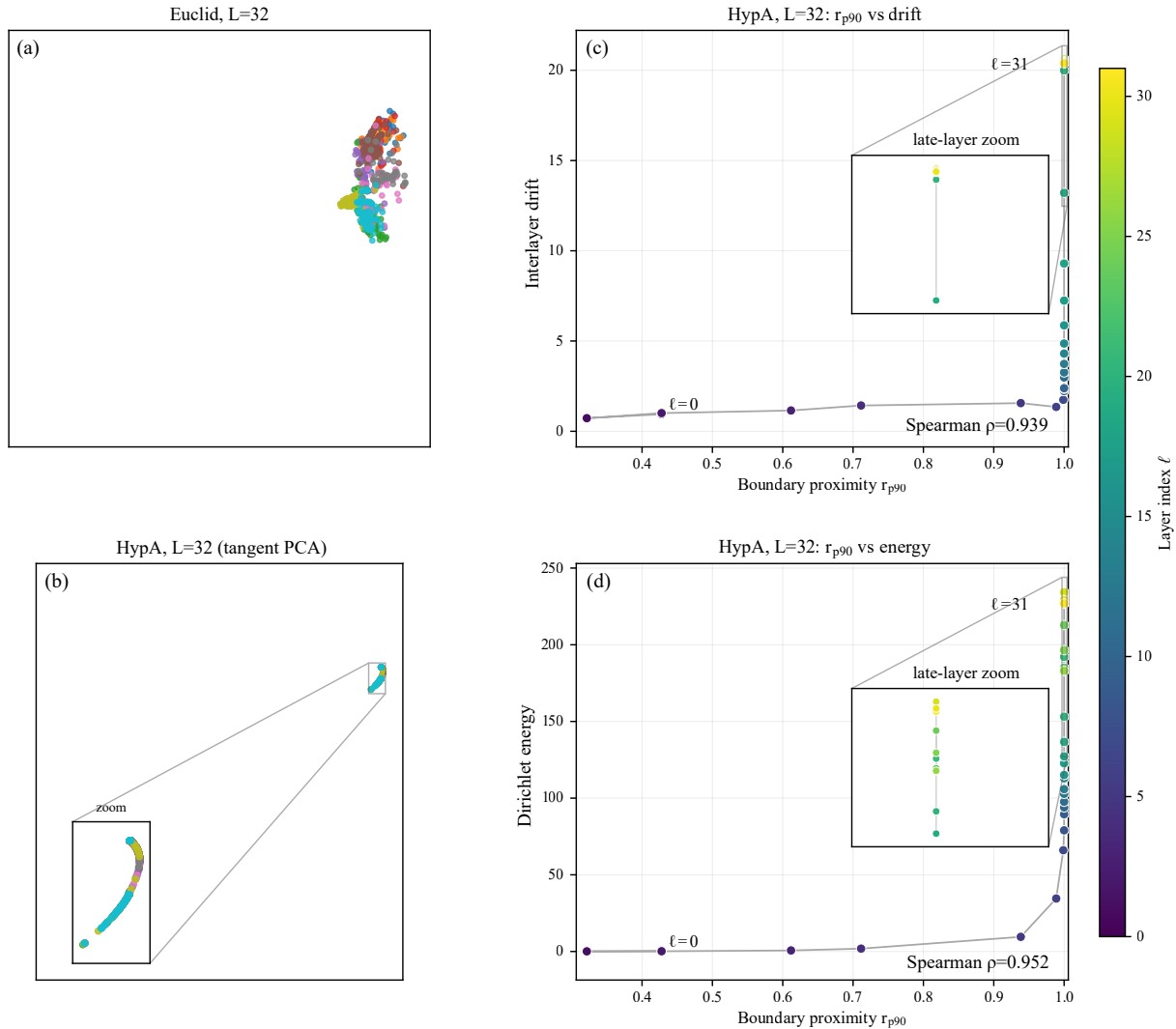

Figure 2: **Illustrative selected-subset boundary coupling at** $L = 32$**.** Panels (a,b) visualize Euclidean and HYPA states using Euclidean and tangent-space PCA. Panels (c,d) show a seed-0 selected-subset phase view in which upper-tail radius co-varies with geodesic drift and energy. PCA is visualization-only, and this panel is not a complete-test aggregate or an onset estimate; Appendix E.7 provides the aligned three-seed audit.

## 6.4 Class structure as a secondary check

Final-layer separability is a compact secondary summary, not a stand-alone failure score. Figure 4 reports the global Eq. (15) statistic on the complete fixed test probe over three training seeds. Its within-geometry depth trend is nonmonotonic, so it is read with drift, energy, boundary statistics, and endpoint performance.

The Euclidean $L = 32$ numerator and denominator expand, contract, and rebound together over several orders of magnitude; therefore a terminal ratio increase need not mean recovered class structure. Appendix B.6 and Figure B.2 provide the full decomposition. The aligned $L = 32$ drift/energy contraction and the endpoint/dispersion evidence jointly support the interpretation; no single probe is used alone.

Table 2: Aligned hidden-to-hidden audit for the patent $L = 32$ Euclid/HypA/HypT checkpoints. The primary replay removes layer-indexed projection by using the native 128-dimensional hidden parameterization at every hidden layer while retaining the model-specific observation map and metric. Drift and saturation statistics use the complete fixed test-node support, whereas energy uses the same fixed symmetrized 200k-edge support sampled from the full induced graph. Ratios are late-window divided by middle-window medians and are summarized as mean±std over three training seeds. For drift, values near one indicate an elevated plateau; values far below one in both drift and fixed-edge energy indicate late contraction. Hyperbolic energy is shown for completeness; its HypT middle-to-late direction varies across seeds, so it is not load-bearing for the plateau contrast. Full shared-projector, subset, and per-seed energy checks are in Appendix E.7.

| mode | drift late/mid | energy late/mid | first $r_{p90} \geq 0.95$ layer |
|---|---|---|---|
| Euclid | $0.085 \pm 0.116$ | $(1.48 \pm 2.11) \times 10^{-6}$ | – |
| HypA | $0.977 \pm 0.005$ | $0.870 \pm 0.128$ | $3.3 \pm 1.5$ |
| HypT | $0.992 \pm 0.008$ | $0.973 \pm 0.273$ | $1.7 \pm 0.6$ |

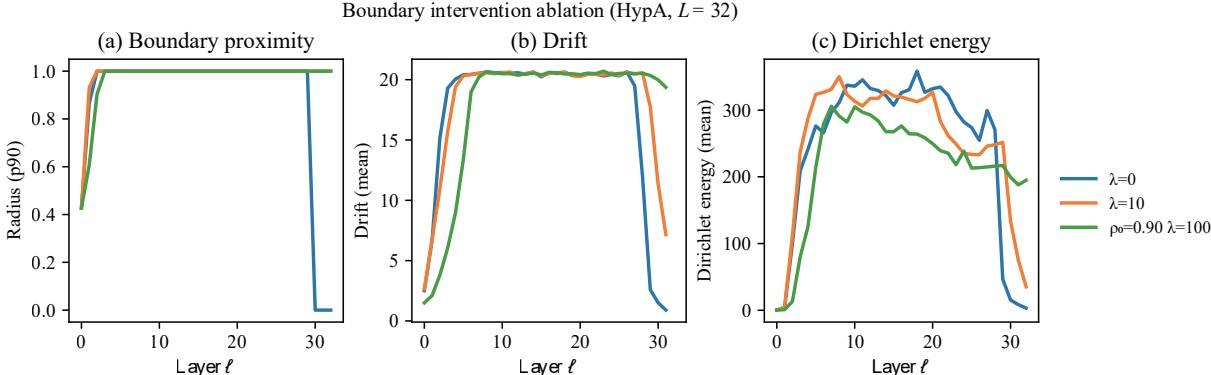

Figure 3: **Radius-targeted perturbation at $L = 32$ (HypA).** The curves compare no control, late-only $\lambda = 10$, and always-on $\lambda = 100$. Upper-tail radius is right-censored at saturation; always-on control mainly lowers the pre-terminal high-energy portion, while drift is nonmonotonic. Appendix E.5 reports the numeric window summary.

## 6.5 Activation replay and grouped diagnostics

The post-hoc ReLU→Identity replay is retained as a secondary sensitivity check, not as a separately trained architecture comparison. Removing the pointwise nonlinearity destabilizes the Euclidean replay but does not remove the elevated HypA drift regime; the full traces are in Appendix B.8. Degree- and class-resolved summaries likewise show broad, geometry-dependent heterogeneity rather than a single degree bin or class driving the result (Figures A.1–A.4).

## 6.6 Patent depth-sweep summary

The patent evidence is therefore read as a joint signature. Euclidean operational summaries show intermediate expansion and later decline. In the aligned $L=32$ audit, the three full-split Euclidean drift and fixed-edge energy ratios are below one, and the Euclidean drift-ratio range over the reported seeds and node subsets is disjoint from the hyperbolic range; endpoint and dispersion evidence support the oversmoothing-like interpretation. The evaluated hyperbolic trajectories become boundary saturated and retain elevated aligned hidden-to-hidden drift plateaus. The radius-targeted perturbation changes energy more clearly than drift but does not establish a smooth control law. Table 3 places the operational full-sweep summaries beside endpoint performance; Appendix B.9 reports the validation/test final-transition stability check.

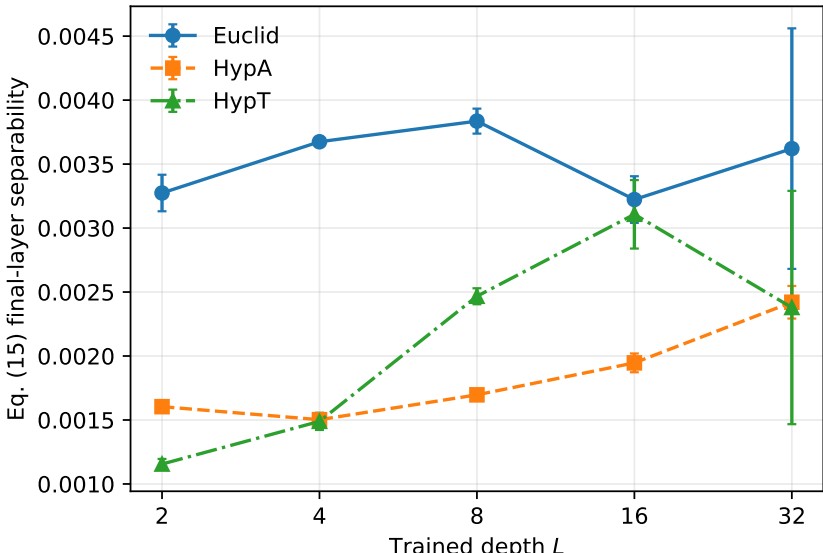

Figure 4: **Final-layer Eq.** (15) **separability versus depth.** Points show mean±std over three training seeds on the fixed test probe. The ratio is used only as a secondary within-geometry class-structure summary.

At $L = 32$, all three patent model families reach collapse-level macro-F1. The joint signatures nevertheless differ: the Euclidean aligned hidden-to-hidden drift and fixed-edge energy show a late-contraction signature, and the operational terminal transition is quiet; the hyperbolic runs remain boundary saturated with near-constant late/middle drift. The aligned drift-ratio ranges remain disjoint across the tested seeds and node subsets. HYPT matches the Euclidean shallow-depth endpoint at $L = 2$, reducing the likelihood that the hyperbolic pattern is explained only by shallow underoptimization. The probes are not accuracy surrogates; they distinguish signatures that the common endpoint collapse does not.

### 6.7 Backbone check on the same patent subgraph (Euclidean GCN)

We repeated the patent experiment with a Euclidean GCN on the same fixed supports. This is a diagnostic backbone check rather than an optimized benchmark because neighbor-sampled GCN normalization is approximate. Endpoint performance again collapses with depth, but the $L = 16$ internal profile peaks in mid-depth layers. The workflow therefore remains informative beyond GraphSAGE while the location of degeneration remains backbone dependent.

**Interpreting Euclid at $L$=32: low energy and a quiet terminal update.** Table 3 uses the final transition defined in Eq. (6). At $L = 32$, the Euclidean model has low final-layer energy and a smaller final-transition drift than at several shallower depths. Figure B.2 shows that the internal class-dispersion trajectory is nevertheless highly nonmonotonic: both the between- and within-class components expand through intermediate layers, contract sharply late, and rebound at the terminal layer. The Eq. (15) ratio can therefore increase without implying recovered class structure. Appendix E.7 independently shows sharp middle-to-late contraction in the width-preserving native hidden trajectory; the terminal width-changing update remains an operational endpoint statistic. We interpret the endpoint as a low-roughness, comparatively weak-update state and avoid inferring a monotone homogenization path from the ratio alone.

Table 3: Patent depth-sweep summaries (mean±std over three training seeds). Each geometry is shown in a separate panel because Euclidean and Poincaré drift/energy magnitudes are not commensurate. Drift is the operational final transition in the common probe space; energy is the last-layer mean on the fixed induced-graph edge set; separability is the final-layer Eq. (15) scalar.

**(a) Euclidean**

| $L$ | test macro-F1 | final drift $D_{p50}$ | final energy | Eq. (15) Sep |
|---|---|---|---|---|
| 2 | $0.309 \pm 0.004$ | $9.090 \pm 0.047$ | $50.101 \pm 3.124$ | $0.003 \pm 0.000$ |
| 4 | $0.239 \pm 0.006$ | $12.257 \pm 0.416$ | $101.860 \pm 6.597$ | $0.004 \pm 0.000$ |
| 8 | $0.120 \pm 0.040$ | $9.877 \pm 1.350$ | $72.797 \pm 22.624$ | $0.004 \pm 0.000$ |
| 16 | $0.013 \pm 0.003$ | $5.761 \pm 0.634$ | $52.783 \pm 8.023$ | $0.003 \pm 0.000$ |
| 32 | $6.097e-04 \pm 1.066e-07$ | $4.016 \pm 1.123$ | $23.812 \pm 7.751$ | $0.004 \pm 0.001$ |

**(b) HypA**

| $L$ | test macro-F1 | final drift $D_{p50}$ | final energy | Eq. (15) Sep |
|---|---|---|---|---|
| 2 | $0.104 \pm 0.002$ | $6.953 \pm 0.240$ | $174.474 \pm 6.464$ | $0.002 \pm 0.000$ |
| 4 | $0.094 \pm 0.022$ | $8.793 \pm 2.690$ | $192.415 \pm 7.259$ | $0.002 \pm 0.000$ |
| 8 | $0.085 \pm 0.007$ | $20.452 \pm 0.053$ | $233.566 \pm 11.214$ | $0.002 \pm 0.000$ |
| 16 | $0.015 \pm 0.001$ | $20.474 \pm 0.036$ | $261.480 \pm 12.924$ | $0.002 \pm 0.000$ |
| 32 | $6.098e-04 \pm 0.000e+00$ | $20.536 \pm 0.090$ | $245.196 \pm 32.467$ | $0.002 \pm 0.000$ |

**(c) HypT**

| $L$ | test macro-F1 | final drift $D_{p50}$ | final energy | Eq. (15) Sep |
|---|---|---|---|---|
| 2 | $0.297 \pm 0.002$ | $6.952 \pm 0.090$ | $115.919 \pm 1.129$ | $0.001 \pm 0.000$ |
| 4 | $0.257 \pm 0.006$ | $8.743 \pm 0.077$ | $118.439 \pm 0.216$ | $0.001 \pm 0.000$ |
| 8 | $0.150 \pm 0.009$ | $11.842 \pm 0.001$ | $115.587 \pm 0.171$ | $0.002 \pm 0.000$ |
| 16 | $0.031 \pm 0.005$ | $11.831 \pm 0.050$ | $122.766 \pm 3.593$ | $0.003 \pm 0.000$ |
| 32 | $6.098e-04 \pm 0.000e+00$ | $10.189 \pm 2.724$ | $69.396 \pm 57.506$ | $0.002 \pm 0.001$ |

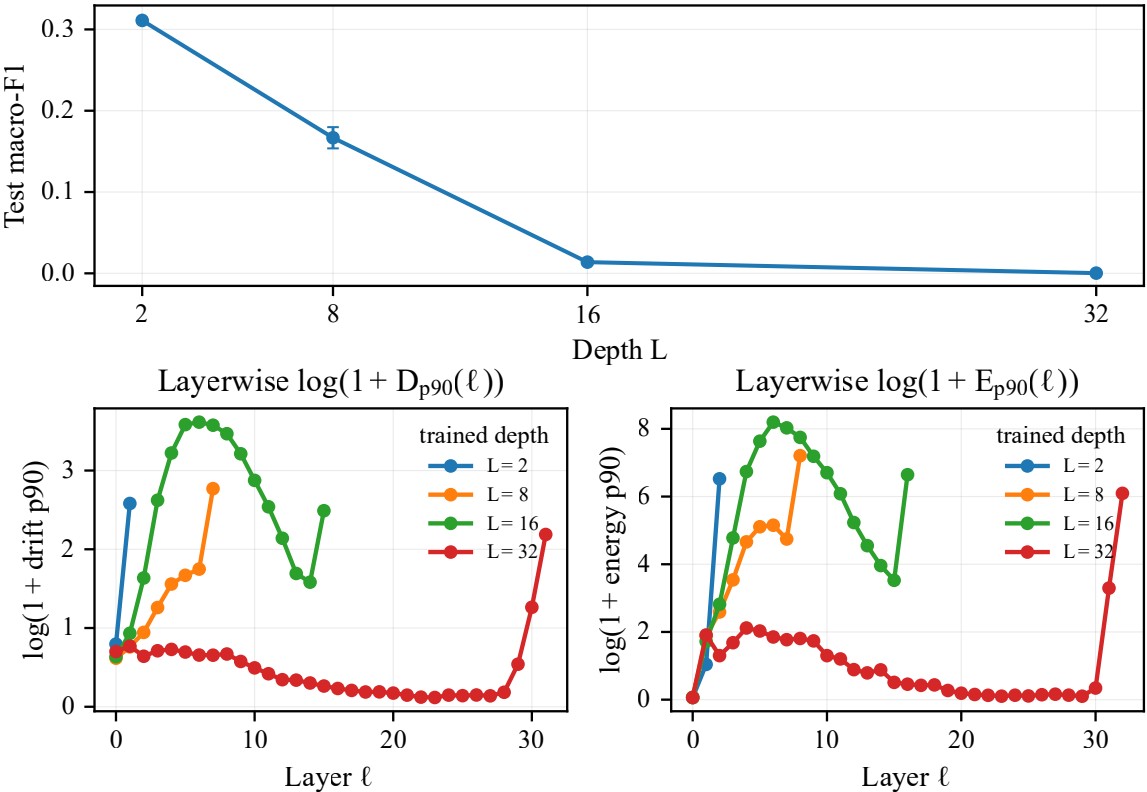

Figure 5: **Euclidean GCN check on the patent subgraph.** Endpoint macro-F1 declines with depth, while upper-tail drift and energy show an architecture-specific mid-depth spike at $L = 16$. Curves use the same fixed supports and three training seeds.

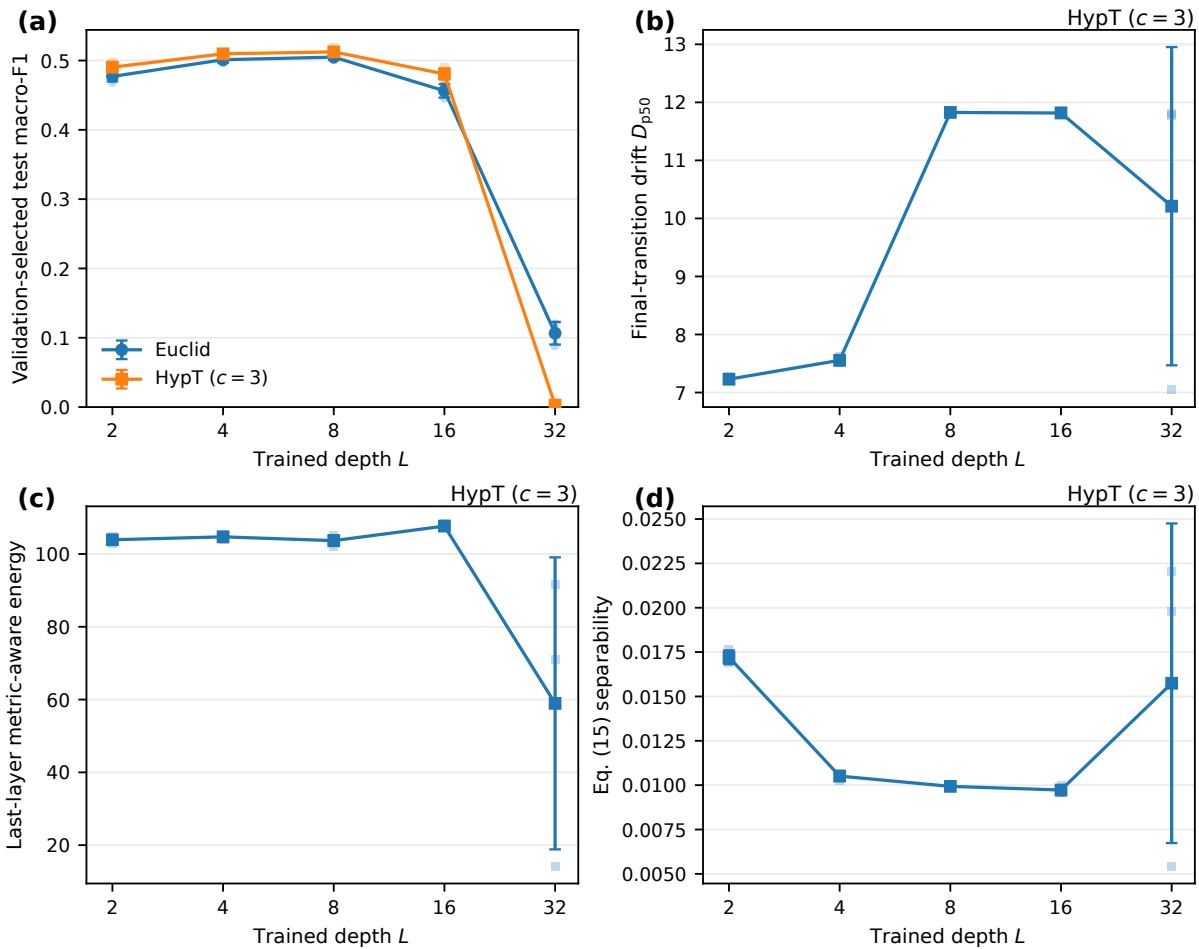

Figure 6: **OGBN-Arxiv state–outcome dissociation over three training seeds.** Panel (a) shows validation-selected test macro-F1. Panels (b–d) show HYPT final-transition drift, last-layer energy, and Eq. (15). Boundary saturation coexists with useful $L = 16$ performance, whereas $L = 32$ collapses reproducibly without a common terminal probe value.

## 6.8 OGBN-Arxiv: saturation without failure and state–outcome dissociation at $L = 32$

We replayed the complete probe family on Euclidean and OGBN-specific HYPT GraphSAGE checkpoints at $L \in \{2, 4, 8, 16, 32\}$ over training seeds 0, 1, and 2. The node support is the complete fixed test split (48,603 nodes), energy uses one fixed symmetrized 200k-edge-derived set, and endpoints are the stored test values of validation-selected checkpoints. Table A.5 records the dataset-specific training configuration.

**Saturation without failure at $L = 16$.** HYPT retains test macro-F1 $0.481 \pm 0.008$ at $L = 16$ (Euclidean: $0.456 \pm 0.010$), even though final $r_{p90}$ and $r_{p50}$ reach the projection cap in all three seeds and saturation begins in the early hidden layers. This is the clearest public-benchmark reason not to treat boundary proximity as a failure criterion.

**$L = 32$: reproducible pipeline failure with different selected-checkpoint states.** All three HYPT runs reach the same collapse-level test macro-F1, but final drift, energy, separability, saturation onset, and terminal radius distributions differ across seeds (Figure C.13). Because validation selects very early checkpoints at epochs 7, 3, and 2 of 200, we interpret this row as a reproducible failure of the stated training-and-selection pipeline, not as multiple converged realizations of one deep hyperbolic mechanism. The narrower reproducible

Table 4: Complete OGBN-Arxiv GraphSAGE readout (mean±std over seeds 0–2). Drift is the operational final transition; energy and Eq. (15) separability are last-layer values. "Best ep." is the range of validation-selected epochs out of 200, and the final column gives the first $r_{p90}/r_{p50}$ saturation layer for HYPT. The dagger marks the very-early selected $L=32$ checkpoints.

| Mode | $L$ | test macro-F1 | drift $p50$ | energy | Eq. (15) Sep | best ep. | first sat. $p90/p50$ |
|---|---|---|---|---|---|---|---|
| EUCLID | 2 | $0.477 \pm 0.007$ | $6.982 \pm 0.342$ | $15.50 \pm 1.35$ | $0.01577 \pm 0.00098$ | 96–133 | – |
| EUCLID | 4 | $0.501 \pm 0.004$ | $9.662 \pm 0.342$ | $23.56 \pm 2.04$ | $0.01745 \pm 0.00026$ | 178–188 | – |
| EUCLID | 8 | $0.505 \pm 0.003$ | $12.451 \pm 0.551$ | $26.09 \pm 0.86$ | $0.01945 \pm 0.00046$ | 165–200 | – |
| EUCLID | 16 | $0.456 \pm 0.010$ | $10.716 \pm 0.242$ | $15.18 \pm 0.83$ | $0.02601 \pm 0.00195$ | 98–141 | – |
| EUCLID | 32 | $0.107 \pm 0.016$ | $10.358 \pm 1.149$ | $6.48 \pm 0.91$ | $0.01974 \pm 0.00196$ | 194–198 | – |
| HYPT | 2 | $0.490 \pm 0.007$ | $7.229 \pm 0.035$ | $103.91 \pm 0.94$ | $0.01721 \pm 0.00039$ | 121–198 | 2/2 |
| HYPT | 4 | $0.510 \pm 0.003$ | $7.555 \pm 0.063$ | $104.71 \pm 0.67$ | $0.01051 \pm 0.00019$ | 116–125 | 4/4 |
| HYPT | 8 | $0.512 \pm 0.006$ | $11.826 \pm 0.021$ | $103.67 \pm 1.46$ | $0.00993 \pm 0.00007$ | 127–195 | 4–5/5–6 |
| HYPT | 16 | $0.481 \pm 0.008$ | $11.817 \pm 0.032$ | $107.66 \pm 0.63$ | $0.00972 \pm 0.00022$ | 182–197 | 4/5–6 |
| HYPT$^\dagger$ | 32 | $0.00277 \pm 0.00000$ | $10.211 \pm 2.742$ | $58.95 \pm 40.12$ | $0.01574 \pm 0.00901$ | 2–7 | 2–9/3–14 |

observation is that the shared endpoint scalar is insufficient to identify the selected-checkpoint diagnostic state, which motivates the joint, support-controlled readout.

Once a probe quantile reaches the projection cap, saturated geodesic magnitudes are strongly affected by curvature and $\varepsilon_{\mathrm{proj}}$. We use drift to read approach, persistence, and departures within a run, not to compare saturated plateau levels across datasets or curvatures.

**Portability coverage at a glance.** The complete current probe family is instantiated on two large graphs (the patent citation graph and OGBN-Arxiv), replicated Euclidean components are reported on Cora and Texas, and GCN checks provide patent and OGBN-Arxiv backbone references. Appendix E.1 maps the exact probe and replication coverage. These results characterize portability under the reported settings; they do not establish universal thresholds, onset depths, or mechanisms.

## 7 Discussion and limitations

**What the protocol adds.** The contribution is a structured output rather than a new scalar: an evidence vector, a candidate-depth stability interval, an onset interval when reproducible, and an uncertainty/indeterminate flag. On the patent graph, the joint evidence distinguishes a contraction-dominated Euclidean signature from a boundary-saturated hyperbolic signature. The $L = 32$ hidden-to-hidden contrast is aligned across all three GraphSAGE modes, while the broader depth sweep remains operational rather than intrinsic. On OGBN-Arxiv, it shows the converse: boundary saturation can occur without endpoint failure, and one endpoint failure can hide different validation-selected diagnostic states.

**Scope and measurement limitations.** The detailed boundary-associated mechanism is a patent stress-test finding. OGBN-Arxiv establishes a complete second instantiation but not a shared threshold or mechanism; Cora, Texas, and the GCN checks provide component and backbone coverage. The full-sweep 64-dimensional common space uses layer-indexed projectors: drift mixes representation change with map change, while energy avoids that additive cross-layer term but remains subspace-dependent. The aligned audit covers only the $L = 32$ EUCLID/HYPA/HYPT width-preserving hidden transitions; no matched shallower-depth or GCN replay was run. It supports the tested separation between a Euclidean late-contraction signature and elevated hyperbolic drift plateaus, but it does not make the full sweep intrinsic or establish invariant onset layers. Width-changing final transitions remain endpoint summaries rather than fixed-point tests. Node diagnostics use split-restricted supports, whereas energy uses a fixed edge sample from the full induced graph; a matched split-restricted edge audit is a useful future check.

**Mechanistic interpretation.** The patent radius-targeted perturbation is timing-sensitive and right-censored in upper-tail radius. It changes the high-energy portion of the trajectory more clearly than drift

but does not demonstrate a smooth radius-control law, a bulk-radius reduction in the displayed window, or preserved predictive performance. Ambient-coordinate/geodesic comparisons are consistent with metric amplification near the boundary, but remain observation-coordinate diagnostics rather than coordinate-invariant causal proof.

**Practical use and extensions.** A practitioner can run the workflow on a candidate sweep, prefer models whose probe trajectories remain stable, and avoid deeper candidates that enter an unstable regime without endpoint benefit. Choosing a wider model, another convolution, or a regularizer still requires separate intervention evidence. The current probe family is complementary rather than exhaustive; rank-based diagnostics are a natural additional component, especially where Dirichlet-style quantities are ambiguous. Further work should include attention/diffusion backbones, matched edge-support audits, broader aligned-space replays, and prospective use of the stability output for model selection or early stopping.

## 8 Conclusion

We introduced fixed-point probing, a fixed-support post-training workflow for auditing GNN representation trajectories. The patent study shows that similarly poor deep endpoints can accompany different joint diagnostic signatures, while the complete OGBN-Arxiv replay shows that boundary saturation can coexist with useful performance and that a reproducible endpoint collapse need not map to a unique validation-selected diagnostic state.

The workflow therefore does not turn drift, energy, separability, boundary proximity, or an endpoint score into a universal failure label. Its value is the joint, support-controlled diagnosis and the explicit reporting of measurement and seed instability.

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

# APPENDIX

## Supplementary and Reproducibility Details

**Reader's map.** Appendix A is the authoritative reproducibility and package specification. Appendix B contains the class-dispersion, activation-replay, split-stability, and additional depth/backbone panels moved from the main narrative. Appendix C reports public-benchmark details and intervention pilots. Appendix D gives geometric derivations and probe interpretation. Appendix E collects scope, uncertainty, budget, runtime, intervention, width, and observation-map audits.

## A  Experimental and Reproducibility Details

**How to read Appendix A.** Appendix A documents the experimental protocol and reproducibility details underlying all reported figures. Table A.1 serves as the central index that maps each figure to a minimal set of deterministic intermediate artifact(s) (primarily CSVs, plus coordinate/manifest exports where needed) and the exact script(s) required for regeneration. The following sections specify (i) how fixed probe/evaluation subsets are constructed and reused, (ii) training hyperparameters, (iii) dataset construction and statistics, and (iv) released diagnostics and their seed-dependence.

## Reproducibility Statement

The principal quantitative figures are generated from versioned intermediate artifacts produced by the analysis pipeline. Most figures are CSV-based; the geometry visualization additionally uses deterministic coordinate exports and a selected-node manifest. For the principal claim-bearing figures, Table A.1 lists the primary artifact file(s) and script(s); the release manifest provides the complete inventory. Unless otherwise stated, stochastic components are controlled by recorded seeds. Numerical CSVs are deterministic given the released checkpoints, software environment, graph bundle, and fixed subsets; rendered PDF/SVG output may vary at the metadata or backend level across library versions.

### A.1  Probe Node Set and Evaluation Edge Set

**Operational fixed-point notion.** A fixed-point-like label is used only for a near-stationary block under a native or shared observation map. Layer-indexed common-space and width-changing drifts remain reproducible transition summaries but are not sufficient fixed-point evidence. The label never implies convergence to a true attractor or a dynamical-systems theorem.

**Training-time induced subgraph and fixed subsets.** The training-time induced subgraph is constructed once and reused throughout. For the canonical patent node-level summaries, however, the probe nodes are *split specific*: the protocol records one fixed $\mathcal{S}_{\text{test}}$ and one fixed $\mathcal{S}_{\text{val}}$, and reuses the relevant set across depths, geometries, training seeds, and activation replays. Because each temporal split contains exactly 50,000 nodes and the canonical node budget is also 50,000, the canonical pass uses the complete test or validation split. Smaller-budget studies sample subsets within the designated split. The fixed symmetrized edge set $\mathcal{E}_{\text{eval}}^{\text{sym}}$ is defined once on the induced subgraph and reused for energy evaluation.

**Probe node sets $\mathcal{S}_{\text{test}}$ and $\mathcal{S}_{\text{val}}$.** The exact split-specific node IDs are stored in the protocol outputs and are the source of truth for the canonical node-level diagnostics. Drift, separability, and boundary summaries use the displayed split's fixed node set, with distances instantiated under the geometry-specific metric (Euclidean distance for EUCLID; Poincaré geodesic distance for HYPA/HYPT). Evaluation-time neighbor sampling and the fixed layer projections are controlled by the recorded evaluation seed; they do not change the selected node IDs.

Table A.1: Selected principal figure-to-artifact map. The table lists the numerical artifacts and regeneration scripts for the principal quantitative figures; the release manifest is the complete inventory.

| Figure | Primary numerical artifact(s) | Regeneration script |
|---|---|---|
| Fig. 1 | seed-0 test probe CSVs for Euclid/HypA at $L = 16$ and matching layerwise energy CSVs | `build_known_artifacts.py` |
| Fig. 2 | deterministic selected-node, coordinate, phase, and manifest exports | `plot_patent_geometry.py` |
| Table 2 and Appendix Table E.10 | $L = 32$ EUCLID/HYPA/HYPT native/shared-projector per-run and aggregate alignment CSVs | `summarize_observation_map_alignment.py` |
| Fig. 4 and Table 3 | canonical per-run CSVs; audited macro-F1, final-drift, and final-energy master | `recompute_patent_probe.py`; `finalize_patent_probe.py` |
| Fig. B.2 | canonical Euclidean $L = 32$ per-layer Eq. (12)/(13)/(15) values over three training seeds | `plot_patent_secondary.py` |
| Fig. B.4 | canonical seed-0 ReLU/identity replay CSVs at $L = 16$ | `plot_patent_secondary.py` |
| Fig. B.5 | released split-specific final-transition aggregates | `build_known_artifacts.py` |
| Figs. A.1–A.5 | canonical grouped drift CSVs with Poincaré drift in hyperbolic modes; HypA $L = 16$ energy CSV for Fig. A.5 | `plot_patent_grouped.py` |
| Fig. C.12 | canonical patent test/validation final transitions and public `public_fp_long.csv` | `plot_public_final_transition.py` |
| Fig. 5 | GCN depth, probe-layer, energy-layer, and final-layer summaries | `plot_patent_gcn_check.py` |
| Fig. 6 and Table 4 | OGBN-Arxiv current-definition per-run and mean/std CSVs over the complete 30-run grid | `run_ogbn_complete_probe.sh`; `make_ogbn_figures.py` |
| Fig. C.13 | OGBN-Arxiv three-seed layerwise normalized-radius audit and checkpoint metadata | `audit_ogbn_boundary_3seed.py`; `make_ogbn_figures.py` |
| Fig. C.1 | Single-seed GCN OGBN-Arxiv reference CSVs | `plot_ogbn_reference.py` |

**Evaluation edge set** $\mathcal{E}_{\text{eval}}$. We computed the metric-aware Dirichlet energy (Eq. (8)) on a fixed evaluation edge set $\mathcal{E}_{\text{eval}}$ restricted to the induced subgraph. Self-loops were excluded from the analysis. For directed graphs, we evaluated the symmetrized set $\mathcal{E}_{\text{eval}}^{\text{sym}}$ (Eq. (9)) by treating the edges as undirected. Unless otherwise stated, we used $|\mathcal{E}_{\text{eval}}| = 200{,}000$ sampled once with a fixed seed and reused across all depths/geometries/activations.

## A.2 Reproducibility Commands

This section describes the exact commands used to generate the released deterministic artifacts from the trained model checkpoints. Each command corresponds to a *deterministic analysis step* in the post-training pipeline and produces the CSV/JSON/manifest artifacts used by the principal quantitative figures and tables.

Notably, these commands operate exclusively on (i) fixed trained checkpoints, (ii) released induced-subgraph bundles (Appendix A.4), and (iii) fixed probe/evaluation subsets (Appendix A.1). No stochastic resampling was performed beyond those explicitly controlled by the listed random seeds. The commands in Table A.2 regenerate the numerical artifacts from the recorded inputs and seeds. We validate numerical tables and source CSVs, rather than claiming byte-identical PDF files across plotting-library and TeX backends.

Table A.2 lists the minimal set of commands required to produce each class of intermediate artifact. For readability, we show only the key arguments that affect reproducibility; full commands (including paths and environment setup) are available in the accompanying scripts. In the released `scripts_bundle`, the original versioned filenames are accompanied by clean canonical aliases and a rename map. Table A.2 lists the canonical aliases used in the paper.

## A.3 Hyperparameter and Training Configuration

Table A.3 summarizes the main patent GraphSAGE configuration, and Table A.5 records the distinct OGBN-Arxiv GraphSAGE configuration used for the complete public-benchmark replay. Within the patent study, models shared the same optimizer and base hyperparameters across depths and geometries, and the epoch budget was increased for $L \geq 16$ to reduce undertraining concerns. OGBN-Arxiv instead uses a uniform 200-epoch budget at every depth; absolute diagnostic scales and the $L = 32$ training behavior are therefore not compared directly across the two datasets.

Table A.2: **Principal post-hoc regeneration commands.** Paths are symbolic. The supplied batch runner discovers the recorded checkpoint tree and writes collision-free outputs. Hyperbolic drift is explicitly Poincaré; Eq. (15) is explicitly the global-prototype, squared-distance statistic.

| Script | Key command / contract | Output |
| --- | --- | --- |
| `recompute_patent_probe.py` | `-sub-dir SUBGRAPH -train-script TRAIN -ckpt CKPT -mode MODE -depth L -split test -num-probe 50000 -seed-eval 0 -act relu -out-csv OUT.csv`. For hyperbolic modes the script always evaluates Poincaré geodesic drift and records raw and $\sqrt{c}$-normalized radii separately. | Per-layer canonical CSV; optional class/degree grouped CSV |
| `run_patent_probe_all.py` | `-project-root PROJECT -output patent_probe_out -resume` | Canonical 45 test runs plus selected validation, identity, and grouped-drift runs |
| `finalize_patent_probe.py` | `-protocol-out patent_probe_out -known-master patent_probe_known_master.csv -template paper_template.tex -output-tex 8183.tex` | Eq. (15) aggregate, patent summary tables, canonical analysis master, and validation report |
| `plot_patent_secondary.py` | `-protocol-root 8183_final -out-root PLOT_OUT` | Appendix dispersion and activation-replay figures in PDF/SVG/PNG, with the dispersion audit CSV |
| `plot_patent_grouped.py` | `-protocol-root 8183_final -hypa-energy-csv ENERGY.csv -out-root PLOT_OUT` | Grouped patent appendix figures with explicit scale policies |
| `plot_public_final_transition.py` | `-protocol-root 8183_final -public-long-csv public_fp_long.csv -out-root PLOT_OUT` | Figure C.13 using transition $L-1 \to L$ only |
| `build_known_artifacts.py` | Audited released aggregate/per-seed CSVs; no checkpoint replay | Main layerwise figure, appendix split-stability panel, uncertainty panels, and radius alignment |
| `summarize_observation_map_alignment.py` | `-audit-root diagnostic_csv_core/observation_map_alignment -out-dir ALIGNMENT_SUMMARY` | Validates the expected $3 \times 3$ $L = 32$ mode/seed grid and regenerates Tables 2 and E.10 from the released audit CSVs |
| `plot_ogbn_reference.py` | OGBN-Arxiv GCN depth-sweep CSVs | Single-seed coordinate-level GCN reference panel in Appendix C |
| `run_ogbn_complete_probe.sh` | `PROJECT_ROOT OUTPUT_ROOT full`; validates the complete fixed test probe and fixed evaluation-edge hashes, then replays the current definitions on seeds 0/1/2, depths 2/4/8/16/32, and EUCLID/HYPT | 30 per-run probe/energy CSVs, run metadata, mean/std table, and validation report |
| `audit_ogbn_boundary_3seed.py` | `-poc-root OGBN_ROOT -out-dir BOUNDARY_AUDIT` | Per-run and layerwise saturation-onset, persistence, deficit, and checkpoint-selection audit |
| `make_ogbn_figures.py` | `-family-per-run PER_RUN.csv -boundary-per-run BOUNDARY.csv -boundary-layerwise LAYERWISE.csv -out-dir FIG_OUT` | Figure 6 and Appendix Figure C.13 in PDF/SVG/PNG |

## A.4 Dataset Construction and Data Card

**Source data and graph construction.** We constructed a directed citation graph from USPTO utility patents granted between 2010 and 2020. Nodes correspond to patents and directed edges represent citations (citing → cited) with respect to temporal ordering. For GNN message passing, we used the reverse propagation direction (cited → citing) such that neighborhood aggregation for a newer patent draws only from older cited patents. This prevents information flowing from future nodes back to past nodes under a temporal split. The node features are abstract embeddings, and the labels are CPC subclasses after frequency filtering. We used a temporal split by grant year to reflect realistic deployment settings: training (2010–2016), validation (2017–2018), and test (2019–2020). For context, the underlying raw citation graph prior to temporal induction contains approximately 3.1 M nodes and 11.4 M directed edges (counts may vary slightly with the crawl snapshot and filtering); in this paper, all experiments and released diagnostics operate on the fixed induced subgraph described as follows.

**Training-time induced subgraph.** All probe/evaluation artifacts are defined on the *fixed induced subgraph* used in our runs (constructed once prior to training and released as part of the reproducibility bundle). Canonical node probes are selected within the designated temporal split, whereas the fixed evaluation-edge set is sampled from the induced subgraph and then reused across conditions. The bundle includes nodes from all temporal splits (training/validation/test), with labels masked by the split; notably, message passing uses the cited→citing propagation direction, which blocks future-to-past information flow under the temporal split.

Table A.3: Main patent GraphSAGE hyperparameters and training configuration. Public-dataset and intervention settings are reported separately.

| Item | Value |
|------|-------|
| Optimizer | Adam |
| Learning rate | $3 \times 10^{-3}$ |
| Weight decay | $1 \times 10^{-4}$ |
| Hidden dimension $d$ | 128 |
| Batch size | 2048 |
| Neighbor fanout (training) | 15 |
| Neighbor fanout (evaluation) | 100 |
| Epochs | 200 ($L \leq 8$), 800 ($L \geq 16$) |
| Hyperbolic curvature $c$ | 1.0 (HYPA), 3.0 (HYPT) |
| Projection $\varepsilon_{\mathrm{proj}}$ | $1 \times 10^{-5}$ |
| Radius threshold $\tau_r$ | 0.95 |
| Radius-penalty cap $\rho_0$ (intervention only) | 0.90 |
| Radius-penalty weight $\lambda$ (intervention only) | 0 / 10 / 100 |
| Late-only schedule (intervention only) | last 25% of epochs |
| Stabilizer $\varepsilon$ | $1 \times 10^{-8}$ |

Therefore, probe construction is compatible with the temporal setting and does not introduce additional leakage beyond the standard transductive observation of features/edges.

**Released subgraph bundle (data card).** For reproducibility, we provide a self-contained *subgraph bundle*, which includes the following: (i) induced-subgraph node features and labels (`x.npy`, `y.npy`); (ii) induced-subgraph edge index (`sub_edge_index.npy`); (iii) temporal split masks (`train_mask.npy`, `val_mask.npy`, `test_mask.npy`); (iv) split-specific selected-node ID files for the canonical test and validation probes (with earlier subset metadata retained for provenance); (v) fixed symmetrized evaluation edges for Dirichlet energy (`eval_edges_200k_sym.npy`); (vi) an optional node-index back-map (`x.node_idx.npy`) for relating induced-subgraph rows to the original graph indexing; and (vii) a metadata file (`meta.json`) recording snapshot identifiers, preprocessing configurations, checksums, and induced-subgraph statistics used for the reported runs.

**Induced-subgraph statistics (exact).** In the released bundles used in the reported experiments, the induced subgraph contains $|V| = 479{,}533$ nodes and $|\mathcal{E}| = 1{,}864{,}213$ directed edges, with an input feature dimension $d_{\mathrm{in}} = 384$. We used canonical split-specific probe budgets $|\mathcal{S}_{\mathrm{test}}| = |\mathcal{S}_{\mathrm{val}}| = 50{,}000$ and a balanced temporal split with $|\mathcal{V}_{\mathrm{train}}| = |\mathcal{V}_{\mathrm{val}}| = |\mathcal{V}_{\mathrm{test}}| = 50{,}000$ nodes. Thus, each canonical test/validation probe is the complete corresponding split. Labels correspond to 316 CPC subclasses ($|C| = 316$). Class frequencies were highly imbalanced across the induced subgraph, and the minimum/median/maximum per-class counts were 60/497/63,220, respectively (median computed over subclasses with nonzero counts). Given the released bundle, these statistics are deterministic and support exact reproduction.

**Fixed evaluation edges.** For Dirichlet-energy evaluation, we sampled a fixed set of $|\mathcal{E}_{\mathrm{eval}}| = 200{,}000$ citation edges from the induced subgraph with a fixed seed, excluded self-loops, and symmetrized them for evaluation (Eq. (9)). This edge set was reused across all depths, geometries, and activations to eliminate resampling variance.

## A.5 Released reproducibility packages and diagnostics

This section defines the *authoritative specification* of the released reproducibility artifacts. The public camera-ready archive is organized into five packages: `diagnostic_csv_core`, `paper_figure_csv_small`, `public_fp_csv`, `patent_subgraph_bundle`, and `scripts_bundle`. Review-time supplementary material will contain a lightweight reviewer bundle with the figure-linked artifacts, public-benchmark summaries, canonical scripts, and a compact diagnostic excerpt; the full packages will be posted immediately upon acceptance.

Package roles.

- **diagnostic_csv_core:** deterministic patent-subgraph diagnostics and per-seed summaries used in the main paper, including the $L = 32$ all-mode native/shared-projector alignment audit.

- **paper_figure_csv_small:** compact figure-linked artifacts (CSV/JSON/manifest exports and pre-rendered PDF/SVG figures) for the principal paper figures, including Figure 2 and Figure 5.

- **public_fp_csv:** Cora/Texas public-benchmark summaries together with the current-definition OGBN-Arxiv three-seed replay, boundary audit, validation metadata, and regeneration scripts.

- **patent_subgraph_bundle:** the full self-contained induced-subgraph data bundle described in Appendix A.4, together with pinned feature-generation metadata (encoder revision, maximum sequence length, pooling, and normalization).

- **scripts_bundle:** curated scripts released under both original filenames and clean canonical aliases, accompanied by a rename map.

At the file level, the released diagnostics include (i) layerwise interlayer drift $D(\ell)$, metric-aware Dirichlet energy $E(\ell)$, label separability $\mathrm{Sep}(\ell)$, and hyperbolic boundary statistics at each training checkpoint; (ii) degree- and class-resolved drift summaries underlying Appendix Figures A.1–A.4; (iii) curvature-sweep diagnostics for HYPT (per-run fixed-point and energy CSVs, plus derived summary CSVs) underlying Appendix Figures B.3–D.5; and (iv) deterministic coordinate/manifest exports for the geometry and regime-view visualizations; and (v) native/shared-projector per-run and aggregate outputs for the $L = 32$ EUCLID/HYPA/HYPT alignment audit. CSV and JSON artifacts include metadata fields (depth $L$, geometry, activation, training seed, split identifier, and/or selection manifest) sufficient to support exact regeneration of the reported plots. Appendix B exclusively reuses these released artifacts and does not introduce additional specifications.

## A.6 Probe-set resampling ablation

To assess whether the qualitative depthwise regime transition could be a probe-set sampling artifact, we reran the fixed-point drift probe while *resampling* the probe node set $\mathcal{S}$ independently across depths ($|\mathcal{S}| = 1000$, 20 resamples) for a representative HYPA configuration (ReLU, train seed 0), while keeping the evaluation-time neighbor-sampling seed fixed (seed_eval=0). Figure A.6 plots drift$_{p90}$ at the final transition $\ell = L - 1$ as a function of depth $L$, showing near-identical curves across resamples. Quantitatively, the standard deviation over resamples was 0.088 at $L = 2$, 0.176 at $L = 4$, and below $3 \times 10^{-3}$ for $L \geq 8$ (see also jitter_by_depth.csv in the experiment outputs). This supports that the depthwise transition observed in Figure B.5 is not driven by probe-node resampling. In the main experiments, we nevertheless fix $\mathcal{S}$ and $\mathcal{E}_{\mathrm{eval}}$ across all conditions to eliminate any such variance and ensure exact reproducibility from the released artifacts.

Table A.4: Random seeds and construction parameters influencing each intermediate artifact. All reported results are reproducible by fixing the listed seeds and the induced-subgraph construction parameters recorded in `meta.json`.

| Artifact/CSV file | Random seed(s)/fixed params | What is fixed (reproducibility note) |
|---|---|---|
| `meta.json` | `seed`=0, `hops`=2, `seeds_per_split`=50,000 | Defines the released *training-time induced subgraph* construction and the balanced split seed set size. In our bundle, these yield an induced subgraph with $|V| = 479{,}533$ and $|\mathcal{E}| = 1{,}864{,}213$ (directed). |
| `selected_nodes/split-*_seed-eval0.npy` ($\mathcal{S}_{\text{test}}$, $\mathcal{S}_{\text{val}}$) | `seed_eval`=0, `num_probe`=50,000 | Fixes the split-specific canonical node IDs. Because each test/validation split has 50,000 nodes, the canonical pass uses the complete displayed split. Smaller budget experiments sample within that split. |
| `fp_L*_*_*.csv` | explicit split-specific selected-node file; `seed_eval`=0 | The selected-node file fixes the node IDs. `seed_eval` controls evaluation-time neighbor sampling and the deterministic layer projection. Given fixed checkpoints and selected-node IDs, drift, Eq. (15), and radius summaries are reproducible. |
| `drift_groups_L*_*_*.csv` | explicit split-specific selected-node file; `seed_eval`=0 | Uses the same node selection and evaluation seed as the corresponding protocol CSV. Degree/class assignments are deterministic given the induced subgraph, labels, and selected nodes. |
| `energy_L*_*_relu.csv` | `seed_edges`=0; `seed_eval`=0 | `seed_edges` fixes $\mathcal{E}_{\text{eval}}$ (reused across conditions). `seed_eval` controls neighbor sampling used to extract layerwise representations before computing Eq. (8). Given fixed checkpoints and fixed $\mathcal{E}_{\text{eval}}$, energy curves are deterministic. |
| `observation_map_alignment/` | training seeds 0–2; projector seeds 0–9; one class-balanced and five random node subsets | Fixes the $L = 32$ native/shared-projector alignment audit for EUCLID, HYPA, and HYPT on the complete test split. The released aggregate table is regenerated from the per-run CSVs by `summarize_observation_map_alignment.py`. |

Table A.5: OGBN-Arxiv GraphSAGE configuration for the complete three-seed replay. This public-benchmark HYPT setting is distinct from the patent HYPT control and from the OGBN intervention baselines.

| Item | Value |
|---|---|
| Architectures / geometries | GraphSAGE; EUCLID and HYPT |
| Depths / training seeds | $L \in \{2, 4, 8, 16, 32\}$; seeds 0, 1, 2 |
| Hidden / output dimension | 128 / 64 |
| Dropout | 0.2 |
| Optimizer / learning rate | Adam / $3 \times 10^{-3}$ |
| Weight decay | $1 \times 10^{-4}$ |
| Batch size / training fanout | 2048 / 15 |
| Evaluation fanout | 100 |
| Epoch budget | 200 at every depth |
| Checkpoint selection | Best validation macro-F1 |
| HYPT input scale / curvature | $\alpha_{\text{in}} = 0.1$, $c = 3.0$ |
| Tangent MLP | 0 (disabled) |
| Projection / stabilizer | $\varepsilon_{\text{proj}} = 10^{-5}$, $\varepsilon = 10^{-8}$ |
| Fixed probe nodes | Complete test split, 48,603 nodes |
| Fixed energy edges | Released symmetrized set derived from the 200k-edge sample |

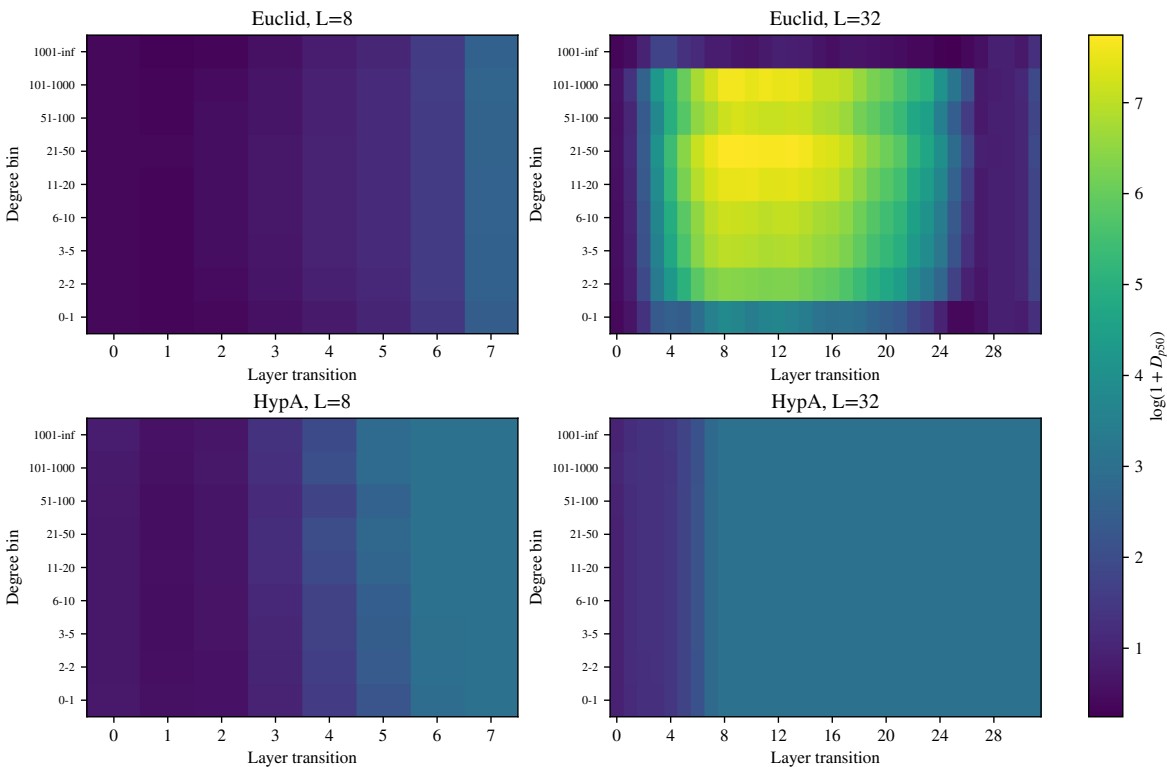

Figure A.1: Degree-resolved geodesic/Euclidean drift heatmaps (p50) for ReLU models at $L \in \{8, 32\}$. All finite cells use one shared numerical normalization of $\log(1 + D_{\mathrm{p50}})$ so that layer and degree patterns are displayed consistently. Because Euclidean and Poincaré distances are not metric-equivalent, quantitative color comparisons are made within a geometry across depth and degree bins, not between geometries.

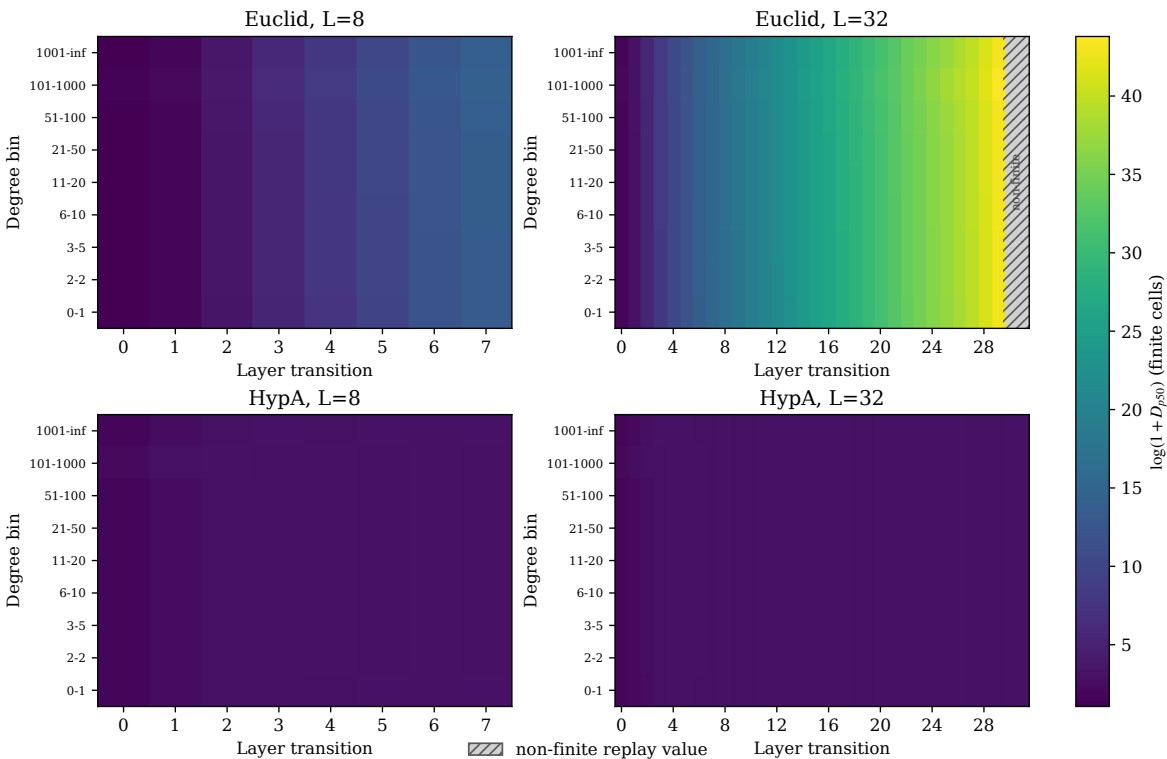

Figure A.2: Degree-resolved geodesic/Euclidean drift heatmaps (p50) for the post-hoc ReLU→Identity replay at $L \in \{8, 32\}$. All *finite* cells use one shared numerical normalization of $\log(1 + D_{p50})$; this scale differs from Figure A.1 because the identity replay has a much larger dynamic range. In the Euclidean $L=32$ replay, the grouped p50 drift becomes non-finite at transitions $30 \to 31$ and $31 \to 32$. These cells are shown by the gray hatched region and are excluded from the finite-value normalization; they indicate numerical divergence, not zero drift or a finite value beyond the yellow end of the colorbar. As in Figure A.1, absolute Euclidean and Poincaré color magnitudes are not interpreted as metric-equivalent.

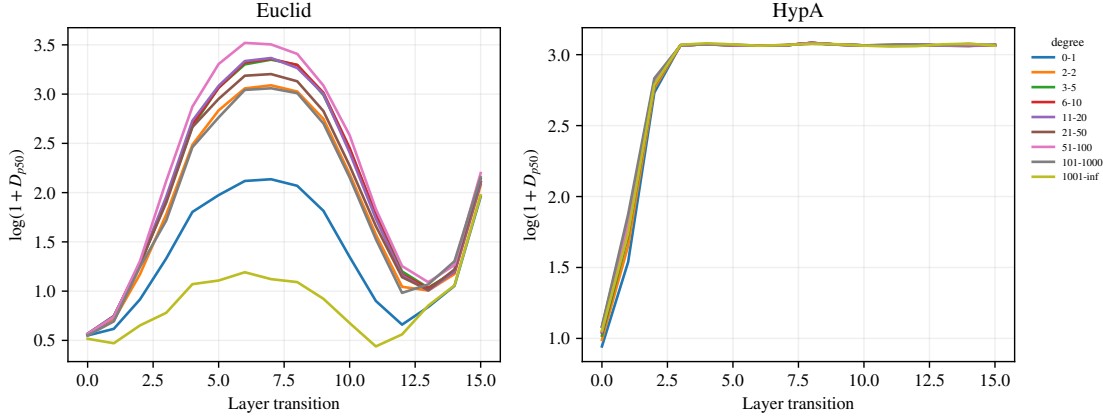

Figure A.3: Representative degree-bin drift trajectories at $L = 16$ on the $\log(1 + D_{p50})$ scale. Euclidean and HYPA panels use independent vertical scales; the figure is intended to compare within-geometry onset patterns, not absolute drift magnitudes across geometries.

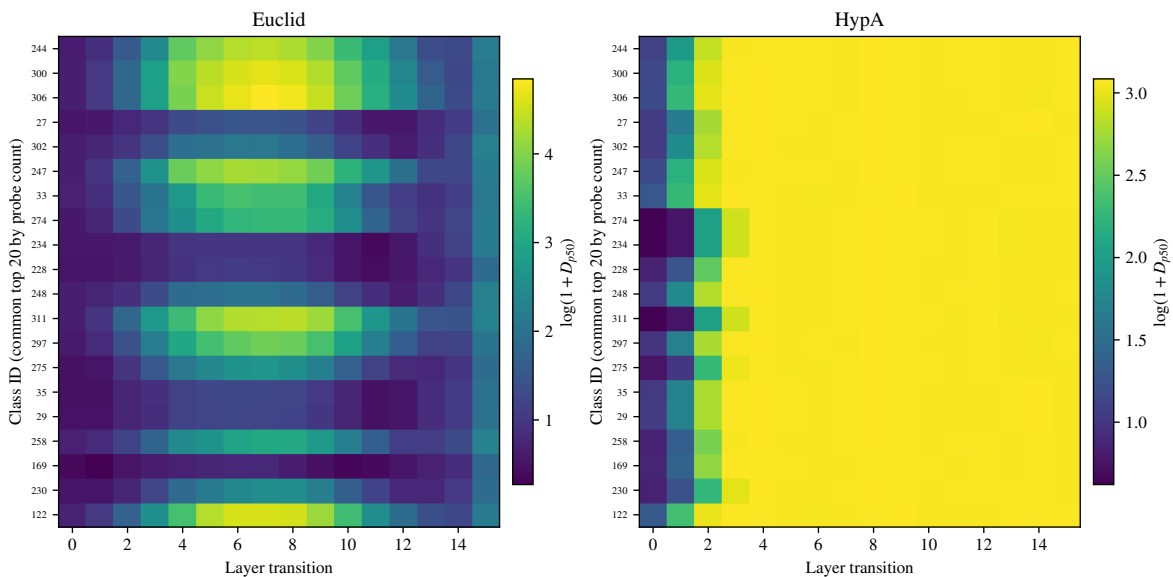

Figure A.4: Class-resolved drift heatmaps at $L = 16$ for a common set of the 20 most frequent represented classes. Each panel has its own normalization and its own colorbar, so colors support within-panel pattern comparison only.

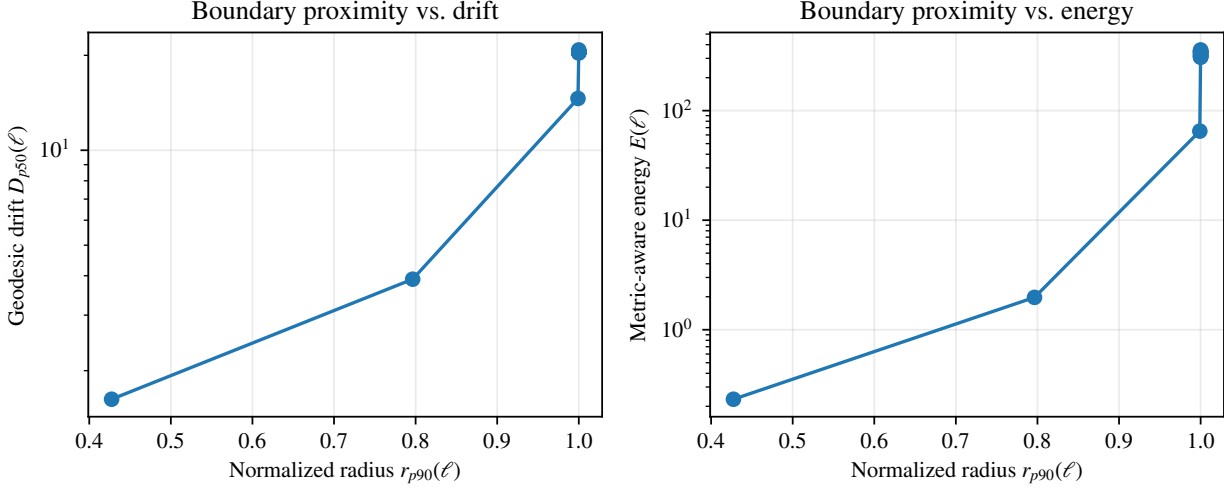

Figure A.5: Boundary association in HYPA at $L = 16$: (a) Poincaré geodesic drift $D_{\mathrm{p50}}(\ell)$ and (b) metric-aware Dirichlet energy $E(\ell)$ plotted against the curvature-normalized upper-tail radius $r_{\mathrm{p90}}(\ell)$. The horizontal axis is not the mean radius $\bar{r}(\ell)$.

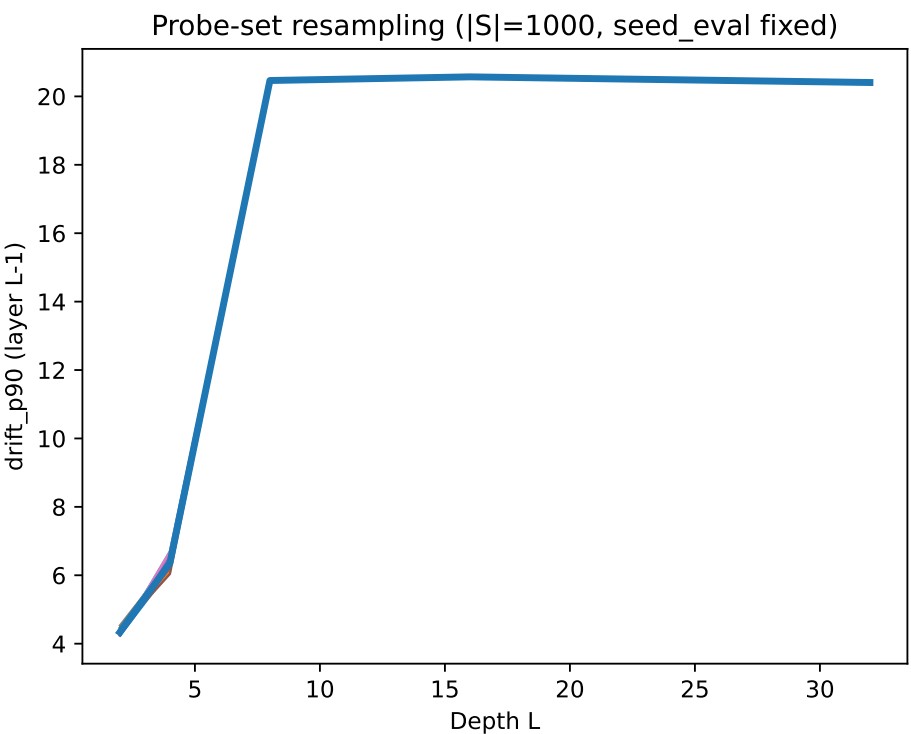

Figure A.6: Probe-set resampling ablation (HYPA, ReLU, train seed 0; $|\mathcal{S}|$=1000; `seed_eval`=0). Each curve corresponds to one resample of $\mathcal{S}$ (resampled independently across depths).

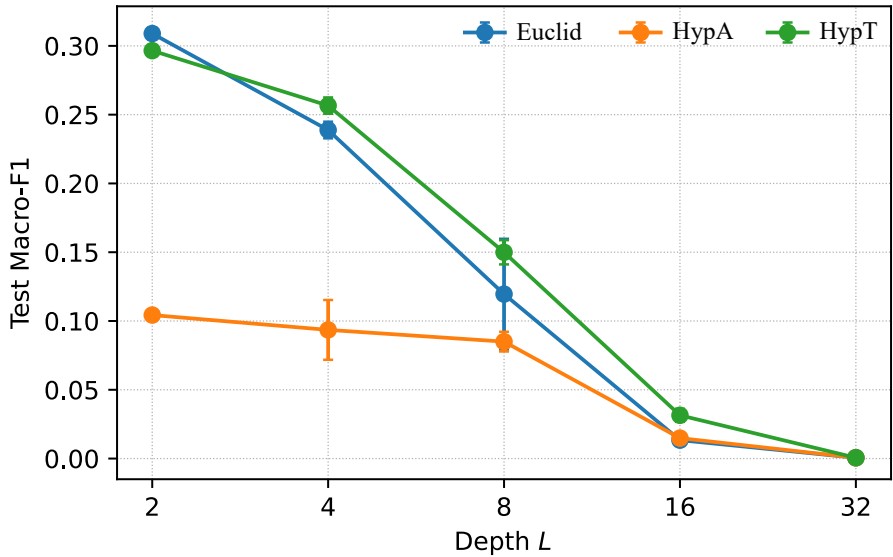

Figure B.1: Macro-F1 (test) versus depth $L$ for Euclidean, HYPA, and HYPT GraphSAGE. Endpoint performance alone does not reveal the layerwise regime transitions exposed by fixed-point probing (see Figures 1 and B.4). The deepest $L$=32 points should be read only as endpoint-collapse references.

## B Additional Depth and Decomposition Results

### B.1 Macro-F1 versus depth

For reference, Figure B.1 summarizes the test macro-F1 as a function of depth for the Euclidean, hyperbolic (HYPA), and tuned hyperbolic (HYPT) GraphSAGE models (mean±std over the three training seeds). HYPT uses a tangent-space classifier head with $\alpha_{\mathrm{in}}$=1.0, curvature $c$=3.0, and a learning-rate split (encoder multiplier 1.0, head multiplier 3.0; base learning rate $3 \times 10^{-3}$). At $L$=32, all three families reach collapse-level macro-F1 in the $10^{-3}$ range; under the strongly imbalanced 316-class setting, these values are consistent with near-single-class prediction and should be read only as endpoint-collapse references. Although the endpoint performance changes with depth, the probe signals in the main text reveal regime transitions in representation dynamics that cannot be reliably inferred from the endpoint performance alone. Full per-seed values are released with the CSV diagnostics to make this variance visible rather than hidden by rounding.

### B.2 Patent-subgraph GCN backbone check

As an architecture robustness check, we repeated the patent-subgraph experiment with a Euclidean GCN backbone on the same patent subgraph under the same fixed-point probing protocol, reusing the same fixed probe-node set $\mathcal{S}$ and fixed evaluation-edge set $\mathcal{E}_{\mathrm{eval}}$ as in the main experiments. Because this GCN variant is evaluated with neighbor-sampling loaders, its degree normalization is approximate on sampled subgraphs; we therefore treat the result as a diagnostic backbone check rather than an optimized benchmark comparison.

The summary plot is shown in Figure 5 in the main text. The key observation is that the qualitative probe–performance decoupling persists beyond GraphSAGE, but with an architecture-dependent internal profile: for $L$=16, both drift and metric-aware Dirichlet energy peak in mid-depth layers rather than only at the final transition. Table B.2 supplements that figure with depthwise probe–performance correlations computed on the same patent-subgraph sweep; we treat them as descriptive because the sweep contains only five depth points.

Table B.1: Macro-F1 (test) versus depth for the same models as Figure B.1. Values report mean±std over the three training seeds (rounded). The $L$=32 entries are collapse-level endpoint references in this imbalanced many-class setting, and full per-seed values are included in the released CSV diagnostics (Appendix A.5).

| mode | $L$ | test macro-F1 |
|---|---|---|
| EUCLID | 2 | $0.309 \pm 0.004$ |
| EUCLID | 4 | $0.239 \pm 0.006$ |
| EUCLID | 8 | $0.120 \pm 0.040$ |
| EUCLID | 16 | $0.013 \pm 0.003$ |
| EUCLID | 32 | $0.001 \pm 0.000$ |
| HYPA | 2 | $0.104 \pm 0.002$ |
| HYPA | 4 | $0.094 \pm 0.022$ |
| HYPA | 8 | $0.085 \pm 0.007$ |
| HYPA | 16 | $0.015 \pm 0.001$ |
| HYPA | 32 | $0.001 \pm 0.000$ |
| HYPT | 2 | $0.297 \pm 0.002$ |
| HYPT | 4 | $0.257 \pm 0.006$ |
| HYPT | 8 | $0.150 \pm 0.009$ |
| HYPT | 16 | $0.031 \pm 0.005$ |
| HYPT | 32 | $0.001 \pm 0.000$ |

Table B.2: Spearman correlation between test macro-F1 and selected probe summaries across depth on the patent subgraph. Correlations were computed over $L \in \{2, 4, 8, 16, 32\}$ using the mean test macro-F1 over three training seeds. (We present $n$, the number of depth points used after excluding undefined probe values.) These correlations are descriptive only: each row uses at most five depth points and is not presented as inferential evidence.

| mode | probe summary | Spearman $\rho$ | $n$ |
|---|---|---|---|
| EUCLID | final drift p50 | 0.700 | 5 |
| EUCLID | final energy mean | 0.400 | 5 |
| EUCLID | Eq. (15) final separability | 0.100 | 5 |
| HYPA | final drift p50 | -1.000 | 5 |
| HYPA | final energy mean | -0.900 | 5 |
| HYPA | Eq. (15) final separability | -0.900 | 5 |
| HYPT | final drift p50 | -0.600 | 5 |
| HYPT | final energy mean | 0.300 | 5 |
| HYPT | Eq. (15) final separability | -0.700 | 5 |

## B.3 Extended Depth Sweeps

Additionally, we swept depths up to $L = 32$ and observed that the qualitative trends reported in the main text persist at larger depths. All layerwise probe traces (drift, Dirichlet energy, separability, and mean radius for hyperbolic models) are provided in the released CSV diagnostics (Appendix A.5).

## B.4 Degree-Resolved Representation Dynamics

We further decomposed representation drift by node degree. Figures A.1 and A.2 use one shared color normalization within each figure, whereas Figure A.3 uses independent geometry-specific vertical scales. These panels are therefore used to compare the location and onset of degree-dependent changes; absolute Euclidean and hyperbolic drift magnitudes are not compared.

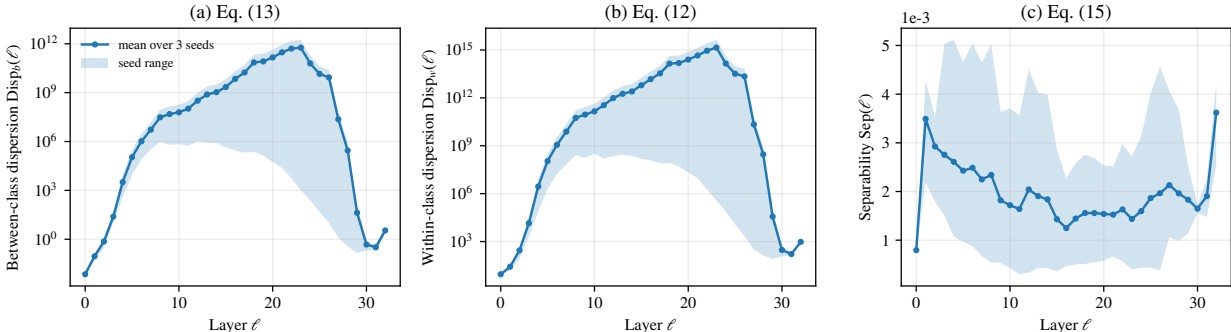

Figure B.2: **Euclidean class-dispersion components at** $L = 32$**.** Curves show the between-class term, within-class term, and Eq. (15) ratio over three training seeds. The ratio is interpreted only with its numerator and denominator.

## B.5 Class-Resolved Dynamics

To assess whether the observed drift pattern is concentrated in particular CPC subclasses, we condition the summaries on a common set of the 20 most frequent represented classes. Figure A.4 uses a separate normalization and colorbar for each geometry; color therefore supports within-panel pattern comparison only. No single displayed class uniquely accounts for the elevated hyperbolic pattern.

## B.6 Alternative separability proxies and dispersion decomposition

In addition to Eq. (15), we evaluated cosine- and kNN-based proxies; their qualitative depth trends were consistent in the tested runs. The main caution is visible in the Euclidean $L = 32$ decomposition below: the between- and within-class terms co-vary over many orders of magnitude, so the ratio can rise while both components contract or rebound.

## B.7 Curvature robustness sweep for HypT

A recurring concern in Euclid–hyperbolic comparisons is whether the observed very-deep degradation in hyperbolic models is an artifact of a particular curvature choice. To probe this, we swept the Poincaré-ball curvature $c \in \{1, 2, 3, 5\}$ for HYPT across depths $L \in \{2, 4, 8, 16, 32\}$ (three training seeds). Figure B.3 summarizes (left) the best validation metric, shown only as a model-selection reference, (middle) the final-transition drift $D_{\mathrm{p90}}(L-1)$, and (right) the last-layer metric-aware Dirichlet energy. Across the tested curvatures, the qualitative very-deep regime persists (e.g., at $L{=}32$), while the absolute probe scales vary with $c$, supporting the view that the late-depth failure is not attributable to a single curvature setting.

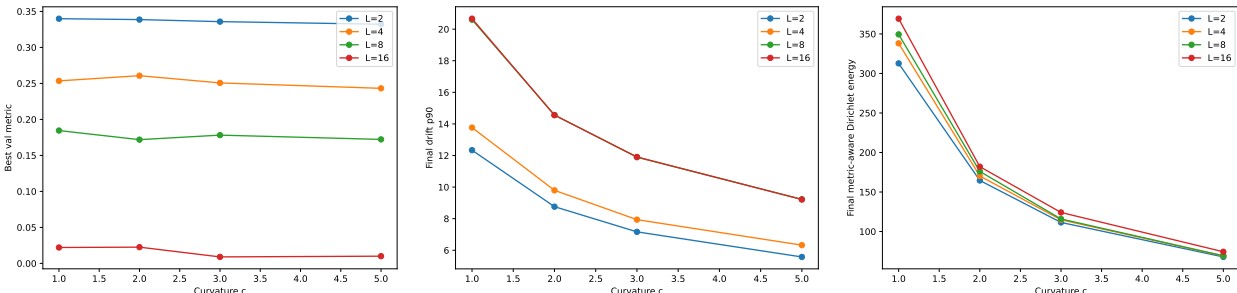

Figure B.3: **Curvature robustness sweep for HypT.** We swept $c \in \{1, 2, 3, 5\}$ across depths up to $L=32$. Left: best validation metric (reference only for model selection). Middle: final-transition drift $D_{\mathrm{p90}}(L-1)$. Right: last-layer metric-aware Dirichlet energy. Varying $c$ changes absolute probe scales, but does not remove the qualitative very-deep regime transition.

### B.8 Pointwise-nonlinearity replay

We replayed the same seed-0 $L = 16$ checkpoints with the trained ReLU path and with the intermediate pointwise ReLU removed at evaluation. Weights, supports, evaluation sampling, and layer projections remain fixed. This is a post-hoc sensitivity test, not a separately optimized identity-activation model. Removing ReLU destabilizes the Euclidean replay, while the HYPA replay remains elevated through most transitions and changes abruptly near the end.

### B.9 Validation/test split stability of the operational final transition

The validation and test probes show closely aligned depthwise ordering for the operational final-transition drift. This supports split stability of that summary but does not convert a width-changing or layer-indexed transition into an intrinsic fixed-point measurement.

## C Cross-Dataset Portability Checks on Public Benchmarks

This appendix reports direct public-dataset instantiations of fixed-point probing on Cora, Texas, and OGBN-Arxiv. The complete current geometry-aware probe family is reported for OGBN-Arxiv GraphSAGE in the main text and detailed here with boundary-distribution and intervention audits. Cora and Texas provide replicated Euclidean depthwise components under the released public-benchmark convention: drift and class separability on both datasets, with metric-aware Dirichlet energy additionally reported for Cora. Their released separability statistic is labeled as the public-benchmark convention and is not relabeled as the current global Eq. (15) statistic.

The purpose is protocol instantiation and diagnostic coverage, not optimized benchmark performance or transfer of numerical thresholds. OGBN-Arxiv additionally includes a single-seed GCN reference and exploratory boundary-intervention pilots. Appendix E.1 maps each dataset to its exact coverage, replication convention, and supported conclusion.

### C.1 Depth Sweeps and Layerwise Diagnostics

Figures C.6 and C.7 show class separability at the final layer as a function of depth. Across both benchmarks, separability exhibited a nonmonotonic dependence on depth, qualitatively mirroring the trends observed in the patent citation graph.

Figures C.8 and C.9 show layerwise interlayer drift and class separability at a fixed depth ($L = 16$). These probes illustrate that stable endpoint performance (or small performance changes) does not necessarily coincide with stable layerwise representation dynamics.

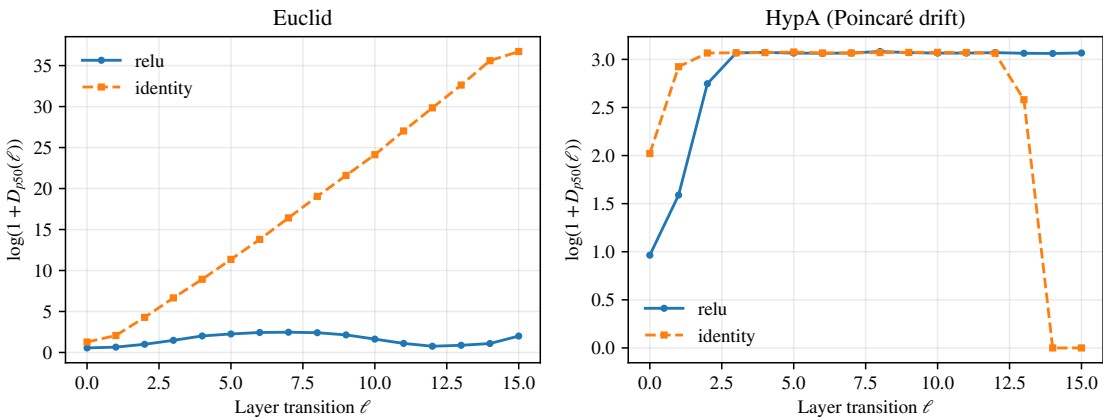

Figure B.4: **Post-hoc ReLU→Identity replay at** $L = 16$. The geometries use independent vertical scales. Removing the pointwise nonlinearity does not remove the elevated HYPA drift regime.

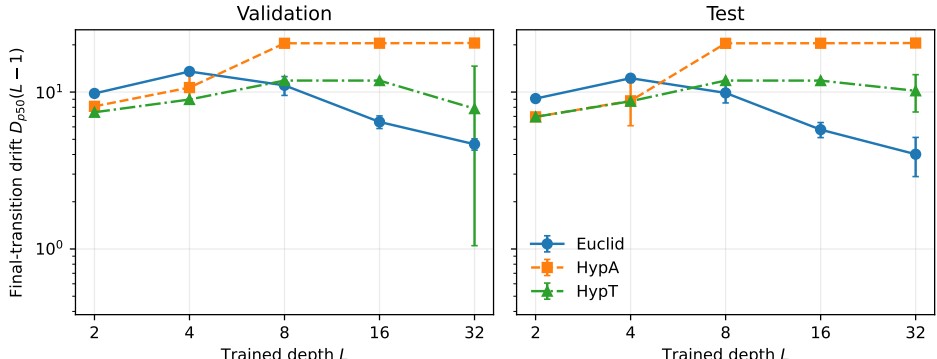

Figure B.5: **Operational final-transition drift on validation and test probes.** Points show mean±std over three training seeds. Each value is $D_{p50}(L - 1)$ on a fixed split-specific probe; aligned hidden-to-hidden dynamics are audited separately in Appendix E.7.

For completeness, we also present the metric-aware Dirichlet energy for Cora (Figure C.10) and show how the validation accuracy can obscure substantial variations in separability (Figure C.11). A single-seed OGBN-Arxiv GCN reference is reported in Figure C.1. We also report exploratory OGBN-Arxiv intervention pilots in Figure C.2 and tables C.1 to C.3, together with appendix sanity and robustness checks in Figures C.3 to C.5 and table C.4.

## C.2  OGBN-Arxiv complete GraphSAGE replay and GCN reference check

The complete current-definition OGBN-Arxiv GraphSAGE replay is reported in the main text (Figure 6 and Table 4) over training seeds 0, 1, and 2. The full test split and one fixed symmetrized evaluation-edge set are reused across every depth, geometry, and seed. Figure C.13 provides the corresponding boundary-persistence and terminal-distribution audit.

Figure C.1 supplies a single-seed GCN backbone reference. The complete current-definition GraphSAGE result in Figure 6 and Table 4 is the authoritative OGBN-Arxiv readout.

At $L = 32$, the three validation-selected HYPT checkpoints occur at epochs 7, 3, and 2 of 200 and all have test macro-F1 0.0027686. Their terminal normalized-radius means are 0.137, 0.880, and approximately 1.000; the corresponding median radii are 0, approximately 1, and approximately 1. The shared endpoint collapse therefore does not identify a unique terminal boundary distribution. We report the runs as outputs of the

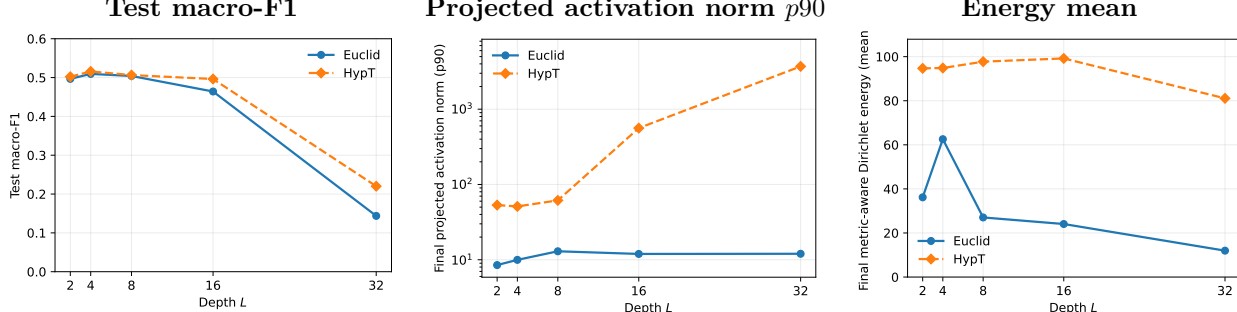

Figure C.1: **OGBN-Arxiv GCN single-seed reference diagnostics.** The same fixed-subset reporting pattern is applied to a GCN backbone. Left: test macro-F1. Middle: projected activation norm $p90$ before any hyperbolic exponential map. Right: metric-aware Dirichlet energy. This is a seed-0 architecture reference, not a three-seed complete-family replication.

| Setting | Role | $\alpha_{\text{in}}$ | $c$ | tMLP | $r_{\text{p90}}^{\text{late}}$ | test macro-F1 |
|---|---|---|---|---|---|---|
| a0.3_c0.3_tmlp0 | performance baseline used | 0.3 | 0.3 | 0 | 1.000 | 0.491 |
| a1.0_c1.0_tmlp0 | stable baseline used | 1.0 | 1.0 | 0 | 1.000 | 0.486 |
| a0.3_c1.0_tmlp0 | reference candidate | 0.3 | 1.0 | 0 | 1.000 | 0.489 |
| a0.3_c3.0_tmlp0 | reference candidate | 0.3 | 3.0 | 0 | 1.000 | 0.484 |
| a0.1_c1.0_tmlp0 | reference candidate | 0.1 | 1.0 | 0 | 1.000 | 0.483 |

Table C.1: **Representative boundary-saturated, non-collapsed OGBN-Arxiv $L$=16 HypT baselines.** The first two rows are the exact performance-oriented and stability-oriented configurations used in the intervention experiments. The remaining rows are representative boundary-saturated reference candidates from the same seed-0 baseline search. Here $r_{\text{p90}}^{\text{late}}$ is the late-window mean normalized radius. This table is an evidence map for the selected operating points, not a top-five ranking.

stated training-and-selection protocol and do not interpret them as converged replicas of one very-deep mechanism.

### C.3 OGBN-Arxiv intervention pilots on HypT

To probe whether the boundary-control interpretation transfers beyond the patent graph, we ran a small set of exploratory radius-penalty pilots on OGBN-Arxiv with HYPT. Because an exploratory $L$=32 search frequently produced an *origin-side collapse* before the intended late-only window, we concentrated the most interpretable controlled comparisons on $L$=16, where a small baseline search identified boundary-saturated yet non-collapsed configurations. In this subsection, $r_{\text{p90}}^{\text{late}}$ denotes the mean of the layerwise upper-tail radius summary $r_{\text{p90}}(\ell)$ over the designated late-layer window, and $\text{frac}_{\text{sat,last}}$ denotes the corresponding last-layer saturation ratio.

For the intervention pilots, we selected two representative $L$=16 baselines that span a more performance-oriented operating point (`performance: a0.3_c0.3_tmlp0`) and a more stable operating point (`stable: a1.0_c1.0_tmlp0`). The setting names and curvatures here match the intervention directories and training commands. Figure C.2 summarizes the OGBN-Arxiv $L$=16 feasibility and intervention outcomes at a glance, while Tables C.2 and C.3 report the corresponding numeric details for the initial three-seed late-only pilot and the always-on seed-0 sweep.

We then made the schedule control explicit by repeating late-only sweeps with start fractions 0.75, 0.60, and 0.50. Across these runs, $r_{\text{p90}}^{\text{late}}$ and $\text{frac}_{\text{sat,last}}$ remained at 1.0 in all tested seeds and conditions, even though the training logs showed that the late-only switch became active and the radius-penalty term became nonzero once the scheduled start epoch was reached. In other words, under the tested public $L$=16 settings, applying the same boundary penalty from the last 50% of training was still effectively a no-op.

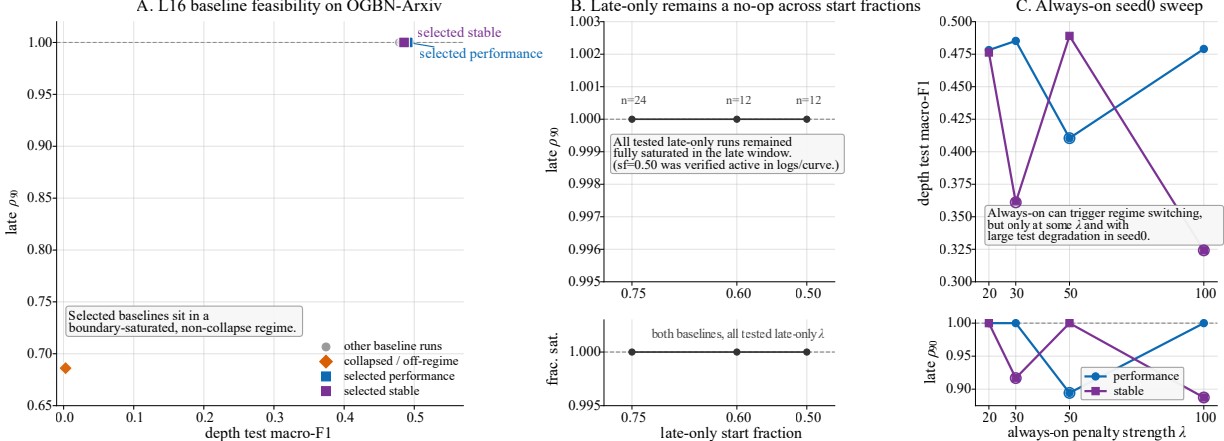

Figure C.2: **Three-panel summary of OGBN-Arxiv $L$=16 intervention pilots on representative HypT baselines.** (A) Baseline feasibility: the selected `performance` and `stable` settings lie in a *boundary-saturated but non-collapsed* regime at $L$=16. (B) Late-only radius penalties remain effectively no-op across start fractions 0.75, 0.60, and 0.50: the boundary proxies stay saturated even when the scheduled penalty becomes active in the training logs. (C) The always-on seed-0 sweep over $\lambda \in \{20, 30, 50, 100\}$ is strongly nonmonotonic: some $\lambda$ values move the boundary proxies, but only with a substantial test macro-F1 cost, whereas adjacent $\lambda$ values are complete no-ops. Taken together, the OGBN-Arxiv pilots support timing-sensitive and basin-like regime switching rather than a smooth, performance-preserving control band.

| Label | Condition | test macro-F1 | $r^{\text{late}}_{\text{p90}}$ | $\text{frac}_{\text{sat,last}}$ |
|---|---|---|---|---|
| performance | $\lambda = 0$ | $0.483 \pm 0.006$ | 1.000 | 1.000 |
| performance | $\lambda = 10$ late-only | $0.487 \pm 0.006$ | 1.000 | 1.000 |
| performance | $\lambda = 100$ always-on | $0.437 \pm 0.079$ | 0.956 | 0.672 |
| stable | $\lambda = 0$ | $0.480 \pm 0.005$ | 1.000 | 1.000 |
| stable | $\lambda = 10$ late-only | $0.481 \pm 0.007$ | 1.000 | 1.000 |
| stable | $\lambda = 100$ always-on | $0.457 \pm 0.050$ | 0.958 | 0.674 |

Table C.2: **Initial OGBN-Arxiv $L$=16 intervention pilot on two representative HypT baselines (three seeds).** The weak late-only control is effectively a no-op: both $r^{\text{late}}_{\text{p90}}$ and the last-layer saturation ratio remain at 1.0 while endpoint performance stays near the $\lambda$=0 baseline. The stronger always-on control partially reduces the boundary proxies, but it also increases variance and degrades endpoint performance. These runs are therefore consistent with a timing-sensitive intervention effect, but not yet with a performance-preserving public control band.

Finally, we ran an always-on seed-0 sweep over $\lambda \in \{20, 30, 50, 100\}$ to test whether the OGBN-Arxiv response is smooth in the intervention strength. The outcome was strongly nonmonotonic (Table C.3): some $\lambda$ values moved the boundary proxies substantially, but only at the cost of a large drop in test macro-F1, while adjacent $\lambda$ values were complete no-ops.

Taken together, these OGBN-Arxiv pilots support three points that are consistent with the main patent-subgraph narrative while remaining appropriately cautious. First, public benchmark hyperbolic models can also realize a boundary-saturated yet non-collapsed regime. Second, once such a regime is established, late-only boundary control can be genuinely active yet still fail to move the representation trajectory. Third, always-on controls can induce regime switches, but the resulting response is strongly timing-sensitive and nonmonotonic under the tested HypT settings.

| Label | $\lambda$ | $\Delta$ test | $r_{\mathrm{p90}}^{\mathrm{late}}$ | $\mathrm{frac}_{\mathrm{sat,last}}$ | Interpretation |
|---|---|---|---|---|---|
| performance | 20 | +0.000 | 1.000 | 1.000 | no-op |
| performance | 30 | +0.007 | 1.000 | 1.000 | no-op |
| performance | 50 | −0.067 | 0.894 | 0.026 | moved / costly |
| performance | 100 | +0.001 | 1.000 | 1.000 | no-op |
| stable | 20 | +0.000 | 1.000 | 1.000 | no-op |
| stable | 30 | −0.115 | 0.917 | 0.063 | moved / costly |
| stable | 50 | +0.013 | 1.000 | 1.000 | no-op |
| stable | 100 | −0.152 | 0.887 | 0.026 | moved / costly |

Table C.3: **Always-on OGBN-Arxiv $L{=}16$ seed-0 sweep relative to the corresponding $\lambda{=}0$ baseline.** The OGBN-Arxiv intervention response is highly nonmonotonic. For the performance-oriented baseline, only $\lambda{=}50$ moved the boundary proxies, and it did so with a marked loss in test macro-F1. For the stable baseline, $\lambda{=}30$ and $\lambda{=}100$ both moved the proxies, again with substantial endpoint degradation. We therefore interpret these pilots as supportive diagnostic evidence for basin-like regime switching, not as evidence for a smooth or dataset-universal intervention threshold.

### C.3.1 Boundary-intervention sanity checks and robustness at $L{=}16$

We performed a focused set of sanity and robustness checks to evaluate whether the OGBN-Arxiv $L{=}16$ boundary-saturated but non-collapsed regime can be reliably controlled by the same radius-based penalty family. We considered the same two representative HYPT baselines used in the OGBN-Arxiv intervention pilot: a performance-oriented setting and a stability-oriented setting. For late-only interventions, we swept the penalty onset and confirmed from the training logs and curve files that the regularizer was implemented correctly: the late-only switch became active at the specified epoch and the radius-penalty term became nonzero in the objective. Despite this correct activation, the boundary proxies stayed saturated throughout training, with $r_{\mathrm{p90}}$ and $\mathrm{frac}_{\mathrm{sat}}$ remaining essentially at 1.0. This shows that late-only control is genuinely active yet still ineffective in moving the trained system away from the boundary-saturated regime.

We then examined cross-seed robustness for the always-on controls. The initial seed-0 sweep suggested several nonmonotonic moved cases, but these did not generalize across seeds. When the informative always-on settings were rerun for seeds 1 and 2, all of them behaved as no-ops with saturated boundary proxies and near-baseline test performance. We therefore interpret the initial seed-0 pattern not as evidence of a robust public control band, but rather as seed-conditioned fragility. Figure C.4 summarizes this contrast: late-only conditions remain pinned to the saturated boundary level across all tested start fractions, while the moved always-on cases are confined to the original seed-0 exploratory sweep.

To probe the seed-0 anomalies more directly, we repeated the anomalous conditions multiple times. The performance-oriented `lam50` case did not reproduce the earlier moved behavior, and the stability-oriented `lam30` case moved only rarely. By contrast, the stability-oriented `lam100` case consistently reduced the boundary proxies in all repeats, but this came with a marked drop in validation/test macro-F1. Table C.4 collects the family-level summary, while Figure C.5 shows the repeated stable-`lam100` trajectories directly. A more detailed row-wise CSV summary is released with the artifact bundle; in the paper appendix we show only the compact family-level table to keep the narrative readable.

Taken together, these checks support two conclusions. First, the OGBN-Arxiv $L{=}16$ boundary-saturated regime is hard to perturb once it has formed: late-only penalties remain ineffective even when they are demonstrably active. Second, while always-on penalties can trigger a regime switch in some cases, the response is fragile across seeds and, in the one condition that reproduces robustly, does not preserve predictive performance. We therefore use these appendix experiments as supporting evidence for the hardness and fragility of the regime, rather than as a positive intervention result.

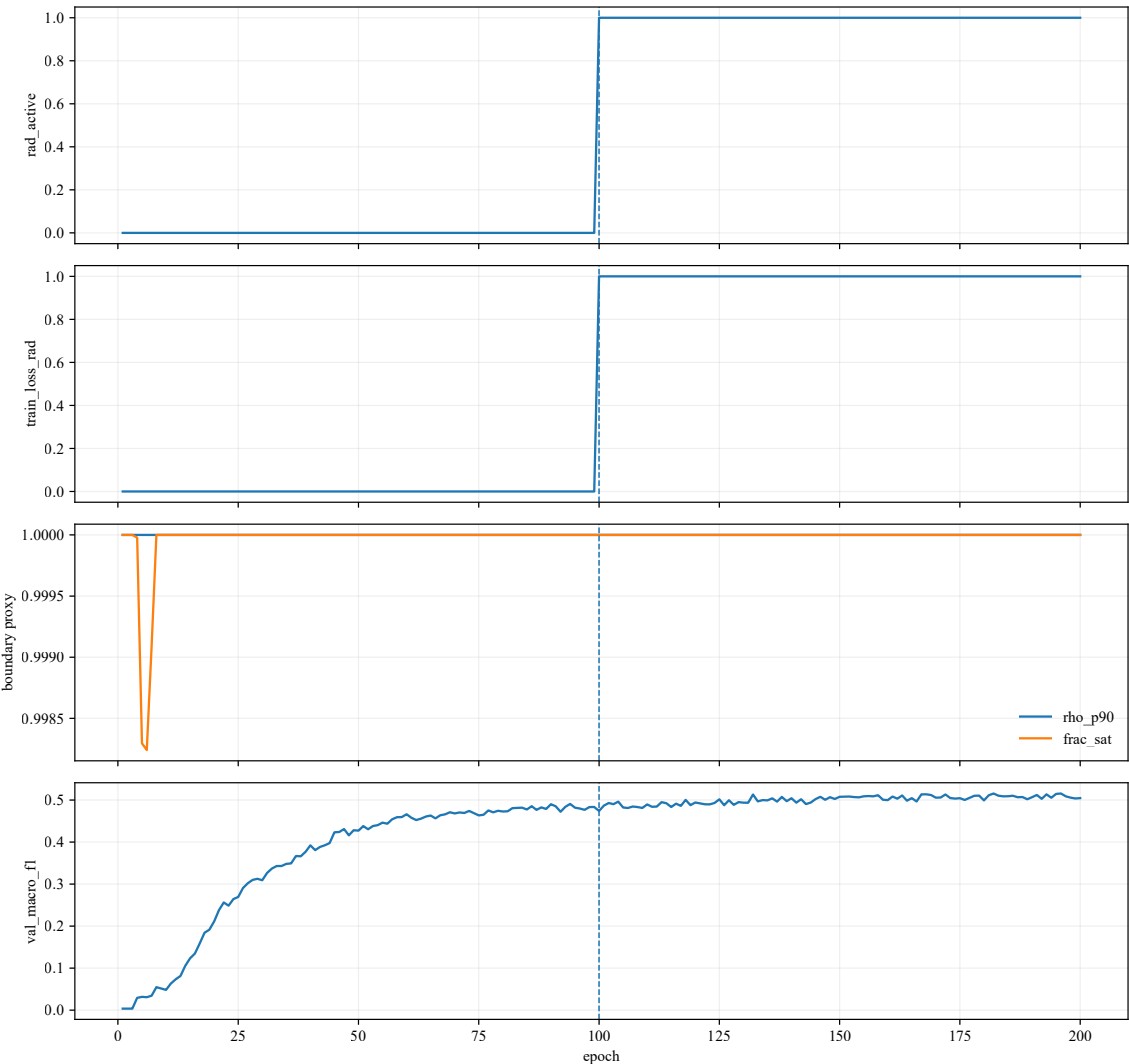

Figure C.3: **Late-only penalty activates, but boundary proxies do not move.** Shown is a representative late-only run at depth 16 (`sf=0.5`, `lam100`). The regularizer activates at the prescribed epoch, as indicated by the jump in `rad_active` and the nonzero `train_loss_rad`. However, both boundary proxies remain saturated ($r_{\mathrm{p90}} \approx 1$, $\mathrm{frac}_{\mathrm{sat}} \approx 1$), and validation macro-F1 continues along its usual trajectory. This confirms that the late-only intervention is implemented correctly, but ineffective in shifting the model away from the boundary-saturated regime.

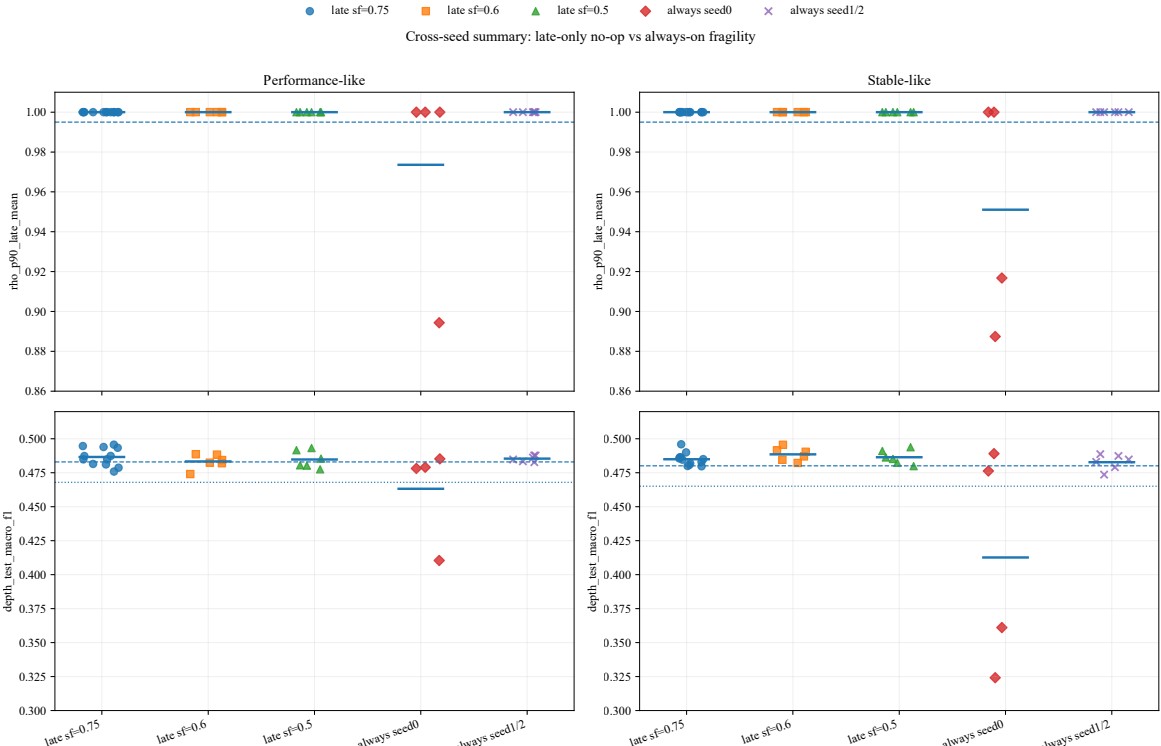

Figure C.4: **Late-only interventions are robustly inactive, whereas always-on responses are seed-fragile.** Cross-seed summary of boundary proxy ($r_{\mathrm{p90}}^{\mathrm{late}}$) and test macro-F1 for the late-only sweeps (sf=0.75, 0.6, 0.5) and the always-on sweeps. All late-only conditions remain at the saturated boundary level across seeds. The moved cases seen in the initial always-on seed-0 sweep do not reproduce for seeds 1 and 2, indicating that the apparent response is not a robust cross-seed control effect.

| Family / setting | $n$ | moved | $r_{\mathrm{p90}}^{\mathrm{late}}$ | $\mathrm{frac}_{\mathrm{sat,last}}$ | $\Delta$ test vs. $\lambda{=}0$ |
|---|---|---|---|---|---|
| late-only, sf= 0.75 | 24 | 0/24 | 1.000 | 1.000 | +0.004 |
| late-only, sf= 0.60 | 12 | 0/12 | 1.000 | 1.000 | +0.004 |
| late-only, sf= 0.50 | 12 | 0/12 | 1.000 | 1.000 | +0.004 |
| always-on, seeds 1–2 ($\lambda \in \{30, 50, 100\}$) | 12 | 0/12 | 1.000 | 1.000 | +0.003 |
| repeat, performance-like lam50 | 8 | 0/8 | 1.000 | 1.000 | −0.001 |
| repeat, stable-like lam30 | 8 | 1/8 | 0.990 | 0.883 | −0.012 |
| repeat, stable-like lam100 | 8 | 8/8 | 0.889 | 0.022 | −0.098 |

Table C.4: **Summary of boundary-intervention robustness at depth** 16. For each intervention family, we report the number of moved runs, the late-window boundary proxy, the last-layer saturation ratio, and the mean test macro-F1 change relative to the matched $\lambda{=}0$ baseline. Late-only settings never produce a moved run, informative always-on settings do not reproduce across seeds 1 and 2, and only the repeated stable-like lam100 condition shows a consistent regime switch, albeit with a clear performance drop.

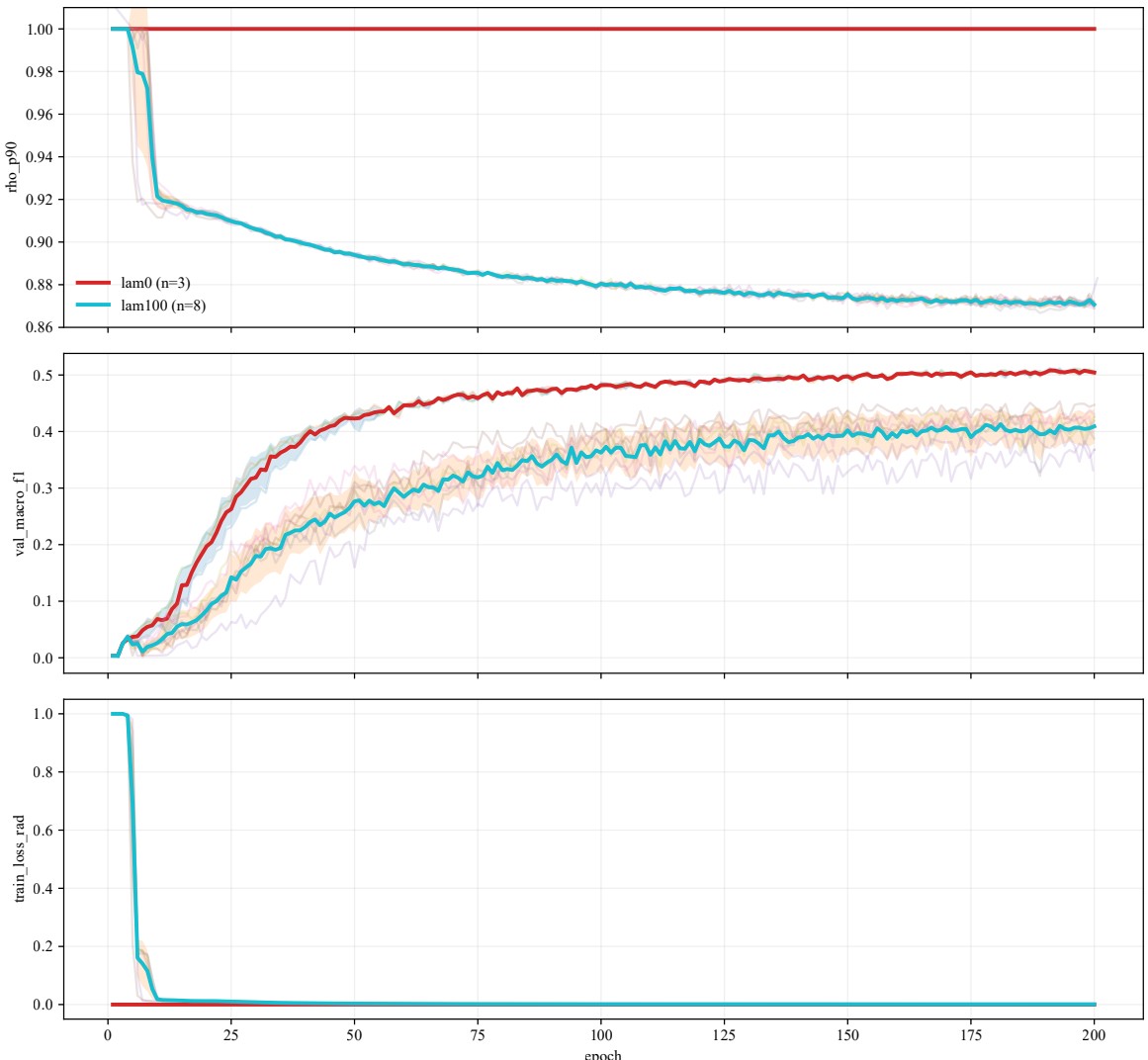

Figure C.5: **In seed-0 repeats, stable `lam100` consistently lowers the boundary proxy but degrades performance.** Mean trajectories with standard-deviation bands for repeated seed-0 runs comparing stable-like `lam0` and stable-like `lam100`. The `lam100` condition consistently drives $r_{\mathrm{p90}}$ downward and keeps the RAD loss active, whereas `lam0` stays boundary-saturated throughout. However, this boundary reduction comes with substantially lower validation macro-F1, showing that the repeatable regime switch does not preserve predictive performance.

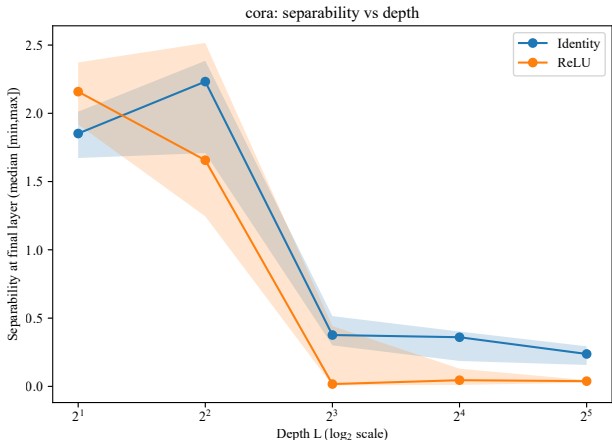

Figure C.6: Class separability at the final layer as a function of depth on the Cora citation network. Depth is shown on a $\log_2$ scale. Shaded area shows the min–max range across five training seeds on the fixed public split.

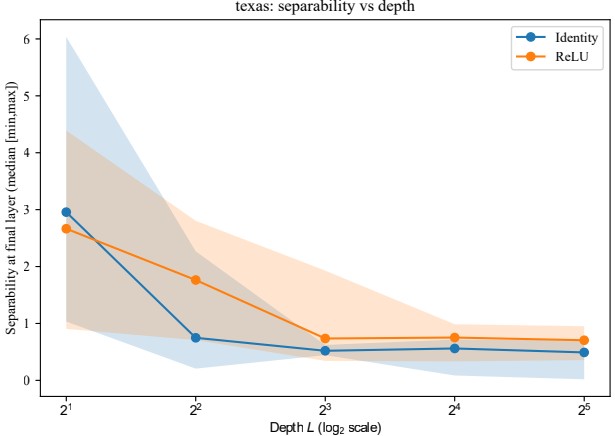

Figure C.7: Class separability at the final layer as a function of depth on the Texas web graph. Shaded area shows the min–max range across five training seeds and ten official data splits (50 seed–split runs).

Table C.5: Test accuracy versus depth on the Cora citation network. Values show the median [min, max] across five training seeds on the fixed public split (reported for reference only).

| depth | identity | relu |
|---|---|---|
| 2 | 0.7180 [0.7010, 0.7450] | 0.7130 [0.6890, 0.7550] |
| 4 | 0.7070 [0.5790, 0.7510] | 0.4900 [0.3700, 0.5600] |
| 8 | 0.1440 [0.0910, 0.1490] | 0.3190 [0.1030, 0.3190] |
| 16 | 0.1300 [0.0640, 0.3190] | 0.3190 [0.1300, 0.3190] |
| 32 | 0.1300 [0.0910, 0.3190] | 0.3190 [0.1440, 0.3190] |

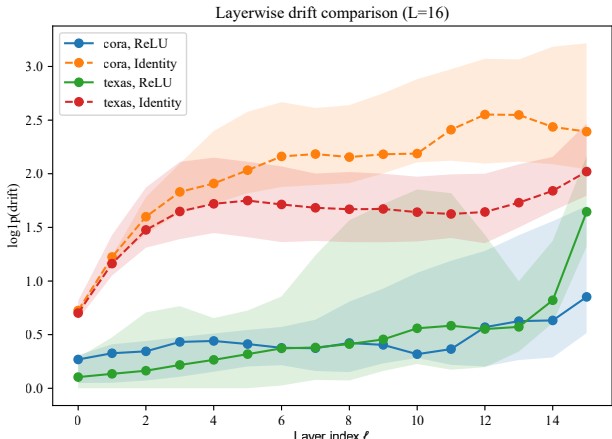

Figure C.8: Layerwise interlayer drift at fixed depth $L = 16$ on public benchmarks. Curves compare ReLU and identity activations on the same axes. Drift is shown using $\mathrm{log1p}(x) = \log(1 + x)$. For Cora, bands summarize five training seeds on one fixed public split; for Texas, bands summarize five seeds across ten official splits (50 seed–split runs).

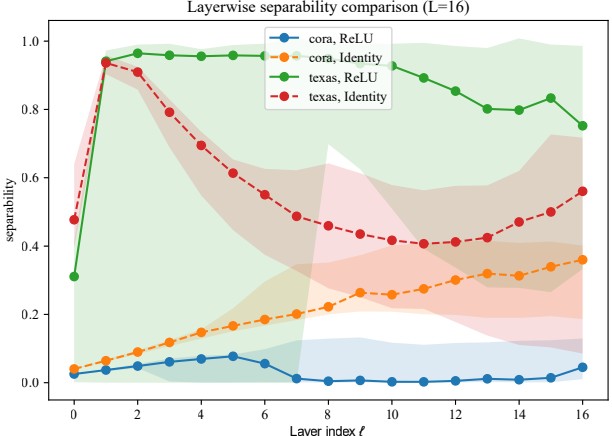

Figure C.9: Layerwise class separability at fixed depth $L = 16$ on public benchmarks. For Cora, bands summarize five training seeds on one fixed public split; for Texas, bands summarize five seeds across ten official splits (50 seed–split runs).

Table C.6: Test macro-F1 versus depth on the Cora citation network. Values show the median [min, max] across five training seeds on the fixed public split (reported for reference only).

| depth | identity | relu |
|---:|---|---|
| 2 | 0.7071 [0.7048, 0.7318] | 0.7061 [0.6996, 0.7495] |
| 4 | 0.6832 [0.4936, 0.7381] | 0.4210 [0.2345, 0.5465] |
| 8 | 0.0360 [0.0238, 0.0371] | 0.0691 [0.0267, 0.0691] |
| 16 | 0.0329 [0.0172, 0.0691] | 0.0691 [0.0329, 0.0691] |
| 32 | 0.0329 [0.0238, 0.0987] | 0.0691 [0.0360, 0.0691] |

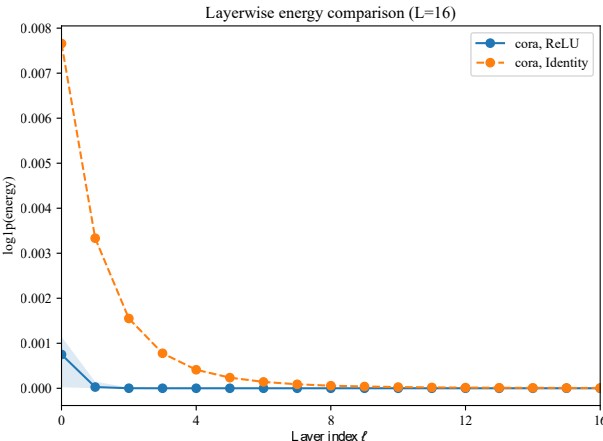

Figure C.10: Layerwise metric-aware Dirichlet energy at fixed depth $L = 16$ on the Cora dataset. Energy is shown using $\log1p(x)$ for numerical stability. Shaded area shows the min–max range across five training seeds on the fixed public split.

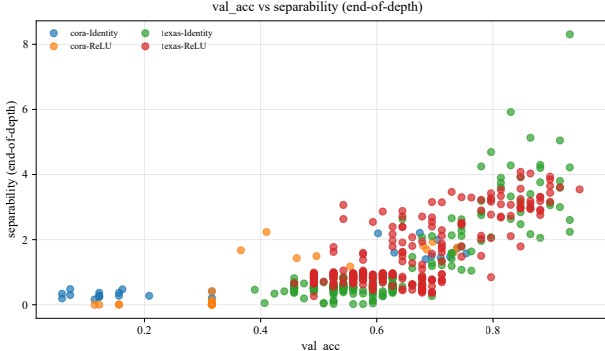

Figure C.11: Validation accuracy versus class separability on public benchmarks. Despite similar accuracy values, separability varies substantially, illustrating that endpoint performance alone does not reliably reflect internal representation quality.

Table C.7: Test accuracy versus depth on the Texas web graph. Values show the median [min, max] across five training seeds and ten official data splits (50 seed–split runs; reported for reference only).

| depth | identity | relu |
|---:|---|---|
| 2 | 0.8108 [0.5946, 0.9189] | 0.7838 [0.5676, 0.8919] |
| 4 | 0.6216 [0.4595, 0.7838] | 0.6757 [0.5676, 0.8108] |
| 8 | 0.5946 [0.4865, 0.6486] | 0.6216 [0.5405, 0.7297] |
| 16 | 0.5946 [0.4865, 0.6486] | 0.6216 [0.4865, 0.7297] |
| 32 | 0.5946 [0.4324, 0.6486] | 0.6216 [0.5135, 0.7297] |

Table C.8: Test macro-F1 versus depth on the Texas web graph. Values show the median [min, max] across five training seeds and ten official data splits (50 seed–split runs; reported for reference only).

| depth | identity | relu |
|---|---|---|
| 2 | 0.6726 [0.3212, 0.8157] | 0.6255 [0.2615, 0.8605] |
| 4 | 0.2703 [0.1309, 0.6846] | 0.4262 [0.2615, 0.7560] |
| 8 | 0.1864 [0.1309, 0.2652] | 0.2914 [0.1533, 0.4019] |
| 16 | 0.1826 [0.1309, 0.3764] | 0.2836 [0.1533, 0.4037] |
| 32 | 0.1864 [0.1309, 0.2981] | 0.2890 [0.1533, 0.4212] |

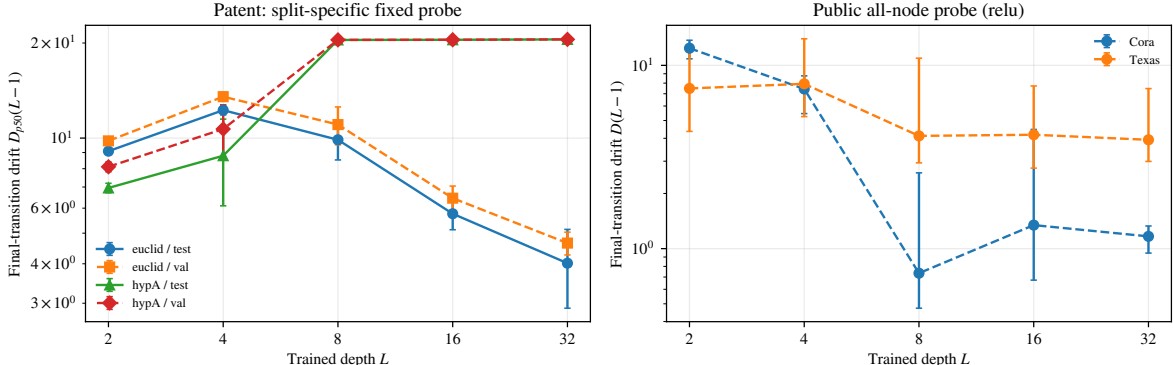

Figure C.12: Canonical final-transition drift as a function of trained depth. Left: patent validation/test summaries use the split-specific fixed probes and show mean±std over three training seeds. Right: Cora and Texas use the all-node public probe under the ReLU setting and show median [min,max] across available runs (Cora: five seeds on one public split; Texas: five seeds across ten official splits). Every point is taken from transition $L - 1 \to L$; no trajectory-median drift is used.

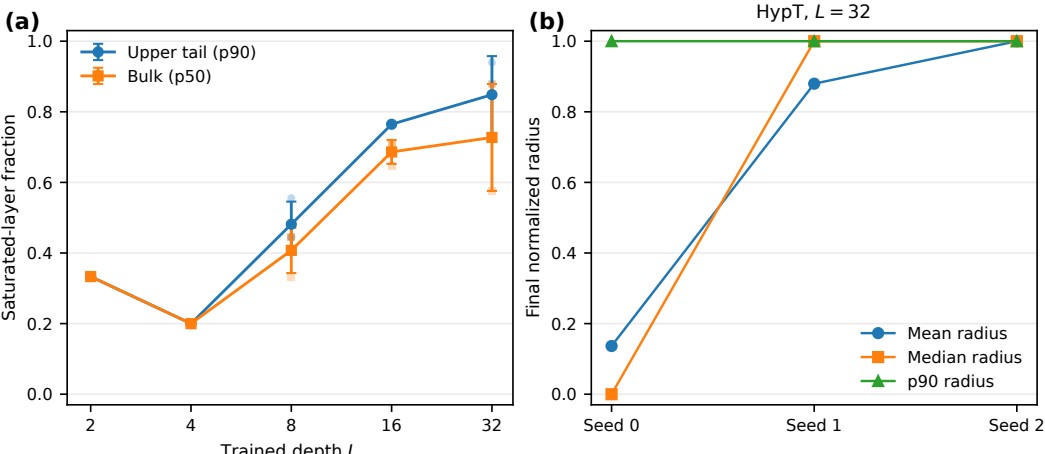

Figure C.13: **Boundary persistence and seed-dependent terminal states on OGBN-Arxiv HypT.**
(a) Fraction of observed layers whose upper-tail or median normalized radius reaches the operational saturation threshold $\tau_r = 0.95$; counts include input and output rows. (b) Final normalized-radius summaries at $L = 32$. The endpoint macro-F1 is identical across the three failed runs, but the terminal boundary distributions differ sharply, showing that $H_\Delta$ or any single terminal-radius statistic is not a reproducible collapse signature.

## C.4 Summary

In summary, the public checks now have explicit, non-interchangeable coverage. OGBN-Arxiv GraphSAGE reports the complete current geometry-aware probe family over three training seeds and demonstrates two forms of state–outcome dissociation: boundary saturation can coexist with non-collapsed performance, and the reproducible $L = 32$ training-and-selection failure exhibits state–outcome dissociation across validation-selected checkpoints. Cora and Texas provide replicated Euclidean depthwise components under their public-benchmark convention, and the OGBN GCN and intervention studies provide additional architecture and control references. These results establish tested protocol instantiation while leaving numerical thresholds and mechanisms dataset-, seed-, and backbone-dependent.

# D Technical details and probe interpretation

**Dirichlet energy evaluation protocol.** For the representation $z^{(\ell)}$, we computed the Dirichlet energy

$$E^{(\ell)} \;=\; \frac{1}{|E_{\text{eval}}|} \sum_{(u,v)\in E_{\text{eval}}} d\Big(z_u^{(\ell)}, z_v^{(\ell)}\Big)^2,$$

where $d$ denotes the geometry-specific distance (Euclidean for EUCLID and Poincaré for HYPA/HYPT). In the patent subgraph, we evaluated $E^{(\ell)}$ on a fixed set of $|E_{\text{eval}}| = 200{,}000$ edges (sampled once and symmetrized) and reported the *last layer* value $E^{(L)}$. Unless otherwise stated, the results are aggregated into three training seeds (mean±std).

**Reporting conventions and endpoint metrics.** In Appendix D, we present multiple quantities for completeness. Unless otherwise stated, the *accuracy* values (val/test) are provided for reference only; *macro-F1* is the primary endpoint performance metric used and discussed in the main text, which is consistent with the class-imbalanced setting of the patent citation graph.

The per-seed results are summarized as the mean±std over three training seeds. All underlying per-seed probe values and endpoint metrics are included in the released CSV diagnostics (Appendix A.5), combined with scripts that deterministically regenerate every reported plot and table. We have omitted additional seed-wise tables to avoid redundancy because rounding in compact tables can obscure small but real differences.

## D.1 Operational definition of fixed-point-like plateaus

A block is called fixed-point-like only when consecutive representations are compared under a common native or shared observation map. For an aligned drift summary, a predeclared block of $k$ consecutive transitions may be labeled plateau-like when

$$D_{p50}(\ell) < \tau_D \qquad \text{for all } \ell \in \{\ell_0, \ldots, \ell_0 + k - 1\},$$

with $k = 3$ in the reported audit and $\tau_D$ chosen within an experiment family. The label is descriptive and does not imply a true attractor. No universal $\tau_D$ is transferred across geometries.

Layer-indexed common-space drift and width-changing transitions do not satisfy this alignment condition. They remain reproducible operational summaries, but a low value cannot by itself establish a plateau. The reported diagnoses therefore use full trajectories and joint probe couplings, and the $L = 32$ aligned contrast across EUCLID, HYPA, and HYPT is checked against Appendix E.7.

## D.2 Mean computation in hyperbolic space

For the hyperbolic models, the class prototypes were computed using a tangent-space mean at the origin of the Poincaré ball. Specifically, the embeddings were first mapped to the tangent space using the Riemannian logarithmic map $\log_0(\cdot)$, averaged in the Euclidean space, and mapped back using the exponential map $\exp_0(\cdot)$. This choice provides numerical stability and scalability while yielding behavior qualitatively similar to the Fréchet mean in our setting. Because this approximation is anchored at the origin, we do not present

it as an origin-invariant geometric summary; its role here is operational and comparative, with the same construction reused across depths and model variants within the protocol.

## D.3   Metric-aware Dirichlet energy

We employed the metric-aware Dirichlet energy to quantify the graph-local representation roughness. This formulation generalizes the classical Euclidean Dirichlet energy by replacing squared Euclidean distances with squared distances under the representation metric $d(\cdot, \cdot)$. Consequently, smoothness is measured in a geometry-consistent manner and is naturally connected to Laplacian-based analyses of oversmoothing when instantiated in the Euclidean space.

## D.4   Poincaré ball geometry and distance

We summarize the Poincaré ball geometry used throughout this study and fix the notation for the hyperbolic distance $d_{\mathrm{Hyp}}$ referenced in the main text.

**Poincaré ball.**   The $d$-dimensional Poincaré ball is

$$\mathbb{B}^d = \{u \in \mathbb{R}^d : \|u\|_2 < 1\},$$

equipped with a Riemannian metric with constant negative curvature. Unless otherwise stated, we used curvature $c = 1.0$ (unit ball).

**General curvature.**   Generally, for a curvature $c > 0$, the $d$-dimensional Poincaré ball is defined as

$$\mathbb{B}_c^d = \{u \in \mathbb{R}^d : \|u\|_2 < 1/\sqrt{c}\},$$

with a constant negative curvature $-c$. The corresponding geodesic distance is expressed by

$$d_{\mathbb{B}_c}(u, v) = \frac{1}{\sqrt{c}} \operatorname{arcosh}\left(1 + 2c \frac{\|u - v\|_2^2}{(1 - c\|u\|_2^2)(1 - c\|v\|_2^2)}\right). \tag{19}$$

Throughout the main text, we present expressions in unit-ball form ($c = 1$) for readability. All the hyperbolic probes were evaluated using the appropriate curvature for each model (e.g., $c = 1$ for HYPA and $c = 3$ for HYPT; see Table A.3). The boundary proximity is reported using the normalized radius $\sqrt{c}\,\|z\|$, such that the values approaching 1 consistently indicate saturation across the curvatures.

**Geodesic distance.**   For $u, v \in \mathbb{B}^d$, the geodesic distance is expressed in closed form as follows:

$$d_{\mathbb{B}}(u, v) = \operatorname{arcosh}\left(1 + 2\frac{\|u - v\|_2^2}{(1 - \|u\|_2^2)(1 - \|v\|_2^2)}\right). \tag{20}$$

Throughout the paper, we denote the Poincaré ball geodesic distance by

$$d_{\mathrm{Hyp}}(u, v) := d_{\mathbb{B}}(u, v). \tag{21}$$

**Exponential and logarithmic maps at the origin.**   Let $0 \in \mathbb{B}^d$ be the origin. Riemannian exponential and logarithmic maps at the origin are

$$\exp_0(v) = \tanh(\|v\|_2) \frac{v}{\|v\|_2}, \qquad \log_0(u) = \operatorname{artanh}(\|u\|_2) \frac{u}{\|u\|_2}, \tag{22}$$

with the conventions $\exp_0(0) = 0$ and $\log_0(0) = 0$.

**Relation to observed representations.**   In hyperbolic GNN models, internal activations are parameterized in the Euclidean tangent space and mapped to the Poincaré ball using an exponential map of the origin. As described in Section 4, we defined the observed representations $\mathbf{z}_i^{(\ell)} = \exp_0(\tilde{\mathbf{h}}_i^{(\ell)}) \in \mathbb{B}^d$ and evaluated all geometric probes, including drift, Dirichlet energy, separability, and boundary monitoring, on $\{\mathbf{z}_i^{(\ell)}\}$ using $d_{\mathrm{Hyp}}$.

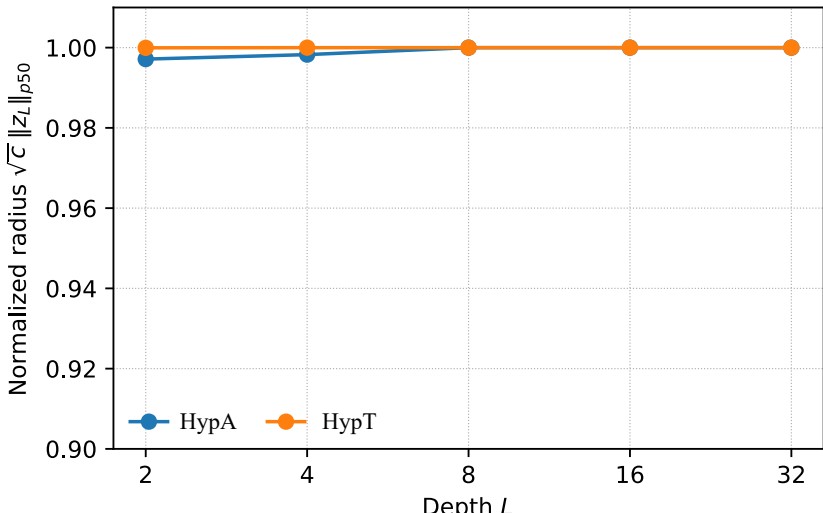

Figure D.1: Median normalized radius $r_{\mathrm{p}50}(L)$ at the final layer versus trained depth on the patent subgraph (representative run). Values near 1 indicate boundary saturation.

## D.5 Boundary saturation and metric amplification

**Boundary monitoring.** To monitor boundary effects, we track the curvature-normalized radius

$$r_i^{(\ell)} := \sqrt{c}\,\|\mathbf{z}_i^{(\ell)}\|_2 \in [0,1),$$

which provides a curvature-comparable notion of boundary proximity. We summarize it by the mean normalized radius

$$\bar{r}(\ell) := \frac{1}{|\mathcal{S}|}\sum_{i\in\mathcal{S}} r_i^{(\ell)},$$

and, where relevant, by upper-tail summaries such as

$$r_{\mathrm{p}90}(\ell) := \mathrm{p}90_{i\in\mathcal{S}}\, r_i^{(\ell)},$$

together with the boundary occupancy ratio from Section 4.4. Figure D.1 shows the median normalized radius

$$r_{\mathrm{p}50}(L) := \mathrm{p}50_{i\in\mathcal{S}}\, r_i^{(L)}$$

at the final layer as a function of trained depth for HYPA ($c = 1$) and HYPT ($c = 3$) on the patent subgraph (representative run). Values close to 1 indicate boundary saturation in a curvature-comparable way, and saturation is already visible at shallow depths for both hyperbolic variants.

**Local metric amplification.** The Poincaré ball model is endowed with a conformal Riemannian metric

$$g_{\mathbf{x}} = \lambda(\mathbf{x})^2 I, \qquad \lambda(\mathbf{x}) = \frac{2}{1 - \|\mathbf{x}\|_2^2},$$

diverging as $\|\mathbf{x}\|_2 \to 1$. For sufficiently small updates,

$$d_{\mathrm{Hyp}}(\mathbf{x}, \mathbf{x} + \Delta\mathbf{x}) \approx \lambda(\mathbf{x})\|\Delta\mathbf{x}\|_2.$$

Thus, a large measured drift can arise from metric amplification near the boundary, even when the intrinsic coordinate updates remain modest. The same mechanism amplifies the squared-distance quantities (e.g., metric-aware Dirichlet energy) by approximately $\lambda(\mathbf{x})^2$ in small increments.

Figure D.2: **Boundary proximity and deficit across trained depths on the same deterministic subset as Figure 2.** For HYPA, we summarize hyperbolic boundary statistics across trained depths for the same 2,000-node deterministic subset drawn from the fixed probe set $\mathcal{S}$ (top-10 classes; cap 200 nodes per class). Top: curvature-normalized radius summaries (mean, p50, and p90). Bottom: boundary-deficit summaries (mean, p50, and p90), where $\Delta_i^{(\ell)} = 1 - \sqrt{c} \, \|\mathbf{z}_i^{(\ell)}\|_2$. This figure preserves the original same-subset depth-sweep boundary summary corresponding to the geometry panels in Figure 2. For log-scale visualization of the boundary deficit, exact zeros are plotted at $\epsilon = 1.1 \times 10^{-5}$ and marked by open triangles.

### D.5.1 Layerwise coupling plot: boundary proximity versus drift/energy

To directly visualize how boundary proximity relates to the measured dynamics, we construct a layerwise coupling plot for the same HYPA configuration highlighted in Figure 2. For each layer $\ell$, we compute the upper-tail boundary-proximity proxy

$$r_{\mathrm{p90}}(\ell) := \mathrm{p90}_{i \in \mathcal{S}} \, \sqrt{c} \, \|\mathbf{z}_i^{(\ell)}\|_2$$

together with the interlayer drift $D(\ell)$ and Dirichlet energy $E(\ell)$ on the released deterministic phase-summary subset. Figure 2 shows that both drift and energy rise sharply as $r_{\mathrm{p90}}$ approaches 1 in this selected-subset view. Appendix E.7 shows that the complete-test-split native-space trajectory reaches boundary saturation earlier and then remains elevated. Together, these views support a boundary-pressure regime while leaving the precise onset layer observation-map- and subset-dependent.

**Empirical separation of coordinate steps and geodesic drift.** To disambiguate metric amplification from large coordinate changes, we present (i) Euclidean coordinate-step magnitude in the Poincaré ball and

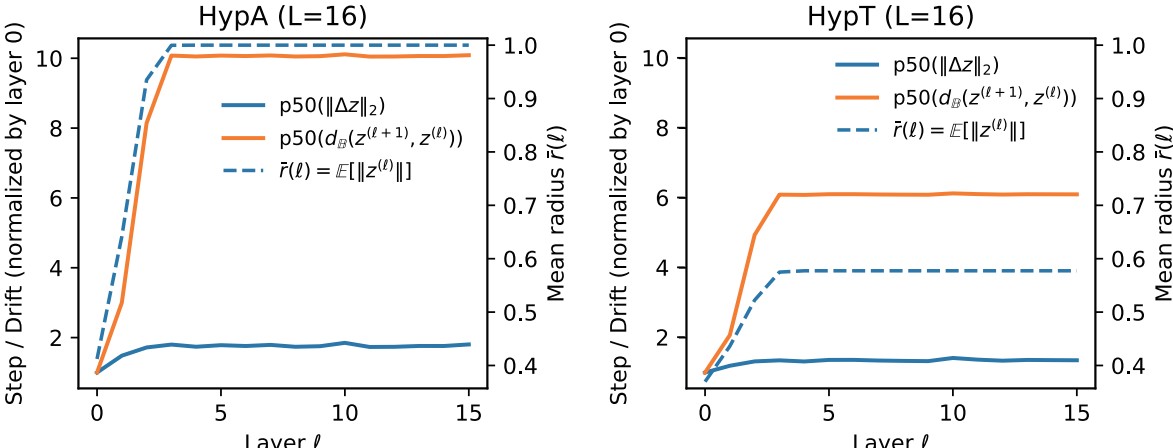

Figure D.3: Separating coordinate updates from metric-induced drift in hyperbolic GNNs. Layerwise comparison of (left) a representative hyperbolic baseline (HYPA) and (right) a minimally tuned hyperbolic control (HYPT) at depth $L = 16$ on the patent subgraph. For each layer $\ell$, we present the median Euclidean coordinate-step magnitude $p50(\|\Delta z\|_2)$ and the median geodesic drift $p50(d_{\mathbb{B}_c}(z^{(\ell+1)}, z^{(\ell)}))$, both normalized by their layer-0 values (left axis), together with the mean normalized radius $\bar{r}(\ell) = \mathbb{E}_{i \in \mathcal{S}}[\sqrt{c} \|\mathbf{z}_i^{(\ell)}\|_2]$ (right axis). In HYPA, geodesic drift increases sharply as embeddings approach the boundary ($\bar{r} \to 1$) despite modest coordinate-level updates. HYPT, introduced to discard trivial underoptimization, exhibits qualitatively similar boundary-associated drift amplification. Overall, these trends indicate that the observed boundary-associated amplification reflects metric amplification near the boundary rather than unusually large coordinate updates; they do not by themselves fix a universal onset layer.

(ii) the corresponding geodesic drift across layers, combined with the mean normalized radius (Figure D.3). We observed that the geodesic drift can grow sharply as the embeddings approach the boundary, even when coordinate-step magnitudes do not increase comparably, which is consistent with amplification by the local metric factor.

**Hyperbolic baselines and the role of HypT.** We consider two hyperbolic variants. HYPA is a representative off-the-shelf hyperbolic instantiation in a shared training protocol. HYPT is a minimally tuned hyperbolic control (tangent-space classifier head, input scaling $\alpha_{\text{in}}$, curvature $c$, and a learning-rate split between the encoder and head) introduced to ensure competitive shallow-depth performance. We included HYPA to reflect typical hyperbolic behavior and HYPT to argue against shallow-depth underoptimization as the primary explanation for the observed boundary-associated amplification.

**Curvature-normalized boundary deficit across $c$.** Raw Poincaré radii are not directly comparable across curvatures because the boundary radius depends on $c$: $\mathbb{B}_c^d = \{u \in \mathbb{R}^d : c\|u\|_2^2 < 1\}$ has boundary at $\|u\|_2 = 1/\sqrt{c}$. We therefore normalize radii by defining the curvature-normalized radius

$$\tilde{r}_i^{(\ell)} := \sqrt{c} \|\mathbf{z}_i^{(\ell)}\|_2 \in [0, 1),$$

and the corresponding *boundary deficit*

$$\Delta_i^{(\ell)} := 1 - \tilde{r}_i^{(\ell)}.$$

In this parameterization, $\Delta \approx 0$ indicates boundary saturation in a curvature-comparable way. Figure D.4 reports bulk deficit summaries across curvatures, showing that upper-tail proximity can saturate while bulk deficit remains informative for distinguishing depth regimes.

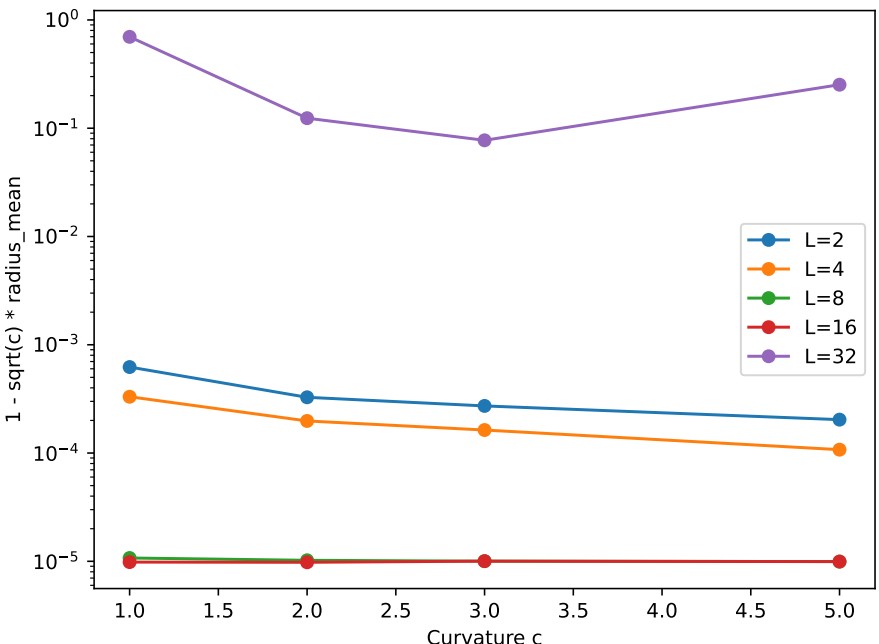

Figure D.4: **Curvature-normalized boundary deficit (bulk).** Mean boundary deficit $\Delta_{\mathrm{mean}} = 1 - \sqrt{c}\, r_{\mathrm{mean}}$ across curvatures $c$ for HYPT. A log scale highlights that bulk deficits can remain discriminative even when upper-tail proximity saturates, supporting a heterogeneous boundary-dominated regime at extreme depth.

**Heterogeneity of the extreme-depth regime.** To summarize distributional distortion beyond bulk averages, we compare the median and mean deficits:

$$\Delta_{\mathrm{p50}}^{(\ell)} := \mathrm{median}_{i \in S}\, \Delta_i^{(\ell)}, \qquad \Delta_{\mathrm{mean}}^{(\ell)} := \frac{1}{|S|} \sum_{i \in S} \Delta_i^{(\ell)},$$

and define a simple heterogeneity proxy

$$H_\Delta^{(\ell)} := \Delta_{\mathrm{p50}}^{(\ell)} - \Delta_{\mathrm{mean}}^{(\ell)}.$$

The corresponding export artifact labels this same summary as `gap_deficit_p50_mean`; throughout the paper we refer to it uniformly as $H_\Delta$. Figure D.5 shows that $H_\Delta$ stays near zero for shallower depths but becomes qualitatively distinct at very large depth ($L$=32), consistent with a boundary-dominated regime whose distributional shape differs from shallower settings.

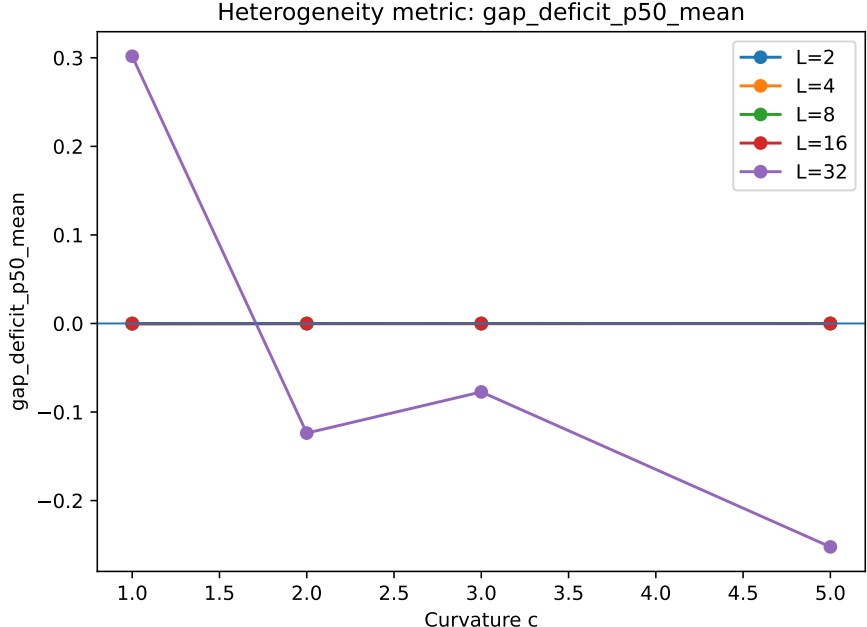

Figure D.5: **Boundary-deficit heterogeneity across curvatures.** We plot $H_\Delta = \Delta_{\mathrm{p50}} - \Delta_{\mathrm{mean}}$ across curvatures for HYPT (the exported figure file labels the same quantity as `gap_deficit_p50_mean`). Shallower depths remain close to zero under this summary, whereas $L=32$ is clearly separated, indicating a qualitatively distinct very-deep regime with distributional distortion (tail saturation but nontrivial bulk).

### D.6 Separating geodesic amplification from ambient-coordinate change

A central concern in hyperbolic representation analysis is whether large measured interlayer drift reflects a genuine representational change or amplification induced by the Poincaré metric near the boundary. To disambiguate these effects, we explicitly compared multiple notions of drift computed for the same learned representation.

**Euclidean-coordinate drift.** First, we computed the coordinate-level drift by measuring the Euclidean displacement. $\|\Delta z^{(\ell)}\|_2 = \|z^{(\ell+1)} - z^{(\ell)}\|_2$ in ambient space. This quantity reflects the intrinsic update magnitude, which is independent of hyperbolic metrics.

**Geometry-aware (hyperbolic) drift.** We then computed the corresponding geodesic drift using the Poincaré distance $d_{\mathrm{Hyp}}(z^{(\ell+1)}, z^{(\ell)})$, which is a geometrically consistent concept in the main text.

**Comparison and interpretation.** As shown in Figure D.3, geodesic drift can grow sharply as representations approach the boundary even when Euclidean-coordinate updates remain modest. This separation indicates that the observed boundary-associated drift amplification in hyperbolic models cannot be solely attributed to remarkably large coordinate updates. Instead, it reflects the metric amplification induced by boundary proximity.

Therefore, we interpret boundary-associated drift amplification as a *geometry-driven amplification regime* rather than a pure measurement artifact. In addition, we emphasize that coordinate-level drift provides a complementary diagnostic and that geometry-aware and coordinate-level probes should be interpreted jointly, rather than in isolation.

### D.7 Seed sensitivity and determinism

All the reported mean±std values were computed for the three training seeds. Given a fixed checkpoint and released probe/evaluation subsets (Appendix A.1), the probe computation is deterministic. For transparency and precise reproduction, the reproducibility bundle released full-per-seed CSV diagnostics for the reported depth, geometry, and activation settings (Appendix A.5), together with scripts and manifests for the principal quantitative figures and tables.

### D.8 Architectural scope

The main mechanism study focuses on GraphSAGE-style message passing, with Euclidean GCN checks reported on the patent graph and OGBN-Arxiv. The probe definitions are not tied to GraphSAGE-specific internal operations, but the current experiments do not establish architecture-universal regime boundaries. Extending the analysis to attention-based or spectral/diffusion GNNs remains an important next step and may reveal further architecture-dependent variations in regime structure.

### D.9 Probe as diagnostic tool

We emphasize that fixed-point probing is a diagnostic instrument rather than a training objective. The incorporation of probe-derived signals as regularizers remains an interesting direction for future research.

# E   Robustness, sensitivity, and scope checks

This appendix collects the additional checks that make protocol coverage, uncertainty, robustness, runtime cost, and scope explicit. Appendix E.1 distinguishes complete current-family instantiations from public-convention component checks; the remaining subsections report seed uncertainty, budget sensitivity, runtime, intervention interpretation, hidden-width sensitivity, and observation-map sensitivity. Unless stated otherwise, values are reported as mean±std over the seeds or repeats identified in the corresponding caption.

## E.1   Cross-dataset evidence and scope of claims

This section states the empirical coverage directly. The complete current geometry-aware probe family is reported on the patent GraphSAGE study and on the three-seed OGBN-Arxiv GraphSAGE sweep. Cora and Texas provide replicated Euclidean components under the released public-benchmark convention; their separability values are not relabeled as the current global Eq. (15) statistic. The evidence supports successful protocol instantiation and dataset-dependent diagnostic readings, while the detailed patent boundary-coupled mechanism remains a bounded stress-test finding.

Table E.1: **Cross-dataset evidence map.** The table lists exact probe coverage, replication convention, direct evidence, and the conclusion supported by each dataset/backbone.

| Dataset | Backbone / geometry | Probes reported | Seeds / splits | Direct evidence | Supported conclusion |
|---|---|---|---|---|---|
| Patent citation | GraphSAGE; EUCLID, HYPA, HYPT | Current drift, energy, global Eq. (15) separability, normalized radius / boundary statistics; aligned $L = 32$ hidden-transition audit for all three modes | Three training seeds; fixed temporal split; fixed $\mathcal{S}$ and $\mathcal{E}_{eval}$ | Figs. 1–3; Tables 2, 3, and E.10; Appendix Figs. B.2, B.4, and B.5 | Bounded mechanism case study: aligned late Euclidean contraction contrasts with boundary-saturated hyperbolic drift plateaus at $L = 32$; onset timing is not treated as observation-map invariant. |
| Patent citation | Euclidean GCN | Drift-$p90$, energy-$p90$, endpoint macro-F1 | Three training seeds; same temporal split and fixed subsets | Fig. 5 | The workflow is not tied to GraphSAGE; degeneration placement remains backbone-dependent. |
| OGBN-Arxiv | GraphSAGE; EUCLID and OGBN-specific HYPT | Complete current family: drift, energy, global Eq. (15) separability, normalized-radius onset/persistence, endpoint macro-F1 | Three training seeds at all five depths; complete fixed test split; fixed evaluation edges | Fig. 6; Table 4; Fig. C.13 | Complete public-benchmark instantiation. Boundary saturation alone does not identify failure, and the reproducible $L = 32$ training-and-selection failure exhibits state–outcome dissociation across validation-selected checkpoints. |
| Cora | Euclidean public checks; ReLU and identity variants | Drift, public-convention separability, endpoint accuracy / macro-F1, and metric-aware energy | Five training seeds on one fixed public split; median $[\min, \max]$ or min–max bands | Figs. C.6, C.8–C.12; Tables C.5–C.6 | Replicated Euclidean components expose nonmonotonic depth trajectories and probe–performance decoupling. |
| Texas | Euclidean public checks; ReLU and identity variants | Drift, public-convention separability, endpoint accuracy / macro-F1 | Five training seeds across ten official splits (50 seed–split runs) | Figs. C.7, C.8–C.12; Tables C.7–C.8 | The Euclidean components remain interpretable under substantial seed–split replication on a heterophilous graph. |
| OGBN-Arxiv | GCN reference and HYPT interventions | Coordinate-level activation norm, energy, endpoint F1, normalized-radius intervention summaries | GCN seed-0 reference; intervention pilots and cross-seed/repeat checks | Figs. C.1–C.5; Tables C.1–C.4 | Architecture and control references; no smooth or performance-preserving intervention threshold is inferred. |

Together, these rows establish complete current-family coverage on two datasets and replicated Euclidean components on two additional public datasets. They support protocol instantiation and dataset-dependent state diagnosis, not transfer of an absolute probe scale from one geometry or dataset to another.

Table E.2: **Coverage conventions for the reported evidence.**

| Evidence class | Reporting convention |
|---|---|
| Complete current geometry-aware family | Patent GraphSAGE and OGBN-Arxiv GraphSAGE use the current drift, metric-aware energy, global Eq. (15) separability, and curvature-normalized boundary definitions. |
| Replicated Euclidean public components | Cora/Texas use the released public-benchmark drift/separability convention; Cora additionally reports energy. These values are labeled as components rather than relabeled Eq. (15). |
| Coordinate-level architecture reference | The OGBN-Arxiv GCN norm panel measures projected activation coordinates before any hyperbolic exponential map and is not a Poincaré-radius panel. |
| Mechanism and intervention scope | The patent boundary-coupled interpretation is the bounded mechanism study; OGBN intervention responses remain timing-sensitive and seed-fragile. |

## E.2 Seed-aggregated uncertainty for the canonical test probe

Several mechanism figures in the main text are intentionally shown as representative layerwise traces. To make uncertainty explicit, we report seed-aggregated counterparts for the canonical patent-graph summaries. The node-level quantities in Table E.3 use the canonical *test-restricted fixed probe*, which at the 50k budget equals the complete test split and is reused for every checkpoint; the edge quantity uses the same fixed symmetrized evaluation-edge set throughout. Drift is the final-transition median $D_{\mathrm{p50}}(L-1)$, energy is the last-layer mean, separability is the global Eq. (15) scalar at the final layer, and the boundary column is the curvature-normalized radius $r_{\mathrm{p90}} = \mathrm{p90}_i \sqrt{c}\|z_i\|_2$ from Section 4.4. Values are mean±std over three training seeds. The drift, energy, and Eq. (15) separability entries are generated from the same canonical analysis master used by Table 3; duplicating the representative $L \in \{16, 32\}$ rows here makes the uncertainty audit explicit.

Table E.3: Seed-aggregated canonical test-split summaries at representative depths. The final-transition drift, last-layer energy, and Eq. (15) separability entries are identical to the corresponding rows in Table 3. The drift column is the operational common-probe-space endpoint statistic; Appendix E.7 separately audits width-preserving native hidden-space dynamics. Hyperbolic $r_{\mathrm{p90}}$ is curvature-normalized, so boundary saturation is near one for both $c = 1$ and $c = 3$.

| mode | $L$ | final drift p50 | final energy mean | Eq. (15) Sep | normalized $r_{p90}$ | $n$ |
|---|---|---|---|---|---|---|
| EUCLID | 16 | $5.761 \pm 0.634$ | $52.783 \pm 8.023$ | $0.003 \pm 0.000$ | $-$ | 3 |
| EUCLID | 32 | $4.016 \pm 1.123$ | $23.812 \pm 7.751$ | $0.004 \pm 0.001$ | $-$ | 3 |
| HYPA | 16 | $20.474 \pm 0.036$ | $261.480 \pm 12.924$ | $0.002 \pm 0.000$ | $1.000 \pm 0.000$ | 3 |
| HYPA | 32 | $20.536 \pm 0.090$ | $245.196 \pm 32.467$ | $0.002 \pm 0.000$ | $1.000 \pm 0.000$ | 3 |
| HYPT | 16 | $11.831 \pm 0.050$ | $122.766 \pm 3.593$ | $0.003 \pm 0.000$ | $1.000 \pm 0.000$ | 3 |
| HYPT | 32 | $10.189 \pm 2.724$ | $69.396 \pm 57.506$ | $0.002 \pm 0.001$ | $1.000 \pm 0.000$ | 3 |

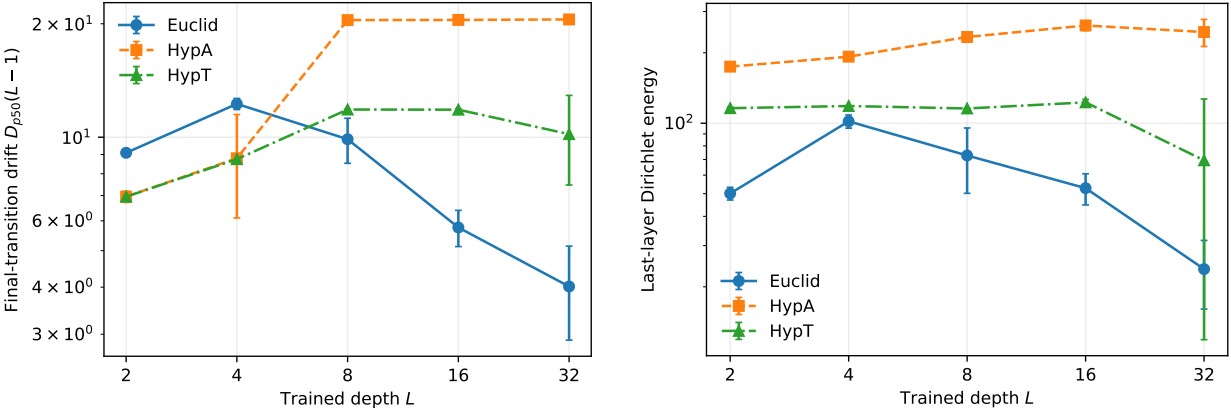

Figure E.1: Seed-aggregated canonical test-split summaries with standard-deviation bars. Left: final-transition drift p50 versus trained depth. Right: last-layer Dirichlet energy mean versus trained depth. These panels use the same summary convention as Table 3.

### E.3 Probe-node and evaluation-edge budget sensitivity

The canonical table in Appendix E.2 is evaluated on the fixed test probe. The budget study below is a separate induced-subgraph probe-budget experiment. For each node budget $|\mathcal{S}| \in \{1k, 5k, 10k, 50k\}$ and training checkpoint, the released canonical probe implementation was run with three independently initialized run seeds. Under `seed_source=subset`, that implementation samples the requested number of probe nodes without replacement from the full induced-subgraph node pool. The same run seed also initializes the stochastic evaluation replay. Thus, the repeat analysis measures combined probe-selection/evaluation-seed variability rather than node-ID-only resampling within a fixed validation split. In particular, the $50k$ runs are different $50k$-node samples from the 479,533-node induced subgraph; they are neither the complete validation split nor a second estimate of the canonical test-split table.

For the node budget, final-transition drift and the qualitative ordering were stable across the tested sizes; increasing $|\mathcal{S}|$ mainly reduced run-seed variability. For the edge budget $|\mathcal{E}_{\mathrm{eval}}| \in \{10k, 50k, 100k, 200k\}$, the 10k-edge estimates were already close to the 200k default, with relative changes below 0.3% in every tested mode/depth condition. The node-budget table focuses on drift because it is the direct nodewise quantity used for budget selection; the Eq. (15) separability values in the final manuscript come from the canonical analysis pass, not from the earlier batchwise budget sweep.

Table E.4: Induced-subgraph probe-node budget sensitivity. For each training checkpoint and budget, three independently seeded probe runs were performed. In the released canonical implementation, each run samples nodes without replacement from the full induced-subgraph node pool, and the same run seed initializes the evaluation replay. Entries show mean±std over the corresponding training and run seeds. The last column is the relative change in final-transition drift p50 from $|\mathcal{S}| = 1k$ to $50k$. These rows are a stability experiment and are not a second estimate of the canonical test-split values in Table E.3.

| mode | $L$ | $|\mathcal{S}| = 1k$ drift | $|\mathcal{S}| = 50k$ drift | $\Delta$ drift (%) |
|---|---|---|---|---|
| EUCLID | 16 | $6.970 \pm 0.417$ | $6.901 \pm 0.410$ | -0.99 |
| EUCLID | 32 | $4.961 \pm 0.424$ | $4.890 \pm 0.369$ | -1.43 |
| HYPA | 16 | $20.480 \pm 0.016$ | $20.477 \pm 0.018$ | -0.01 |
| HYPA | 32 | $20.536 \pm 0.078$ | $20.536 \pm 0.078$ | 0.00 |
| HYPT | 16 | $11.839 \pm 0.041$ | $11.838 \pm 0.039$ | -0.01 |
| HYPT | 32 | $7.841 \pm 5.881$ | $7.841 \pm 5.881$ | 0.00 |

Table E.5: Within-checkpoint variability across independently seeded probe runs for final-transition drift p50. For each training checkpoint and budget, we first computed the sample standard deviation across three run seeds and then summarized those within-checkpoint standard deviations across the 18 checkpoints. Because the canonical probe implementation uses the same run seed for probe-node selection and evaluation-time sampling, this is combined probe/evaluation-seed variability rather than node-ID-only resampling variability.

| $|\mathcal{S}|$ | mean run-seed std | std across ckpts | $n$ |
|---|---|---|---|
| 1,000 | 0.0248 | 0.0478 | 18 |
| 5,000 | 0.0112 | 0.0213 | 18 |
| 10,000 | 0.0079 | 0.0132 | 18 |
| 50,000 | 0.0030 | 0.0053 | 18 |

**Practical recommendation.** Practitioners can start with a small pilot budget, e.g., $|\mathcal{S}| = 1k$ and $|\mathcal{E}_{\mathrm{eval}}| = 10k$, repeat the probe with a few independently initialized run seeds, and increase the budget until the qualitative regime label and the rank/order of depthwise summaries are stable. Once selected, $\mathcal{S}$ and $\mathcal{E}_{\mathrm{eval}}$ should be fixed and reported for all comparisons.

Table E.6: Evaluation-edge budget sensitivity for last-layer Dirichlet energy. The edge-budget experiment keeps the node-side checkpoint and representation fixed and resamples only the evaluation-edge set. The 200k default is conservative: the 10k and 200k estimates differ by less than 0.3% in every condition.

| mode | $L$ | $|\mathcal{E}_{\text{eval}}| = 10k$ energy | $|\mathcal{E}_{\text{eval}}| = 200k$ energy | $\Delta$ energy (%) |
|---|---|---|---|---|
| EUCLID | 16 | $52.817 \pm 6.878$ | $52.781 \pm 6.948$ | -0.07 |
| EUCLID | 32 | $23.843 \pm 6.734$ | $23.813 \pm 6.714$ | -0.13 |
| HYPA | 16 | $261.834 \pm 11.224$ | $261.477 \pm 11.195$ | -0.14 |
| HYPA | 32 | $245.888 \pm 28.217$ | $245.194 \pm 28.123$ | -0.28 |
| HYPT | 16 | $122.753 \pm 3.110$ | $122.768 \pm 3.110$ | 0.01 |
| HYPT | 32 | $69.243 \pm 49.719$ | $69.396 \pm 49.802$ | 0.22 |

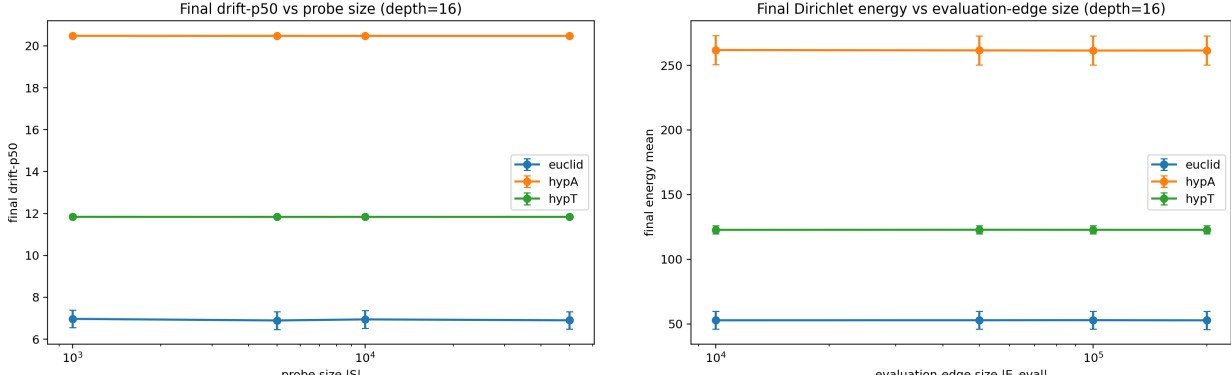

Figure E.2: Budget sensitivity at $L = 16$. Left: final-transition drift p50 versus probe-node budget for independently seeded induced-subgraph probe runs. Right: last-layer Dirichlet energy mean versus evaluation-edge budget. Error bars show standard deviations over the corresponding training-seed and repeat-seed variability.

### E.4    Wall-clock runtime and monitoring cost

The probes require no backpropagation and operate on fixed subsets. With the default budgets $|\mathcal{S}| = 50k$ and $|\mathcal{E}_{\text{eval}}| = 200k$, the full diagnostic takes approximately 17–24 seconds per checkpoint in our environment (Table E.7). The small pilot budget takes approximately 9–10 seconds per checkpoint. In practice, the full default probe is most naturally run at validation intervals or saved checkpoints, while the smaller pilot budgets are cheap enough for lightweight training-time monitoring; we do not recommend running the full default probe after every mini-batch.

Table E.7: Default per-checkpoint runtime in seconds at $|\mathcal{S}| = 50k$ and $|\mathcal{E}_{\text{eval}}| = 200k$. The total combines the probe pass and the Dirichlet-energy pass by quadrature for the displayed standard deviation.

| mode | L | probe time | energy time | total time |
|---|---|---|---|---|
| EUCLID | 16 | $6.795 \pm 0.253$ | $9.929 \pm 0.167$ | $16.724 \pm 0.303$ |
| EUCLID | 32 | $9.178 \pm 0.186$ | $14.444 \pm 0.131$ | $23.622 \pm 0.228$ |
| HYPA | 16 | $6.897 \pm 0.157$ | $9.693 \pm 0.170$ | $16.590 \pm 0.231$ |
| HYPA | 32 | $9.283 \pm 0.250$ | $14.417 \pm 0.340$ | $23.700 \pm 0.422$ |
| HYPT | 16 | $6.830 \pm 0.229$ | $9.706 \pm 0.190$ | $16.536 \pm 0.298$ |
| HYPT | 32 | $9.276 \pm 0.123$ | $14.719 \pm 1.444$ | $23.995 \pm 1.449$ |

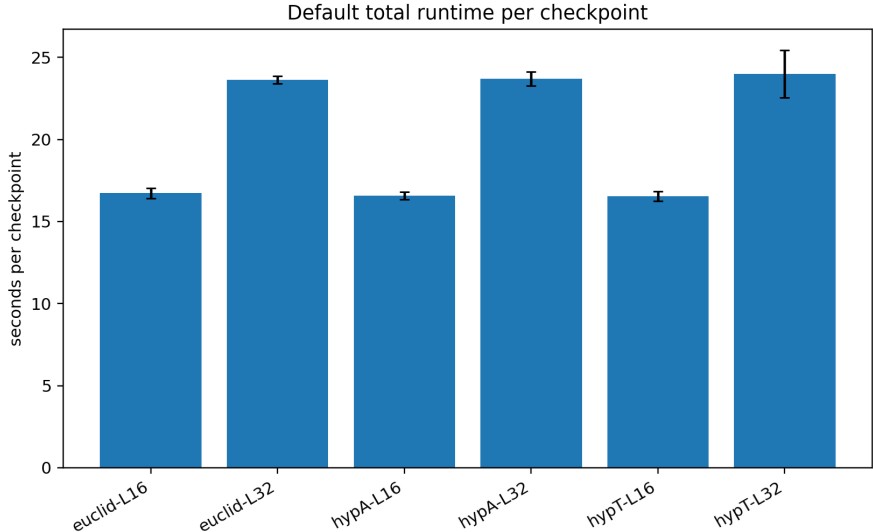

Figure E.3: Default total runtime per checkpoint at $|\mathcal{S}| = 50k$ and $|\mathcal{E}_{\text{eval}}| = 200k$.

### E.5    Boundary-intervention interpretation for Figure 3

We make the schedule explicit: $\lambda = 10$ is a late-only/delayed intervention, whereas $\lambda = 100$ is applied globally from epoch 0. We also avoid interpreting the intervention as a smooth or performance-preserving control law. A terminal-window average over layers 24–32 can be misleading because the $\lambda = 0$ trace contains a terminal drop/collapse artifact; therefore, Table E.8 reports the primary numeric summary over the pre-terminal saturated window, layers 5–28. The interpretation should also account for the different metric sensitivity of the two probes: local geodesic drift scales approximately as $\lambda(x)\|\Delta x\|_2$, while the squared distances in metric-aware Dirichlet energy are approximately $\lambda(x)^2$-sensitive. Thus a radius perturbation can reduce the high-energy portion of the trajectory more clearly than it reduces median drift, whose layerwise value is also affected by direction and timing.

Table E.8: Pre-terminal saturated-window summary for the boundary-intervention diagnostic at $L = 32$ for HYPA. The global $\lambda = 100$ condition reduces the high-energy portion of the trajectory in this window, whereas the drift response is not a clean monotonic attenuation. This asymmetry is expected because Dirichlet energy is a squared-distance statistic, while drift is a first-order displacement statistic; the intervention is therefore interpreted as a diagnostic perturbation rather than as a remedy or control law.

| condition | schedule | window | $r_{p90}$ | drift p50 | energy mean | energy max | use |
|---|---|---|---|---|---|---|---|
| $\lambda = 0$, no radius control | none | 5–28 | $1.000 \pm 0.000$ | $20.149 \pm 1.699$ | $311.134 \pm 28.291$ | 358.352 | primary |
| $\lambda = 10$, late-only control | late-only / delayed | 5–28 | $1.000 \pm 0.000$ | $20.491 \pm 0.138$ | $297.545 \pm 37.358$ | 350.118 | primary |
| $\lambda = 100$, global from epoch 0 | global / always-on from epoch 0 | 5–28 | $1.000 \pm 0.000$ | $19.943 \pm 2.296$ | $257.816 \pm 31.666$ | 305.717 | primary |

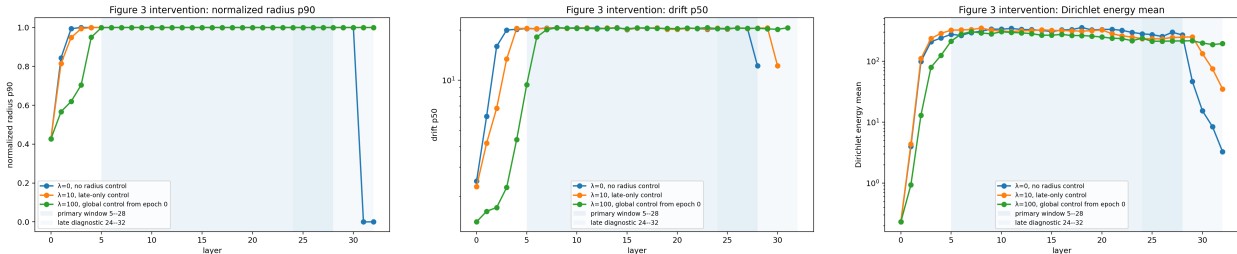

Figure E.4: Boundary-intervention diagnostics with the primary pre-terminal window shaded. Left: upper-tail normalized radius. Middle: drift p50. Right: Dirichlet energy mean. The weak late-only intervention remains boundary-saturated, while the always-on $\lambda = 100$ condition reduces the high-energy portion of the saturated trajectory.

The revised interpretation is consequently cautious: late-only control remains boundary-saturated and does not materially suppress the amplification. Strong global control from epoch 0 changes the saturated trajectory and reduces the high-energy portion of the curve, but the response is timing-sensitive and nonmonotonic, especially for drift. This difference is compatible with the conformal-factor asymmetry above: the energy probe is a squared-distance statistic and is expected to respond more sharply to changes in boundary pressure than the median drift probe. We therefore treat this intervention as diagnostic evidence consistent with a boundary contribution, not as a robust radius-control law or a performance-preserving regularizer.

## E.6 Hidden-width sensitivity

To assess encoder-capacity sensitivity, we vary the GraphSAGE hidden width in a compact sweep over $d \in \{64, 128, 256\}$ at $L \in \{16, 32\}$ for all three geometries. This sweep varies the encoder hidden width while keeping both the raw input text-feature dimension and the output representation dimension fixed (output dimension 64). Thus, it tests sensitivity to encoder capacity rather than a change to the input text embedding or the final output representation dimension. Across widths, the absolute scale of the probes and endpoint metrics varies, but changing hidden width does not remove the severe $L = 32$ degradation regime. We report endpoint macro-F1, final-transition drift, and last-layer energy here. The separability column from the earlier analysis is omitted because it used a batchwise proxy rather than the global Eq. (15) computation.

Table E.9: Hidden-width sensitivity sweep. The swept dimension is the GraphSAGE encoder hidden width; the raw input text-feature dimension and output representation dimension are fixed, with output dimension 64. Values are mean±std over three training seeds. Drift uses the final transition and energy the last layer.

| $d$ | mode | $L$ | test macro-F1 | final drift p50 | final energy mean |
|---|---|---|---|---|---|
| 64 | Euclid | 16 | $0.008 \pm 0.003$ | $5.173 \pm 0.424$ | $26.040 \pm 3.564$ |
| 64 | Euclid | 32 | $0.001 \pm 0.001$ | $2.701 \pm 1.730$ | $34.189 \pm 12.966$ |
| 64 | HypA | 16 | $0.007 \pm 0.003$ | $20.556 \pm 0.006$ | $288.151 \pm 2.227$ |
| 64 | HypA | 32 | $0.001 \pm 0.000$ | $6.825 \pm 11.822$ | $97.735 \pm 134.797$ |
| 64 | HypT | 16 | $0.015 \pm 0.005$ | $11.840 \pm 0.015$ | $118.099 \pm 0.380$ |
| 64 | HypT | 32 | $0.001 \pm 0.000$ | $11.834 \pm 0.094$ | $78.575 \pm 35.980$ |
| 128 | Euclid | 16 | $0.013 \pm 0.003$ | $6.721 \pm 0.821$ | $51.071 \pm 15.287$ |
| 128 | Euclid | 32 | $0.001 \pm 0.000$ | $4.564 \pm 1.926$ | $16.876 \pm 13.097$ |
| 128 | HypA | 16 | $0.014 \pm 0.004$ | $20.482 \pm 0.049$ | $264.856 \pm 12.673$ |
| 128 | HypA | 32 | $0.001 \pm 0.000$ | $20.519 \pm 0.118$ | $181.991 \pm 41.026$ |
| 128 | HypT | 16 | $0.023 \pm 0.005$ | $11.816 \pm 0.023$ | $122.181 \pm 2.183$ |
| 128 | HypT | 32 | $0.001 \pm 0.000$ | $10.232 \pm 2.768$ | $62.976 \pm 47.709$ |
| 256 | Euclid | 16 | $0.012 \pm 0.008$ | $7.779 \pm 0.700$ | $71.797 \pm 7.830$ |
| 256 | Euclid | 32 | $0.001 \pm 0.000$ | $3.255 \pm 1.590$ | $50.598 \pm 19.797$ |
| 256 | HypA | 16 | $0.020 \pm 0.001$ | $20.480 \pm 0.070$ | $255.395 \pm 7.582$ |
| 256 | HypA | 32 | $0.001 \pm 0.000$ | $13.601 \pm 11.780$ | $98.472 \pm 75.789$ |
| 256 | HypT | 16 | $0.030 \pm 0.009$ | $11.813 \pm 0.047$ | $122.427 \pm 1.730$ |
| 256 | HypT | 32 | $0.001 \pm 0.000$ | $10.195 \pm 2.729$ | $47.466 \pm 33.475$ |

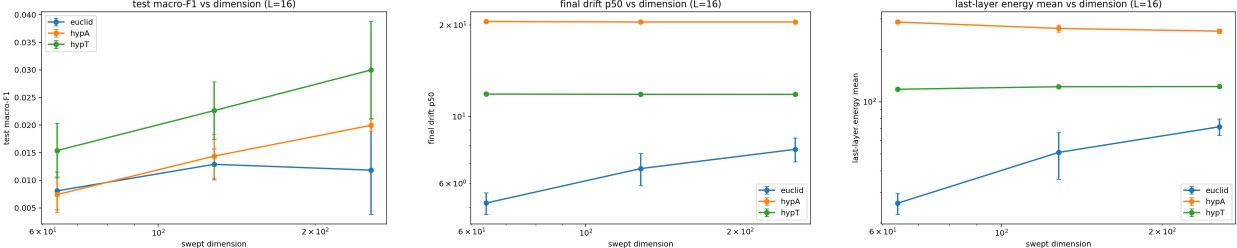

Figure E.5: Hidden-width sensitivity at $L = 16$. Changing hidden width affects endpoint metrics and absolute probe scales, but does not remove the qualitative geometry-dependent differences.

### E.7 Observation-map alignment sensitivity for the $L = 32$ patent trajectory

The full-sweep common-probe representation uses layer-indexed projectors, so consecutive hidden-layer drift can include observation-map change. This audit replays the saved patent $L = 32$ Euclid, HypA, and HypT checkpoints, restricted to width-preserving hidden-to-hidden transitions. It does not cover shallower depths, separability, width-changing interfaces, or the GCN backbone and therefore does not replace the full-sweep reporting convention.

The primary replay removes layer-indexed projection by using each model's native 128-dimensional hidden parameterization on the complete fixed test split. The Euclidean observation is the identity; the hyperbolic observations retain the exponential map and Poincaré metric defined in Section 4. A sensitivity replay uses one shared $128 \rightarrow 64$ projector across all hidden layers before the same model-specific observation map. Energy uses the same fixed 200k symmetrized edge set. Middle and late source-layer windows are 8–15 and 24–30. The audit spans three training seeds, ten shared-projector seeds, one class-balanced subset, and five random subsets.

Table E.10: Observation-map sensitivity for the very-deep patent trajectories. The primary replay uses the native-width hidden parameterization with the model-specific observation map and metric. Drift and saturation statistics use the complete fixed test-node support, whereas energy uses the same fixed symmetrized 200k-edge support sampled from the full induced graph. Ratios are late-window median divided by middle-window median and are summarized as mean±std over three training seeds. For drift, ratios near one indicate an elevated plateau; ratios far below one in both drift and fixed-edge energy indicate late contraction. Shared-projector error is the relative error in the drift ratio across ten fixed projector seeds (mean / maximum). The subset range covers the class-balanced and five random node subsets across the three training seeds; energy remains the same fixed global edge trajectory in the subset check. The first saturation layer is reported only for hyperbolic modes and is the first source layer whose target-layer normalized upper-tail radius satisfies $r_{p90} \geq \tau_r = 0.95$.

| mode | drift late/mid | energy late/mid | first saturation layer | shared-proj. error | subset drift range |
|------|----------------|-----------------|------------------------|--------------------|--------------------|
| Euclid | $0.085 \pm 0.116$ | $(1.48 \pm 2.11) \times 10^{-6}$ | – | 3.5%/8.2% | 0.011–0.330 |
| HypA | $0.977 \pm 0.005$ | $0.870 \pm 0.128$ | $3.3 \pm 1.5$ | 0.5%/1.9% | 0.972–0.982 |
| HypT | $0.992 \pm 0.008$ | $0.973 \pm 0.273$ | $1.7 \pm 0.6$ | 0.8%/2.1% | 0.986–1.001 |

The release also records an auxiliary pre-specified *late co-amplification* screen. It is not the regime classifier in Protocol 1; it asks only whether drift and energy begin a new joint increase within the designated late window. A false value is expected for Euclidean contraction and for a hyperbolic plateau entered before that window. The paper-facing label is instead based on the aligned late/middle ratios together with consistency across training seeds, shared projectors, and node subsets; no new absolute cross-dataset threshold is introduced. Because the width-changing hidden-to-output interface is excluded, the aligned late hidden-layer contraction is compatible with the terminal rebound visible in the operational output-layer dispersion and final-transition summaries.

The aligned measurements support two distinct tested $L = 32$ hidden-to-hidden regimes. For Euclid, the complete-test native drift late/middle ratios are 0.015, 0.020, and 0.219 across the three training seeds, and the corresponding fixed-edge energy ratios are $4.8 \times 10^{-8}$, $5.0 \times 10^{-7}$, and $3.9 \times 10^{-6}$; late contraction is present in every tested full-split seed. Across ten shared-projector seeds per checkpoint, the Euclidean drift-ratio relative error averages 3.5% and is at most 8.2%. Every reported Euclidean node-subset ratio remains below one (range 0.011–0.330), although the class-balanced subset weakens the contraction. For HypA and HypT, saturation occurs early and the combined full-split and subset drift ratios lie in $[0.972, 1.001]$; shared-projector errors average 0.5% and 0.8% and are at most 1.9% and 2.1%, respectively. Across all reported full-split and node-subset checks, the Euclidean range $[0.011, 0.330]$ and hyperbolic range $[0.972, 1.001]$ are disjoint. Hyperbolic fixed-edge energy is reported for completeness rather than as load-bearing evidence for the middle-to-late plateau contrast. The HypT native full-split energy ratios are 1.27, 0.73, and 0.92, so the direction of the middle-to-late change varies across seeds, whereas its drift ratios remain 0.986–1.001. The plateau conclusion therefore rests on drift persistence and boundary saturation; the broader boundary-associated

interpretation additionally uses the ambient-coordinate check. Energy remains a descriptive component of the reported evidence vector. Thus the audit supports late Euclidean contraction versus elevated hyperbolic drift plateaus at $L = 32$; it does not validate shallower-depth trajectories, width-changing transitions, separability, or an invariant onset layer.

