# OpenReview forum: "Fixed-Point Probing for GNN Depth Diagnostics: A Geometry-Consistent Protocol with a Patent-Citation Case Study"
_TMLR — Under review for TMLR_

### Review · Reviewer_Qyzb · 2026-06-01

**Summary Of Contributions:**

It is widely known that the performance of graph neural networks often degrade with increasing depth. Prior works usually provide only an endpoint metric or a single probe to analyze the performance degradation but this cannot reveal the exact cause of the degradation. This work introduces fixed-point probing which reports multiple metrics on a fixed probe-node and evaluation-edge sets to provide a detailed analysis on the GNN training dynamics. The authors test the proposed protocol on a patent-citation dataset across different aspects to demonstrate the effectiveness of the protocol.

Strengths
- The experiments are extensive. The protocol is tested across different models (euclidean vs hyperbolic) and depths.
- The authors provide an extensive empirical analysis in Section 6.
- The authors promise to release a public reproducibility archive, which can be helpful for practitioners and researchers in this area.

Weakness
- The authors only focus on a patent-citation dataset, which questions the universality of the proposed protocol across different datasets.
- The robustness of the proposed protocol to the randomness of the probe-node sampling is unclear.

**Additional Comments:**

None

**Audience:**

Yes

**Audience Explanation:**

The researchers across different areas, especially in graph neural networks, would be interested in the findings of the paper.
Specifically, the authors present a new protocol to analyze the training dynamics of GNN which can be helpful for researchers in this area.
Also, the authors promise to release a public reproducibility archive.

**Broader Impact Concerns:**

No.

**Claims And Evidence:**

No

**Claims Explanation:**

First, the authors argue that their experiments on the patent-citation graph is just a stress-test case study and do not serve as an evidence for the universality across different datasets. However, if a new protocol is to be proposed, then the authors should clearly demonstrate its universality across different datasets. Otherwise, it is not very clear if the proposed protocol only works for a specific dataset or can be successfully applied to others.

Next, it is not very clear if the proposed protocol is robust to the randomness of the probe-node sampling since many figures are missing the standard deviations. Also, in Figure 4 for L=32, the standard deviation is so large that one interval even largely overlaps with the others. Thus, it is unclear if the proposed diagnostics can provide a robust diagnostic for deep networks.

**Requested Changes:**

(major) Some figures are missing standard deviation. Please add standard deviation in the graphs.

(major) How does the number of probe-nodes and evaluation-edges affect the results? How should practitioners choose these numbers in practice?

(major) It is unclear how I should interpret Figure 3. It is not clearly visible why *sufficiently strong and early control attenuates both drift and energy*. Could you provide a more detailed explanation for this?

(minor) What is the actual runtime for the proposed protocol? Is it efficient enough to run along with the training?

(minor) Can you also provide experimental results across different embedding dimensions? I wonder how the embedding dimension affects the training dynamics.

---

> ### Author Response · Authors · 2026-06-03
> **Response to Reviewer Qyzb**
>
> We thank the reviewer for the constructive and detailed feedback. In the revision, we clarified the claim scope, introduced seed-aggregated uncertainty summaries and probe-resampling checks, quantified probe-node/evaluation-edge budget sensitivity, revised the interpretation of Figure 3, measured wall-clock runtime, and added a hidden-width sensitivity sweep.
>
> **Scope and dataset portability.** We agree that the previous presentation made the scope of evidence less visible than it should have been. We now separate two claims. The protocol-level claim is that fixed-point probing is a dataset-agnostic post-training diagnostic protocol using fixed probe nodes, fixed evaluation edges, and geometry-consistent metrics. The bounded case-study claim is that, on the patent stress test, the evaluated hyperbolic models exhibit a boundary-coupled late-depth regime. We do not claim dataset-universal thresholds, layerwise onsets, or mechanism strength. Appendix E.1, Tables E.1--E.2, the public Cora/Texas and OGBN-Arxiv checks, and the same-patent-subgraph GCN check make this scope boundary explicit.
>
> **Standard deviations and probe sampling.** We added seed-aggregated mean$\pm$std counterparts for the core diagnostics in Appendix E.2, Table E.3 and Figure E.2, and revised main-figure captions to state when a panel is a representative fixed-checkpoint trace. We keep the main mechanism panels as trajectory traces for readability, while making the corresponding uncertainty visible in the appendix. Appendix A.6 directly addresses probe-node resampling: the within-checkpoint std of final-transition drift is 0.088 at $L=2$, 0.176 at $L=4$, and below $3\times 10^{-3}$ for $L\geq 8$, with the depthwise regime ordering unchanged.
>
> **Probe-node and evaluation-edge budgets.** We added a budget sensitivity study over $|S|\in{1\mathrm{k},5\mathrm{k},10\mathrm{k},50\mathrm{k}}$ and $|E_{\mathrm{eval}}|\in{10\mathrm{k},50\mathrm{k},100\mathrm{k},200\mathrm{k}}$ in Appendix E.3. Increasing $|S|$ mainly reduces resampling variability: the within-checkpoint resampling std of drift-p50 decreases from about 0.025 at $|S|=1\mathrm{k}$ to about 0.003 at $|S|=50\mathrm{k}$. For Dirichlet energy, 10k and 200k evaluation-edge estimates differ by less than 0.3% in every tested mode/depth condition. We also added a practical recommendation: start with a small pilot budget, increase it until qualitative regime labels and rank/order summaries are stable, then fix and report the subsets.
>
> **Figure 3.** We agree that the original wording made the intervention appear cleaner than warranted. We revised both the labels and interpretation. The $\lambda=10$ condition is late-only/delayed, whereas $\lambda=100$ is applied globally from epoch 0. We added a pre-terminal window summary in Appendix E.5, Table E.8, because a terminal-window average is affected by terminal drop/collapse artifacts in the $\lambda=0$ trace. The revised claim is more cautious: late-only control remains boundary-saturated; $\lambda=100$ reduces the high-energy portion of the saturated trajectory in the pre-terminal window, but drift is nonmonotonic. We no longer claim that strong early control cleanly attenuates both drift and energy. We also added the mechanism link: local geodesic drift scales approximately as $\lambda(x)|\Delta x|$, while squared-distance Dirichlet energy scales approximately as $\lambda(x)^2$, so energy can respond more visibly than median drift.
>
> **Runtime.** Appendix E.4 reports wall-clock measurements. The full default diagnostic takes approximately 16.5--24.0 seconds per checkpoint across the tested default settings, with no backpropagation; a small pilot budget takes about 9--10 seconds. We clarify that the full probe is most naturally run at validation intervals or saved checkpoints, while small pilot budgets are suitable for lightweight training-time monitoring.
>
> **Hidden-width sensitivity.** We added a hidden-width sensitivity sweep over $d\in{64,128,256}$ at $L\in{16,32}$, with Euclid/HypA/HypT and three training seeds (Appendix E.6, Table E.9). This varies the GraphSAGE encoder hidden width while keeping the raw input text-feature dimension and output representation dimension fixed. Thus, it addresses sensitivity to encoder capacity. Absolute scales and endpoint metrics vary, but changing hidden width does not remove the severe $L=32$ degradation regime. If the reviewer instead meant the output representation dimension, that is a separate axis not claimed by this sweep.

---

> > ### Comment · Reviewer_Qyzb · 2026-07-13
> > **Additional comments**
> >
> > I appreciate the authors for the clarification.
> > However, I am still confused about the arguments regarding *dataset-agnostic post-training diagnostic protocol*. I do not see the results of the framework tested on other datasets and my concerns regarding the universality of the framework are not fully addressed in Appendix E.1.

---

> > > ### Author Response · Authors · 2026-07-14
> > > **Response to Reviewer Qyzb**
> > >
> > > We thank Reviewer Qyzb for the follow-up. We now see that our previous response did not
> > > distinguish clearly enough between successful protocol instantiation and dataset-universal
> > > regularity.
> > >
> > > **On the phrase "dataset-agnostic."** The phrase appeared in our earlier response as
> > > shorthand, not as a manuscript claim. We agree that it can be read as asserting a stronger
> > > dataset-universal regularity than we intend. We therefore withdraw the phrase and are
> > > removing the single negated occurrence from the manuscript so that no ambiguity remains.
> > >
> > > **On the location of the public results.** The reviewer is correct that Sections 6.1–6.7
> > > currently report the patent citation graph, while the public-dataset checks are confined to
> > > the appendices. A reader following the main empirical narrative would therefore not
> > > encounter a direct non-patent instantiation of the framework. We have corrected this
> > > organization: the revision will include a compact public-benchmark instantiation in the main
> > > Results section.
> > >
> > > **On the completeness of the public diagnostics.** The previous OGBN-Arxiv presentation did
> > > not assemble and report the complete probe family under the current manuscript definitions
> > > in one place. Figure C.1 emphasizes endpoint macro-F1, projected activation norm, and
> > > metric-aware Dirichlet energy, while geometry-consistent drift, global Eq. (15)
> > > separability, and curvature-normalized boundary statistics were not jointly presented under
> > > the current reporting convention.
> > >
> > > The missing step does not require new model training: all quantities can be obtained by a
> > > post-training replay of the existing checkpoints. We agree that the previous presentation
> > > left the cross-dataset evidence incomplete for evaluating the protocol contribution, and the
> > > revision corrects this by reporting the complete current readout directly in the main
> > > Results section.
> > >
> > > **What the revision will contain.** We are completing a replay of the current complete probe
> > > family on the existing OGBN-Arxiv GraphSAGE checkpoints: geometry-consistent drift,
> > > metric-aware Dirichlet energy, global Eq. (15) separability, and curvature-normalized
> > > boundary statistics. The replay uses the complete fixed OGBN-Arxiv test split, the released
> > > fixed symmetrized evaluation-edge set derived from the 200,000-edge sample, and the same
> > > metric definitions, common observation-space construction, and reporting conventions used
> > > for Table 2. The run metadata and validation reports record the probe-node and
> > > evaluation-edge hashes together with the probe-definition provenance.
> > >
> > > The existing reference checkpoints require no retraining. We are additionally completing
> > > matched training-seed repeats at the main reported depths so that the new main-text summary
> > > includes uncertainty. The revision will distinguish the existing seed-0 full-depth reference
> > > sweep from the depths for which three-seed mean-and-standard-deviation results are
> > > available. We will also add the OGBN-Arxiv-specific training and hyperparameter
> > > configuration, including the distinction between the OGBN-Arxiv HypT setting and the
> > > patent-study HypT setting. We are targeting an upload within one week and would be grateful
> > > if the reviewer could consider that revision before forming a final recommendation.
> > >
> > > **On claim scope.** The revised claim is successful instantiation of the complete protocol
> > > under the reported settings and the information provided by the joint probe readout. We do
> > > not claim dataset-universal thresholds, identical onset depths, or a shared failure
> > > mechanism.
> > >
> > > The revision will state the coverage precisely. The complete geometry-aware probe family,
> > > under the current definitions, is reported on the patent graph and on OGBN-Arxiv. Cora and
> > > Texas provide Euclidean depthwise checks — drift and class separability on both, with the
> > > metric-aware Dirichlet energy additionally reported for Cora — computed under the
> > > public-benchmark convention rather than the current global Eq. (15) definition; we will label
> > > them as such rather than present them as the same statistic. We would welcome any indication
> > > of a remaining claim–evidence mismatch after the revision so that we can address it directly.
> > >
> > > **Other requested changes.** For completeness, the remaining items from the original review
> > > have been addressed: seed-aggregated uncertainty summaries in Appendix E.2; probe-node and
> > > evaluation-edge budget sensitivity and practical guidance in Appendix E.3; the revised
> > > interpretation of Figure 3 in Appendix E.5; wall-clock runtime in Appendix E.4; and a
> > > hidden-width sensitivity analysis addressing encoder capacity in Appendix E.6, with the
> > > output representation dimension held fixed. If the reviewer considers any of these items
> > > still unresolved, we would appreciate knowing which item should receive priority.

---

> ### Author Response · Authors · 2026-07-15
> **Response to Reviewer Qyzb**
>
> We thank the reviewer for the follow-up. We agree that our previous revision did not
> make a complete non-patent instantiation visible enough to evaluate the protocol
> contribution. We have now uploaded a substantive revision addressing this point.
>
> **Complete main-text instantiation on OGBN-Arxiv.** Section 6.8, Figure 6, and Table 4
> now report the complete current-definition probe family for OGBN-Arxiv GraphSAGE over
> training seeds 0–2. The readout includes geometry-consistent drift, metric-aware
> Dirichlet energy, global Eq. (15) separability, curvature-normalized boundary
> statistics, and validation-selected endpoint performance. It uses the complete fixed
> test split of 48,603 nodes and one fixed symmetrized evaluation-edge set derived from
> the released 200,000-edge sample.
>
> This second complete instantiation also provides a result that is complementary to the
> patent case study. At $L=16$, boundary saturation coexists with useful endpoint
> performance. At $L=32$, the three HypT runs reach the same collapse-level endpoint but
> have different validation-selected drift, energy, separability, saturation-onset, and
> terminal-radius states. Thus, the protocol does not treat either boundary proximity,
> endpoint performance, or any individual probe as a universal failure label.
>
> **Claim scope.** We have removed the phrase "dataset-agnostic" from the manuscript. The
> revised claim is successful instantiation of the complete probe family under the
> reported settings on the patent graph and OGBN-Arxiv. We do not claim dataset-universal
> thresholds, identical onset depths, or a shared failure mechanism. Section 7 and
> Appendix E.1 state this coverage and limitation explicitly.
>
> **Other public-benchmark evidence.** Cora and Texas remain replicated Euclidean
> component checks under their released public-benchmark convention; their separability
> statistic is labeled accordingly and is not relabeled as the current global Eq. (15)
> statistic. The OGBN-Arxiv and patent GCN results provide additional backbone references
> rather than architecture-universal evidence.
>
> For completeness, the earlier requested uncertainty, probe-resampling, budget, runtime,
> Figure 3 interpretation, and hidden-width analyses remain reported in Appendices A.6 and
> E.2–E.6. The OGBN-Arxiv-specific training configuration is given in Table A.5.
>
> We hope that the main-text complete instantiation now resolves the concern that the
> protocol had only been demonstrated on the patent graph. We would appreciate any
> indication of a remaining mismatch between the stated claim and the evidence.

---

> > ### Comment · Reviewer_Qyzb · 2026-07-20
> > **Thanks**
> >
> > I sincerely appreciate the authors for clarifying the claim and adding experiments for another dataset. This resolves my main concerns regarding the paper.

---

### Review · Reviewer_JWzM · 2026-06-03

**Summary Of Contributions:**

This paper investigates the late-depth behavior of graph neural networks via fixed-point probing. The method fixes a set of nodes and a set of edges and aggregates several diagnostics in order to characterize late-depth behavior. This is in contrast to prior work that compute these separately and are unable to distinguish between various late-depth regimes. Evaluation is done on a large patent citation graph, for which the method suggests contrasting late-depth behavior between Euclidean and hyperbolic models.

Strengths
1. The paper identifies concrete late-depth phenomena that have been observed but not precisely distinguished or diagnosed in the literature, e.g. oversmoothing, oversquashing, stabilization.
2. The proposed protocol is a fairly simple and reasonable approach for diagnosis.
3. The writing is generally clear and easy to follow. There are explicit mentions of limitations and reproducibility details.

Weaknesses
1. While the authors explicitly acknowledge the limitations of their experimental results, it would significantly strengthen the paper to see what conclusions can be drawn by using the method on other datasets, architectures, or even message passing mechanisms, etc. The paper explicitly states to not make any dataset-universal claims, but such a claim, if supported by evidence via the proposed method, would be highly interesting.

**Audience:**

Yes

**Audience Explanation:**

This work would be of interest to the graph neural network community, whose members are certainly part of TMLR's audience. The primary contribution of this paper is a protocol that can be used and explored in a diverse range of GNN settings.

**Claims And Evidence:**

Yes

**Claims Explanation:**

Yes. I am particularly pleased with the care taken to avoid overclaiming, to the point of mentioning the limitations of the findings throughout. The authors also make a special effort to provide reproducibility details.

**Requested Changes:**

I would ideally like to see the weakness mentioned above addressed in some form. However, I believe the current manuscript as of this review is already a solid work, and these changes are not critical to my recommendation for acceptance.

(minor) The font in the appendix is large and should be the same as the main text.

---

> ### Author Response · Authors · 2026-06-04
> **Response to Reviewer JWzM**
>
> We thank the reviewer for the positive assessment and for highlighting the care taken to scope the claims. We address the two points raised below.
>
> **Appendix font (minor).** The enlarged appendix body text was caused by an unscoped font-size declaration in the appendix header that propagated to the appendix body. We corrected the formatting so that the appendix body is now typeset at the same size as the main text. We also adjusted the affected appendix table sizing to avoid unintended font enlargement.
>
> **Broader datasets, architectures, and message-passing mechanisms.** We agree that this is a valuable direction, and that a dataset-universal regularity, if supported by matching evidence, would be of broad interest. We therefore clarified how the revised manuscript speaks to portability, while deliberately stopping short of a dataset-universal mechanism claim.
>
> - The revised manuscript separates two claims: (i) a protocol-level claim that fixed-point probing is a dataset-agnostic post-training diagnostic using fixed probe nodes, fixed evaluation edges, and geometry-consistent metrics; and (ii) a bounded case-study claim that, on the patent stress test, the evaluated hyperbolic models exhibit a boundary-coupled late-depth regime.
> - Evidence for protocol portability across datasets and backbones is provided by the same-patent-subgraph Euclidean GCN check, and by the public Cora/Texas and OGBN-Arxiv reference checks with GraphSAGE and GCN backbones. Appendix E.1, in particular Tables E.1–E.2, makes explicit what these checks support: protocol portability and bounded scope. It also makes explicit what they do not support: dataset-universal thresholds, architecture-universal layerwise onsets, or a universal mechanism strength.
> - We refrain from asserting a dataset-universal mechanism law because the current evidence does not match such a claim. Asserting it would be the kind of overclaiming the reviewer noted we otherwise avoid; we prefer to keep the stated claims aligned with what the experiments establish.
> - The contribution is the protocol itself, which is provided with code and deterministic artifacts so that the community can apply it across additional datasets, architectures, and message-passing mechanisms. Mapping out which late-depth regularities are dataset-specific and which transfer is, in our view, an important use of the protocol rather than a prerequisite for it.
>
> To reflect this scope, the revised manuscript makes the relevant future-work directions explicit, including attention-based and diffusion-style backbones, alternative boundary-avoidance interventions, and probe-derived signals for model selection or early stopping. We hope this addresses the concern while keeping the claims matched to the evidence.

---

### Review · Reviewer_Hp57 · 2026-07-07

**Summary Of Contributions:**

** Resubmitting as a review***

General comments on the content. This paper tries to position itself as a GNN evaluation paper. That is, they bundle up existing metrics for diagnosing, for instance, oversmoothing in GNNs, adapt it to different geometries (e.g. hyperbolic) and report it with a slew of other metrics as a function of GNN depth (at each layer) to compute an "embedding trajectory". None of the components are really new, the claimed novelty is to try to view these metrics over embedding trajectories rather than at the end points.

Overall, I don't find the method particularly interesting. The method is purely descriptive ("the dirichlet energy goes up between layer X and Z"... but so what? The case against end point metrics, which is ultimately what we care about, is not too clear to me). The authors compare on one use case hyperbolic and Euclidean GNNs, and plot on the same graph, but I am not even sure the distance themselves are comparable / on the same scake--- a lot of the plots have the same shapes between Euclidean and Hyperbolic geometry. The authors spend a good amount of time insisting that they don't mean that their results are universal --- but currently, given the way the paper is written, it makes us wonder why we go through the trouble of reading it at all: if the insights behind euclidean and hyperbolic geometry are not going to hold, then why present it? Wouldn't it be better to show case how this type of method can, for a fixed type of GNN (e.g. Euclidean) lead us to make better choices in terms of depth, width, etc?

The paper is also extremely poorly written. A lot of the details could be put in the appendix -- we are currently drowning in repetitions, list of what the camera ready would like like, etc . But overall, the main problem is that the style is extremely abstruse. In the abstract alone: "the protocol reveals geometry-dependent late-depth regimes." "Taken together, the results point to a boundary-coupled late-depth regime in hyperbolic GNNs that is hard to isolate from endpoint metrics or from any single probe alone, but becomes visible when the probes are read jointly under a shared protocol". "... not as evidence for dataset-universal claims". => I honestly have no idea what the authors are talking about.

" endpoint metrics"? " geometry-consistent"? All these terms should be introduced.
" graph-local roughness"????? -" A radius-penalty perturbation then provides diagnostic evidence consistent with a boundary contribution to the observed amplification in the evaluated setting; "
Overall, this reads like an extensive project report, rather than a fully fleshed paper.

**Audience:**

Yes

**Audience Explanation:**

See comments

**Claims And Evidence:**

Yes

**Claims Explanation:**

NA --- see comments

**Requested Changes:**

See comments

---

> ### Author Response · Authors · 2026-07-08
> **Authors' note on resubmitted review**
>
> We thank Reviewer Hp57 for resubmitting the review in the appropriate form. Since the review indicates that it is a resubmission of the earlier official comment, we note that the substantive concerns were already discussed in the earlier public thread and follow-up.
>
> In particular, our previous responses addressed the protocol contribution beyond individual diagnostics, what fixed-point probing adds beyond endpoint metrics, the intended within-geometry reading of the Euclidean and hyperbolic comparisons, the operational terminology, and the diagnostic/model-selection scope of the practitioner-facing guidance.
>
> We have no further additions at this stage unless Reviewer Hp57 or the Action Editor would like clarification on a specific point not covered in that exchange.

---

### Comment · Reviewer_Hp57 · 2026-06-10

General comments on the content.
This paper tries to position itself as a GNN evaluation paper. That is, they bundle up existing metrics for diagnosing, for instance, oversmoothing in GNNs, adapt it to different geometries (e.g. hyperbolic) and report it with a slew of other metrics as a function of GNN depth (at each layer) to compute an "embedding trajectory". None of the components are really new, the claimed novelty is to try to view these metrics over embedding trajectories rather than at the end points.

Overall, I don't find the method particularly interesting. The method is purely descriptive ("the dirichlet energy goes up between layer X and Z"... but so what? The case against end point metrics, which is ultimately what we care about, is not too clear to me). The authors compare on one use case hyperbolic and Euclidean GNNs, and plot on the same graph, but I  am not even sure the distance themselves are comparable / on the same scake--- a lot of the plots have the same shapes between Euclidean and Hyperbolic geometry.  The authors spend a good amount of time insisting that they don't mean that their results are universal --- but currently, given the way the paper is written, it makes us wonder why we go through the trouble of reading it at all: if the insights behind euclidean and hyperbolic geometry are not going to hold, then why present it? Wouldn't it be better to show case how this type of method can, for a fixed type of GNN (e.g. Euclidean) lead us to make better choices in terms of depth, width, etc?

The paper is also extremely poorly written. A lot of the details could be put in the appendix -- we are currently drowning in repetitions, list of what the camera ready would like like, etc . But overall, the main problem is that the style is extremely abstruse. In the abstract alone:
 "the protocol reveals geometry-dependent late-depth regimes." "Taken together, the results point to a boundary-coupled late-depth regime in hyperbolic GNNs that is hard to isolate from endpoint metrics or from any single probe alone, but becomes visible when the probes are read jointly under a shared protocol". "... not as evidence for dataset-universal claims".
=> I honestly have no idea what the authors are talking about.
- " endpoint metrics"? " geometry-consistent"? All these terms should be introduced.
- " graph-local roughness"?????
-" A radius-penalty perturbation then provides diagnostic evidence consistent with a boundary contribution to the observed amplification in the evaluated setting; "


Overall, this reads like an extensive project report, rather than a fully fleshed paper.

---

> ### Author Response · Authors · 2026-06-10
> **Response to Reviewer Hp57**
>
> We thank Reviewer Hp57 for the comments. We read the main concerns as the contribution beyond existing diagnostics, what fixed-point probing adds beyond endpoint metrics, cross-geometry comparability, and terminology.
>
> We do not claim novelty for each individual diagnostic. The contribution is the **fixed-point probing protocol**: fixed probe nodes, fixed evaluation edges, geometry-consistent measurements on observed representations, and a joint interpretation rule across drift, energy, separability, and boundary pressure. This joint reading is what allows the protocol to distinguish late-depth regimes rather than only reporting a final downstream metric or a single probe. In the patent-citation stress test, the protocol distinguishes benign stabilization, Euclidean oversmoothing-like degradation, and a hyperbolic boundary-associated regime, without claiming dataset-universal thresholds or universal layerwise onsets.
>
> Raw Euclidean and hyperbolic distances or energies are not intended to be directly commensurate. The comparisons are within-geometry depth trends, regime shapes, and probe couplings, not absolute cross-geometry magnitudes. This is the intended reading of the current manuscript and is stated explicitly in the metric-scale paragraph in Section 4.1, in Section 6.1, and in the captions of Figure 1 and Table 2.
>
> The terminology follows the manuscript's operational usage. **Endpoint metrics** refer to endpoint performance metrics, with macro-F1 used as the primary endpoint performance metric and accuracy reported as a reference metric in Appendix D. **Geometry-consistent** means that probes are evaluated on observed representations under the metric induced by the target geometry, as specified in Section 4. **Graph-local roughness** refers to the metric-aware Dirichlet energy on the fixed evaluation edges, as described in Section 4.2 and Appendix D.3. The **boundary-coupled late-depth regime** refers to the Section 6.2 pattern in which boundary proximity co-amplifies geodesic drift and metric-aware graph-local roughness in the evaluated hyperbolic models.

---

> > ### Comment · Reviewer_Hp57 · 2026-06-11
> > **Re**
> >
> > Thank you.
> > The text, as written, makes very clear that you’re not suggesting a new method but a protocol — in fact you repeat this on at least 3 separate occasions throughout the manuscript. However, in my opinion, this is not written as a protocol, but as a project report on a specific dataset. You report plots of diagnostics, but it is unclear how to translate this into actionable feedback for the algorithm (eg should we go deeper? Wider? Use another convolution?). I am not getting how this can help any practioner deploy GNNs on their dataset.
> > As for the definitions— thank you.  The reader eventually gets them from context, but in my opinion, the manuscript needs a complete overhaul both from the content and the style perspective before it can be deemed ready for review.

---

> > > ### Author Response · Authors · 2026-06-11
> > > **Follow-up on practitioner-facing scope**
> > >
> > > We thank Reviewer Hp57 for the follow-up. We understand the remaining concern as whether the protocol provides practitioner-facing guidance rather than only descriptive plots.
> > >
> > > Our intended scope is diagnostic and model-selection oriented, not an automatic architecture-design algorithm. Fixed-point probing is applied to a candidate sweep of depths, geometries, or backbones with the probe nodes and evaluation edges held fixed, and its output is the onset and type of the late-depth regime. A practitioner therefore sees not only that a deeper run has worse endpoint performance, but whether the degradation is accompanied by separability loss, drift and energy amplification, or boundary proximity. The corresponding practitioner-facing uses are concrete but limited: stop short of depths past the observed unstable regime, compare candidate models by the stability of their probe trajectories, and judge whether an observed failure is consistent with oversmoothing-like degradation or with a geometry-specific boundary-associated pattern in the evaluated hyperbolic runs.
> > >
> > > We do not claim that the protocol by itself prescribes interventions such as "go deeper," "increase width," or "switch the convolution." Such intervention policies are downstream uses that require separate intervention evidence, and asserting them here would go beyond the evidence presented in the manuscript. The contribution we do claim is the fixed diagnostic protocol, together with its budget and runtime guidance and bounded case-study evidence of what it reveals beyond endpoint performance. This is the level of claim that we believe is supported by the current manuscript.

---

### Author Response · Authors · 2026-07-15
**Revised manuscript uploaded: main-text public-benchmark instantiation and reading notes**

Dear reviewers and Action Editor,

We have uploaded a revised manuscript that primarily responds to Reviewer Qyzb's
follow-up by reporting a complete non-patent instantiation of the protocol in the main
Results section (Section 6.8, Figure 6, Table 4; OGBN-Arxiv GraphSAGE, three training
seeds).

For readers interested in how the protocol informs concrete choices, Protocol 1
("Practitioner use") and the OGBN-Arxiv candidate sweep in Section 6.8 make the intended
within-geometry model-selection use explicit: for a fixed backbone and geometry, the
workflow retains the deepest stable candidate and flags deeper candidates that add no
endpoint benefit (for example, retaining L≤16 and flagging L=32 on OGBN-Arxiv). It is a
diagnostic and model-selection instrument, not an automatic prescription to change depth,
width, or convolution.

We also note that Euclidean and hyperbolic magnitudes are not treated as commensurate.
All cross-geometry statements concern within-geometry regime shapes and probe couplings
(Section 4.1; Table 1 caption), and the L=32 contrast is stated as disjoint drift-ratio
ranges rather than as a comparison of absolute magnitudes (Section 6.2; Table 2).

We would be grateful if the reviewers could consider this revision before submitting
their final recommendations, and we remain available for any further clarification.

Best regards,
The Authors